# Hydrodynamic and biochemical impacts on the development of hypoxia in the Louisiana–Texas shelf Part 1: roles of nutrient limitation and plankton community

Yanda Ou[1,2] and Z. George Xue[1,2,3]

[1]Department of Oceanography and Coastal Sciences, Louisiana State University, Baton Rouge, LA, 70803, USA.
[2]Center for Computation and Technology, Louisiana State University, Baton Rouge, LA, 70803, USA.
[3]Coastal Studies Institute, Louisiana State University, Baton Rouge, LA, 70803, USA

*Correspondence to*: Z. George Xue (zxue@lsu.edu)

**Abstract.** A three-dimensional coupled hydrodynamic–biogeochemical model with multiple nutrient and plankton functional groups was developed and adapted to the Gulf of Mexico to investigate the role of nutrients and the complexity of plankton community in dissolved oxygen (DO) dynamics. A 15-year hindcast was achieved covering the period of 2006–2020. Extensive model validation against *in situ* data demonstrates that the model was capable of reproducing vertical distributions of DO, spatial distributions of bottom DO concentration, as well as their interannual variations. The study demonstrates that bottom DO dynamics and hypoxia evolution are significantly influenced by both physical processes and local biochemistry, with sedimentary oxygen consumption and vertical diffusion identified as key contributors. Summer hydrodynamics play a critical role in nutrient distribution and limitation: a notable expansion of Si limitation was simulated when coastal currents shifted eastward or northward. This effect, especially pronounced on the western part of the Louisiana-Texas shelf, underscores the importance of nutrient limitation in shaping DO dynamics. The model identifies a bi-peak primary production pattern in spring and early summer, aligned with satellite chlorophyll *a* variations, attributed to the complexity of the plankton community and interactions among different plankton groups. Our findings emphasize the necessity of integrating sophisticated plankton community dynamics into biogeochemical models to understand primary production variability and its impact on bottom hypoxia.

## 1 Introduction

The Louisiana–Texas (LaTex) shelf in the northern Gulf of Mexico (nGoM) has one of the most notorious recurring hypoxia in the world (bottom dissolved oxygen (DO) < 2 mg $L^{-1}$, Rabalais et al., 2002; Rabalais et al., 2007a; Justić and Wang, 2014). Historical observations show that hypoxia usually emerges in mid-May and persists through mid-September (Rabalais et al., 1999, 2002). The hypoxic zone can cover as big as 23,000 $km^2$ and has a volume of up to 140 $km^3$ (Rabalais and Turner, 2019; Rabalais and Baustian, 2020). Although nitrogen (N) is the ultimate limiting nutrient, phosphorus (P) load reduction also leads to a significant reduction of the hypoxia area (Fennel and Laurent, 2018). Transient P limitation on the shelf (Laurent

et al., 2012; Sylvan et al., 2007) was deemed to be associated with the delayed onset and reduction of the hypoxia area.
Sensitivity experiments of hypoxia area reduction to different nutrient reduction strategies suggested that to meet the hypoxic
area reduction goal ($< 5,000$ km$^2$ in a 5-year running average) set by the Hypoxia Task Force (2008), a dual nutrient strategy
with a reduction of 48 % of total N and inorganic P would be the most effective way (Fennel and Laurent, 2018).
Coastal eutrophication in the LaTex shelf leads to a high rate of microbial respiration and depletion of DO (Rabalais et al.,
2007b). Incubation studies in the LaTex shelf suggested that sediment oxygen consumption (SOC) accounted for $20\pm4$ %
(Murrell and Lehrter, 2011) to $25\pm5.3$ % (McCarthy et al., 2013) of below-pycnocline respiration, nearly 7-fold greater than
the corresponding percentage in waters overlying sediments ($3.7\pm0.8$ %, about 20 cm above sediments in McCarthy et al.,
2013). The numerical study by Fennel et al. (2013) calculated the corresponding SOC fraction, which reached 60 % when
applying the water respiration rates of Murrell and Lehrter (2011) and sediment respiration rates of Rowe et al. (2002). Another
numerical study (Yu et al., 2015) also pointed out that on the LaTex shelf, oxygen consumption at the bottom water layer was
more associated with SOC rather than water column respiration. According to in-situ data and statistical analysis, SOC can be
estimated using the bottom temperature and DO concentration (e.g., Hetland and DiMarco, 2008). Nevertheless, many
numerical studies treated SOC only associated with the abundance of organic matter in the sediment (e.g., Justić and Wang,
2014; Fennel et al., 2006; 2011). An instantaneous remineralization parameterization by Fennel et al. (2006, 2011) estimated
SOC as a function of sediment detritus and phytoplankton. Using this scheme, Große et al. (2019) found that the simulated
SOC was supported by Mississippi N supply ($51\pm9$ %), Atchafalaya N supply ($33\pm9$ %), and open-boundary N supply ($16\pm2$
%). However, the instantaneous remineralization parameterization tends to overestimate SOC at the peak of phytoplankton
blooms while underestimate SOC after the blooms. In a realistic environment, there should be a lag between the blooms and
the peak SOC (Fennel et al., 2013). Developments of coupled sediment–water models emphasized the importance of
biogeochemical processes in sediments on the SOC dynamics and evolution of bottom hypoxia in the shelf (Moriarty et al.,
2018; Laurent et al., 2016). However, coupled sediment–water models are computationally more expensive than a simplified
parameterization of SOC. Especially for long-term simulations and time-sensitive forecasts, it is crucial to balance the model's
efficiency with its complexity.
In addition to SOC and excess nutrient supply from the rivers, water column stratification also plays an important role in
regulating the variability of bottom DO concentration in the LaTex shelf. Strong stratification prohibits DO ventilation and
thus reduces DO supply to the bottom water layer (Hetland and DiMarco, 2008; Bianchi et al., 2010;  Fennel et al., 2011, 2013,
2016; Justić and Wang, 2014; Wang and Justić, 2009; Feng et al., 2014; Yu et al., 2015; Laurent et al., 2018). On the shelf,
the Mississippi and the Atchafalaya plume introduce buoyancy, leading to a stable water column and weak DO ventilation
processes (Mattern et al., 2013; Fennel and Testa, 2019). Due to the different distances from major river mouths, the influence
of freshwater-induced buoyancy varies along the shelf. Moreover, the transport and deposition processes of organic matter are
affected by the coastal along-shore current systems, resulting in a SOC gradient across the shelf. For instance, Hetland and
DiMarco (2008) pointed out that in the west of Terrebonne Bay, where stratification is usually weak, bottom hypoxia is mainly
controlled by bottom respiration.

The phytoplankton blooms on the LaTex shelf mainly result from cyanobacteria and diatoms (Wawrik and Paul, 2004;
Schaeffer et al., 2012; Chakraborty et al., 2017). In the Mississippi River plume, diatoms were found as the most diverse algal
class accounting for over 42 % of all unique genotypes observed (Wawrik and Paul, 2004). Cruises data in the nGoM indicated
that diatoms accounted for ~50 to ~65 % (inner-shelf) and ~33 to ~64 % (mid-shelf) of chlorophyll *a* in winter and spring, and
~30 % to ~46 % (inner-shelf) during summer and fall, respectively (Chakraborty and Lohrenz, 2015). A field survey
documented that the biovolume contribution of diatoms to the total phytoplankton could be as high as 80 % and 70 % during
the upwelling seasons in 2013 and 2014, respectively (Anglès et al., 2019). While a lot of existing studies indicated N and P
were more limited than silicon (Si) on the shelf (e.g., for cruises in 2004 in Quigg et al., 2011; for cruises in 2012 in Zhao and
Quigg, 2014; for cruises in 1984, 1994, 2005, 2010, and 2011in Turner and Rabalais, 2013), Si limitation has also been reported
in both plume and shelf water. A bioassay study on sampled collected in spring and summer 2004 showed signs of co-limitation
of N, P, and Si at multiple sites (Quigg et al., 2011). Based on cruises studies in the plume of the Mississippi River in 1992
and 1993, strong Si limitation in spring was found due to the increasing N:Si ratio in the Mississippi River water (Nelson and
Dortch, 1996). Cruise measurements in 1987 and 1988 also suggested the likelihood of Si limitation, which sometimes
overwhelmed the N limitation (Dortch and Whitledge, 1992).

Numerical studies for hypoxia in the LaTex shelf were developed mostly incorporating nutrient flows of N and P only (e.g.,
Fennel et al., 2006, 2011, 2013; Laurent et al., 2012; Laurent and Fennel, 2014; Fennel and Laurent, 2018; Justić et al., 2003;
Justić et al., 2007; Justić and Wang, 2014; Große et al., 2019; Moriarty et al., 2018). In addition, many existing models utilized
an over-simplified lower trophic level model (one phytoplankton + one zooplankton function group or only one phytoplankton
group). The recycling of nutrients in water columns and the associated biogeochemical processes, which may be important to
hypoxia evolution (e.g., in the Chesapeake Bay by Testa and Kemp, 2012), could be over-simplified. Moreover, we noticed
that there was a bi-peak primary production pattern observed by satellite and modeled by Gomez et al. (2018) (see comparisons
of modeled and satellite-derived chlorophyll *a* concentration in that work). Their biogeochemical model incorporated a more
complex community (two phytoplankton + three zooplankton function groups) than other over-simplified models where the
bi-peak pattern was hardly captured (e.g., Fennel et al., 2011). The temporal variation of shelf primary production can further
induce corresponding changes in DO concentration and in the bottom hypoxia. In this study, we aimed to investigate the
possible Si limitation and to assess the impacts of the complexity of the plankton community on DO dynamics and bottom
hypoxia development. We adapted and modified a coupled physical-biogeochemical model covering the entire Gulf of Mexico
(GoM) by introducing the oxygen and P cycles to the North Pacific Ecosystem Model for Understanding Regional
Oceanography (NEMURO, Kishi et al. 2007). The model has two phytoplankton and three zooplankton functional groups for
a more comprehensive representation of the plankton community. We also modified the instantaneous remineralization
parameterization by adding a conceptual sedimentary organic pool (represented by a sedimentary particulate organic N pool,
$PON_{sed}$; Fig. 1) to allow the accumulation of organic matter in the sediment. The influence of the community is represented in
the biogeochemical processes in water columns and sediments and will eventually be reflected in the bottom DO variability.
**2 Methods**
**2.1 Coupled hydrodynamic–biogeochemical model**
We adapted the three-dimensional, free-surface, topography-following community model, the Regional Ocean Model System
(ROMS, version 3.7), on the platform of Coupled Ocean–Atmosphere–Wave–Sediment Transport (COAWST) modeling
system (Warner et al., 2010) to the GoM (Gulf–COAWST). ROMS solves finite difference approximations of Reynolds
Averaged Navier–Stokes equations by applying hydrostatic and Boussinesq approximations with a split explicit time-stepping
algorithm (Haidvogel et al., 2000; Shchepetkin and McWilliams, 2005, 2009). The biogeochemical model applied is primarily
based on the NEMURO developed by Kishi et al. (2007). NEMURO is a concentration-based, lower-trophic-level ecosystem
model developed and parameterized for the North Pacific. The original NEMURO model has 11 concentration-based state
variables, including nitrate ($NO_3$), ammonium ($NH_4$), small and large phytoplankton biomass (PS and PL), microzooplankton,
mesozooplankton, and predatory zooplankton biomass (ZS, ZL, and ZP), particulate and dissolved organic N (PON and DON),
particulate silica (Opal), and silicic acid ($Si(OH)_4$). NEMURO is known for its capability to distinguish ZS, ZL, and ZP and to
provide a detailed analysis of the dynamics of different functional groups. It was widely used in studies of plankton biomass
on regional scales (Fiechter and Moore 2009; Gomez et al., 2018; Shropshire et al., 2020). The embedded Si cycle permits the
inclusion of a diatom group (i.e., PL), one of the dominant phytoplankton groups in the LaTex shelf.
**2.2 Model modification**
In a recent effort, Shropshire et al. (2020) adapted and modified NEMURO to the GoM with five structural changes. (1) The
grazing pathway of ZL on PS was removed since, in the GoM, the PS group is predominated by cyanobacteria and
picoeukaryotes, which are too small for direct feeding by most mesozooplankton (i.e., ZL). (2) Linear function of mortality
was applied for PS, PL, ZS, and ZL, while quadratic mortality was used for ZP, accounting for predation pressure of unmodeled
predators, like planktivorous fish. (3) The ammonium inhibition term in the nitrate limitation function was no longer considered
exponentially but followed the parameterization by Parker (1993). (4) Light limitation on photosynthesis was replaced with
Platt et al.'s (1980) functional form, which was also implemented in the newer version of NEMURO. (5) Constant C: Chl ratio
was replaced with a variable C: Chl model according to the formulation by Li et al. (2010).

Neither the modified (Shropshire et al., 2020) nor the original (Kishi et al., 2007) NEMURO model considered P and oxygen
cycles. In this study, we introduced a P cycle into NEMURO, including three concentration-based state variables: phosphate
($PO_4$), particulate organic P (POP), and dissolved organic P (DOP). The P limitation on phytoplankton growth was introduced
using the Michaelis–Menten formula. In the NEMURO model, N serves as the common "currency" when measuring the
plankton concentration (mmol N m$^{-3}$). In the river-dominated LaTex shelf, rivers supply inorganic and organic nutrients. In
our model, riverine $PO_4$ (Fig. C1c), DOP, and POP were prescribed based on water quality measurements at river gages. When
no measurement was available, the $PO_4$, DOP, and POP were approximated using total nitrate+nitrite ($NO_3$+$NO_2$), dissolved
organic N (DON), and particulate organic N (PON) measurements, respectively, via the Redfield ratio of P: N=1: 16. We
neglected the POP settling process but preserved these pools by introducing the stoichiometric ratio between P and N instead.
In other words, the sinking process of POP is implicitly included by building linkages between PON and POP concentrations,
as the sinking of PON is considered in the model. Governing equations for P state variables are given according to Eqs. 1–3.
Please also refer to the appendices for more details on expressions of modified terms (Appendix A), state variables (Appendix
Table B1), source and sink terms (Appendix Table B2), and values of parameters (Appendix Table B4).
$$\frac{d(PO_4)}{dt} = (ResPSn + ResPLn) \cdot RPO4N$$
$$+(DecP2N + DecD2N) \cdot RPO4N$$
$$+(ExcZSn + ExcZLn + ExcZPn) \cdot RPO4N$$

139 $$-(GppPSn + GppPLn) \cdot RPO4N, \tag{1}$$

144 $$\frac{d(DOP)}{dt} = (DecP2D - DecD2N) \cdot RPO4N$$

$$+(ExcPSn + ExcPLn) \cdot RPO4N, \tag{2}$$
$$\frac{d(POP)}{dt} = (MorPSn + MorPLn + MorZSn + MorZLn + MorZPn) \cdot RPO4N$$
$$+(EgeZSn + EgeZLn + EgeZPn) \cdot RPO4N$$
$$-(DecP2N + DecP2D) \cdot RPO4N, \tag{3}$$

We further adapted the oxygen cycle developed by Fennel et al. (2006, 2013) to NEMURO for hypoxia simulations. However,
our model's biogeochemical processes are slightly different due to the different plankton functional groups considered. Sources
for oxygen are contributed by the photosynthesis of two phytoplankton functional groups. In comparison, the sinks are
attributed to respirations of two phytoplankton functional groups, metabolism of three zooplankton functional groups, light-
dependent nitrification (Olson, 1981; Fennel et al., 2006), aerobic decomposition of particulate and dissolved organic matter
(measured as PON, and DON, respectively), and SOC. Oxygen air–sea flux was estimated following parameterizations by
Wanninkhof's (1992). The biogeochemical dynamics of oxygen were adopted as follows (Eq. 4; also see detailed descriptions
of variables and parameters in Appendix A–B):
$$\frac{d(Oxyg)}{dt} = (rOxNO_3 \cdot GppNPS + rOxNH_4 \cdot GppAPS)$$
$$+(rOxNO_3 \cdot GppNPL + rOxNH_4 \cdot GppAPL)$$
$$- ResPSn \cdot [RnewS \cdot rOxNO_3 + (1 - RnewS) \cdot rOxNH_4]$$
$\qquad -ResPLn \cdot [RnewL \cdot rOxNO_3 + (1 - RnewL) \cdot rOxNH_4]$
$\qquad -rOxNH_4 \cdot (ExcZSn + ExcZLn + ExcZPn)$
$\qquad -2 \cdot Nit \cdot LgtlimN \cdot \hat{r}$
$\qquad -rOxNH_4 \cdot (DecD2N + DecP2N) \cdot \hat{r}$

160 $\qquad -SOC \cdot THK_{bot},$ $\hfill$ (4)


A $PON_{sed}$ pool due to vertical sinking processes of PON was introduced for parameterization of SOC. The SOC scheme (Fennel
et al., 2006) is known as the instantaneous consumption of DO. As soon as the PON falls into the sediment bed, PON will be
decomposed instantaneously. This scheme tends to overestimate SOC at the peak of blooms and to underestimate SOC after
blooms since the lag in SOC demand is neglected (Fennel et al., 2013). We considered such temporal delays in SOC by
introducing a $PON_{sed}$ pool. A portion of the PON ends with $PON_{sed}$, while the rest is buried ($PON_{burial}$) and removed from the
system. The parameterization is shown in the following. 1) Organic matter settling down at the conceptual sediment layer is
remineralized at a temperature-dependent aerobic remineralization rate, $K_{P2N}$. 2) Sediment oxygen is consumed only in the
oxidation of sedimentary organic matter (represented by $PON_{sed}$) and the nitrification of ammonium to nitrate (Fennel et al.,
2006). 3) Oxygen consumption at the conceptual sediment layer directly contributes to oxygen concentration decreases only
at the bottom water column. 4) Sediment denitrification is linearly related to SOC according to observational-based estimates
by Seitzinger and Giblin (1996), but the relationship was modified by Fennel et al. (2006) with a slightly smaller slope of
denitrification on SOC rate, i.e.,
$denitrification\ (mmolN\ m^{-2}\ day^{-1}) = 0.105 \times SOC(mmolO_2\ m^{-2}\ day^{-1}),$ $\hfill$ (5)
5) Aerobic decomposition of $PON_{sed}$, sediment nitrification, and denitrification follow chemical equations according to
(Fennel et al., 2006):
$C_{106}H_{263}O_{110}N_{16}P + 106O_2 \leftrightarrow 106CO_2 + 16NH_4 + H_2PO_4 + 122H_2O,$ $\hfill$ (R1)
$NH_4 + 2O_2 \rightarrow NO_3 + 2H + H_2O,$ $\hfill$ (R2)
$C_{106}H_{263}O_{110}N_{16}P + 84.8HNO_3 \rightarrow 106CO_2 + 42.4N_2 + 16NH_3 + H_3PO_4 + 148.4H_2O,$ $\hfill$ (R3)
6) Nitrate produced in sediments (Eq. R2) is used for denitrification (Eq. R3). The linear assumption in 4) implicitly builds
relationships among the reactions listed in assumption 5). Let's assume that the production rate of $NH_4$ by aerobic
decomposition (Eq. R1) of organic matter is M mmol $m^{-3}$ $day^{-1}$, and that the fraction of denitrification-produced $CO_2$ (Eq. R3)
to the total $CO_2$ production (Eqs. R1 and R3) is $x$. According to the linear assumption abovementioned, the consumption rate
of $NO_3$ during denitrification (Eq. R3) is proportional to the total consumption rate of $O_2$ in the sediment (Eqs. R1 and R2),
yielding $\frac{84.8Mx}{16(1-x)} = 0.105 \times \left[\frac{106M}{16} + \frac{84.8Mx}{8(1-x)}\right]$ and further $x \approx 0.1425$. The oxygen consumption rate (Eq. 6) and organic matter
consumption rate (Eq. 7) due to the coupled aerobic decomposition, nitrification, and denitrification processes can be obtained
by substituting the $x$ value into the stoichiometric ratios according to Eqs. R1–R3.
$Oxyg_{consumption} = \frac{106M}{16} + \frac{84.8Mx}{8(1-x)} = 8.3865M,$ $\hfill$ (6)
$\quad OM_{consumption} = \frac{M}{16} + \frac{Mx}{16(1-x)} = 0.0729M,$ (7)
Accordingly, the SOC and consumption rate of $PON_{sed}$ are given, respectively as follows:
$\quad SOC = Oxyg_{consumption} \cdot THK_{bot} = 8.3865M \cdot THK_{bot},$ (8)
$\quad PON_{sed_{consumption}} = 16 \cdot OM_{consumption} \cdot THK_{bot} = 1.1662M \cdot THK_{bot},$ (9)
where,
$\quad M = \frac{PON_{sed} \cdot VP2N_0 \cdot exp(K_{P2N} \cdot TMP)}{THK_{bot}},$ (10)
$\quad THK_{bot} = thickness\ of\ bottom\ water\ column,$ (11)

We further added light inhibition to nitrification and aerobic decomposition. These parametrizations were applied following
descriptions by Fennel et al. (2006, 2013). For the oxygen-dependent term, an oxygen threshold is specified below which no
aerobic respiration or nitrification occurred. Detailed equations are listed in Appendix A. The structure of the newly modified
NEMURO model is shown in a schematic diagram in Fig. 1.

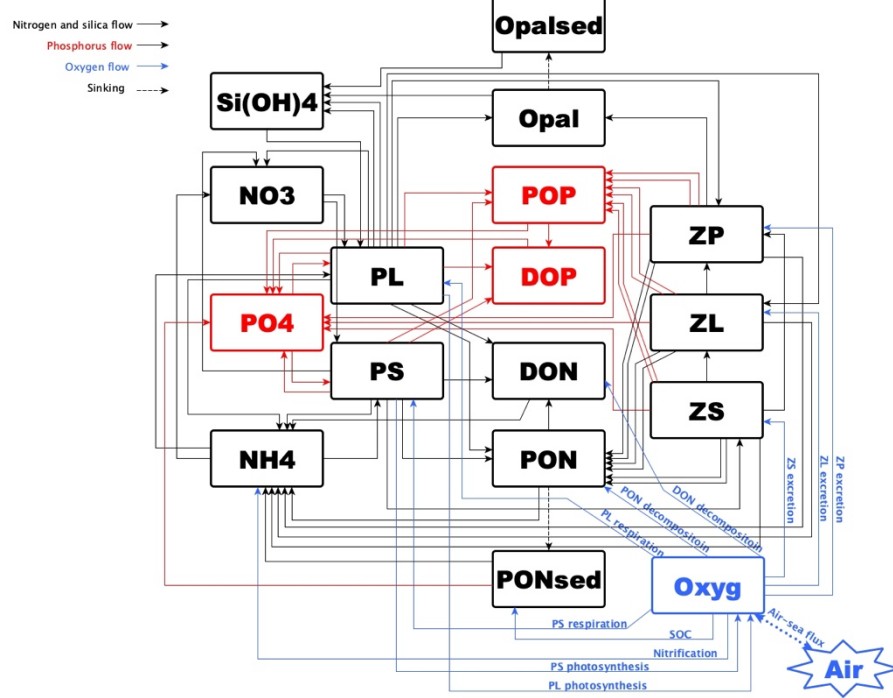


**Figure 1. Schematic diagram of the modified NEMURO model. Note that the P flow and the oxygen flow are two newly added flows**
**to the original NEMURO model.**

## 2.3 Model set-ups

The coupled model was applied to the GoM using Arakawa C-grid with a horizontal resolution of ~5 km (Fig. 2a). There are 334 and 357 interior rho points in the east-west and north-south directions, respectively. The model includes 36 sigma layers vertically. The wetting and drying scheme (Warner et al., 2013) was implemented to provide a more accurate representation of shallow water. The computational time step (i.e., baroclinic time step) was set to 240 seconds, while the number of barotropic time steps between each baroclinic time step was set to 30. Model hindcast was carried out from 1 August 2006 to 26 August 2020, with the first five months as a spin-up period. Model historical and averaged results were output at a daily interval, while the historical fields were output at UTC 00: 00 each day.

The physical model set-ups largely followed an earlier Gulf–COAWST application (Zang et al., 2018, 2019, 2020). Open boundaries were set at the south and east forced by daily water level, horizontal components of 3-D current velocity, horizontal components of depth-integrated current velocity, 3-D water salinity, and 3-D water temperature derived from the Hybrid Coordinate Ocean Model (HYCOM) global analysis products (Bleck and Boudra, 1981; Bleck, 2002) with data assimilated via the Navy Coupled Ocean Data Assimilation system (Cummings, 2005; Cummings and Smedstad, 2013; Fox et al., 2002; Helber et al., 2013). For lateral boundary conditions, we utilized Chapman implicit for free surface and water level (Chapman, 1985), Flather for depth-integrated momentum (Flather, 1976), gradient for mixing total kinetic energy, and mixed radiation-nudging conditions for 3-D momentum, temperature, and salinity (Marchesiello et al., 2001). The nudging time steps for the mixed radiation-nudging condition were set to 1 day for inflows and 30 days for outflows. The boundary nudging technique was performed at the computational grids along the open boundary. The boundary condition types for passive biological and chemical tracers (i.e., PS, PL, ZS, ZL, ZP, $NO_3$, $NH_4$, PON, DON, $Si(OH)_4$, opal, $PO_4$, POP, DOP, and Oxyg) were all prescribed as radiation.

Initial conditions for water level, horizontal components of 3-D current velocity, horizontal components of depth-integrated current velocity, 3-D water salinity, and 3-D water temperature were provided by the same HYCOM products as well. Initial conditions for concentrations of $NO_3$, $PO_4$, and $Si(OH)_4$ were interpolated from measurements provided by the World Ocean Database (WOD, Boyer et al., 2018). Initial conditions for DO concentration were given by World Ocean Atlas (WOA, Garcia et al., 2018). At the sediment layer, $PON_{sed}$, $PON_{burial}$, $opal_{sed}$, and $opal_{burial}$ were initialized as 0.1 mmol $m^{-3}$. Other biological and chemical tracers were initialized as 0.1 mmol $m^{-3}$ due to the lack of observations.

Atmospheric forcings, including surface wind velocity at 10 m height above sea level, net longwave radiation flux, net shortwave radiation flux, precipitation rate, air temperature 2 m above sea level, sea surface air pressure, and relative humidity 2 m above sea level, were derived from the National Centers for Environmental Prediction (NCEP) Climate Forecast System Reanalysis (CFSR) 6-hourly products (for years prior to 2011, Saha et al., 2010) and NCEP CFS Version 2 (CFSv2) 6-hourly

products (for years starting from 2011, Saha et al., 2011) with a horizontal resolution of ~35 km and ~22 km, respectively. In
our model, 63 rivers were considered as horizontal point source forcings along the coastal GoM. They were split into 280
points (red dots in Fig. 2a) sources transporting time-varying salinity (nearly zero), temperature, 3-D horizontal momentum
(based on the magnitude of river discharges), nutrients ($NO_3$, $NH_4$, $PO_4$, $Si(OH)_4$, PON, DON, POP, and DOP; Fig. C1), and
DO to the computational domain. Locations of river point sources of the Mississippi and the Atchafalaya Rivers were shown
as red dots in Fig. 2b. For reconstructions of time series of river forcing terms, we composed measurements from various
sources, including U.S. Geological Survey (USGS) National Water Information System (NWIS), National Oceanic and
Atmospheric Administration (NOAA) Tides and Currents System (TCS), NOAA National Estuarine Research Reserve System
(NERRS), and Mexico National Water Commission (CONAGUA, for rivers in Mexico's territory). Daily averaged river
discharges were given based on measurements by USGS NWIS and CONAGUA. The magnitude of river discharges was
multiplied by 1.4 to account for adjacent watershed areas and the lateral inflow of tributaries (Warner et al., 2005).  River
temperature and salinity time series were reconstructed from measurements by USGS NWIS, NOAA TCS, and NOAA
NERRS. River nutrient concentrations were provided monthly by USGS NWIS and NOAA NERRS and were extended to
daily time series with values in the corresponding months. Riverine DO concentration was set to be a constant (258 mmol m$^-$
$^3$), assuming that riverine DO was saturated at 25 ℃ under 1 atm. Besides, tidal forcings were introduced in the hydrodynamic
model, taking into account the influences of tidal elevations and tidal currents. There were 13 tidal constituents considered in
the model including M2, S2, N2, K2, K1, O1, P1, Q1, MF, MM, M4, MS4, and MN4.

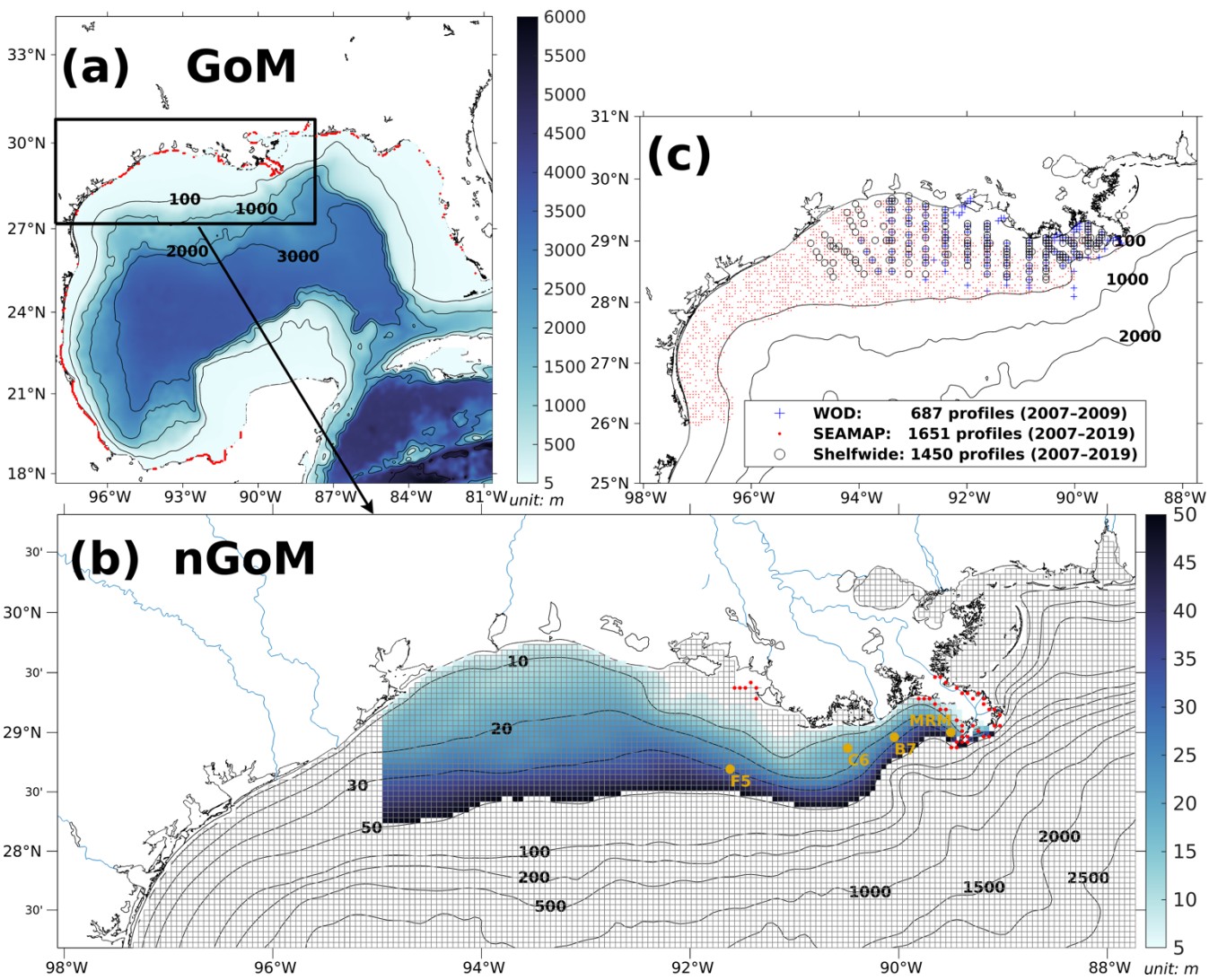

Figure 2. (a) Bathymetry of the entire domain of the Gulf–COAWST, (b) zoom-in bathymetry plot of the northern Gulf of Mexico (nGoM), and (c) locations of observed inorganic nutrient and DO profiles derived from WOD, SEAMAP, and NOAA's shelf-wide cruises. In (a), locations of river point sources are denoted by red dots. In (b), only bathymetry between 6 and 50 m is mapped with colors; computational meshes are split by solid grey lines; main river channels are denoted by solid blue curves; locations of river point sources of the Mississippi and the Atchafalaya Rivers are indicated by red dots; sampling locations for SOC and overlaying water respiration measurements by McCarthy et al. (2013) are denoted by dark yellow dots.

## 3 Biogeochemical model validations

### 3.1 Available measurements

In this section, biogeochemical model validations were conducted for surface inorganic nutrient concentration (i.e., $NO_3$, $PO_4$, and $Si(OH)_4$), types of limited nutrients, ratios of diatom/total phytoplankton, SOC, DO concentration profiles, spatial

distributions of bottom DO concentration and temporal variability of the hypoxic area against multiple field and lab data sets.
Validation of the hydrodynamic model can be found in Zang et al. (2019).

Inorganic nutrient concentrations from WOD and NOAA's shelf-wide cruises were used for model validation. WOD
measurements cover the period from 11 January 2007 to 5 July 2009, while the shelf-wide records cover the 2007-2019 period.
The types of limited nutrients across the LaTex shelf were discussed based on multiple bioassay studies (Turner and Rabalais,
2013; Quigg et al., 2011; Smith and Hitchcock, 1994; Sylvan et al., 2006, 2007; Zhao and Quigg, 2014; Nelson and Dortch,
1996). The diatom percentage of total phytoplankton was derived from measurements by Chakraborty and Lohrenz (2015) and
Schaeffer et al. (2012). The SOC measurements were provided by an incubation study (McCarthy et al., 2013). Available DO
concentration profiles were obtained from the NOAA-supported mid-summer shelf-wide cruises and Summer Groundfish
Survey in GoM supported by the Southeast Area Monitoring and Assessment Program (SEAMAP) conducted annually by the
Gulf States Marine Fisheries Commission. The shelf-wide cruises provided 1450 measured profiles with 70401 available
records from 2007 to 2019. There were at least 83 DO profiles for each summer (June–August, except 2016) from the shelf-
wide cruise observations. The selected SEAMAP DO dataset covers a time range from 2007 to 2019 with measurements
including 1651 profiles with 94200 sampled records. Locations of the selected profiles from different archives were shown in
Fig. 2c. Summer measurements by the shelf-wide cruises were used to validate spatial patterns of bottom DO concentration
and time series of summer hypoxic areas. Estimated hypoxic areas by the cruises are available from 2007 to 2020, with a range
from 5,480 $km^2$ to 22,720 $km^2$.
**3.2 Surface nutrient concentration**
One-to-one comparisons for surface nutrient concentration validation were seldom carried out in previous numerical studies,
where spatial-averaged or temporal-averaged matrices were frequently validated. To provide a more detailed quantification of
model performance in surface nutrients, we performed one-to-one differences between simulations and measurements at each
sampling location on specific dates. Modeled results showed good agreements with the cruise measurements from both shelf-
wide and WOD records (Fig. 3) in terms of magnitudes. There are 86% of surface $NO_3$ differences dropping within a range of
$\pm$ 10 mmol $m^{-3}$ with the most biases ranging from -2.5 to 0 mmol $m^{-3}$ (56%, Fig. 3a). It indicates a slight underestimation,
which is mostly found in the mid and western shelf (>150 km from the Mississippi River mouth, Fig. 3b). Surface $NO_3$ biases
exhibit a higher variance near the mouth than in other regions. There are 92% of surface $PO_4$ bias pairs dropping within $\pm$ 1
mmol $m^{-3}$ (Fig. 3c), exhibiting a more even distribution pattern than the $NO_3$ differences. It results from the model
underestimation in the mid and east shelf but overestimation in the west (Fig. 3d). There are 88% of surface $Si(OH)_4$ differences
within a range of $\pm$ 20 mmol $m^{-3}$ with a slight underestimation (Fig. 3e). We found higher biases near the Mississippi (first to
third quartiles within $\pm$ 8 mmol $m^{-3}$ at 0-150 km) and the Atchafalaya (-5 to 7 mmol $m^{-3}$ at 150-300 km) Rivers mouths (Fig.
3f) than at the western shelf. Mean Mississippi and Atchafalaya riverine $PO_4$ concentrations were 2.7 $\pm$ 0.7 mmol $m^{-3}$ and 2.3
$\pm$ 0.7 mmol $m^{-3}$, respectively, and mean riverine $Si(OH)_4$ concentrations were 118 $\pm$ 23 mmol $m^{-3}$ and 116 $\pm$ 21 mmol $m^{-3}$,

respectively. Thus, the nutrient concentration bias between simulations and observations is acceptable, considering the possible
transient influence from the riverine nutrient loads during a survey.

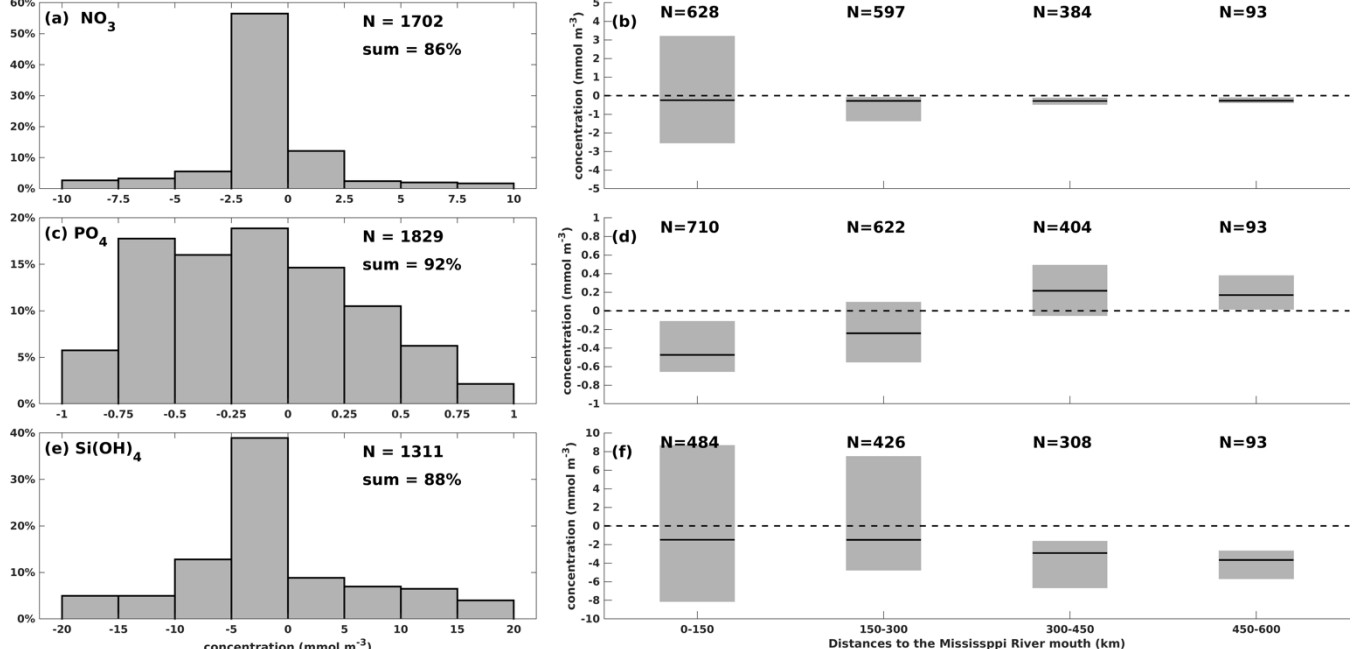

Figure 3. Comparison of surface nutrient concentration between model hindcasts and cruise measurements (both shelf-wide and
WOD) for (a)–(b) NO$_3$, (c)–(d) PO$_4$, and (e)–(f) Si(OH)$_4$. The left bar graphs illustrate the distribution of concentration differences
by percentage within specific concentration ranges, while the right box charts show the first quartiles, third quartiles, and medians
of the concentration differences against the distance to the Mississippi River mouth.

### 3.3 Nutrient limitation

Nutrient limitation could vary among different phytoplankton species with different efficiencies in nutrient uptakes. In our
model, the Si limitation was modeled only for the PL growth. Depth-averaged nutrient limitation coefficients (see Eqs. A9–
A10) over the surface 1 m were compared to bioassay studies. When a modeled coefficient is lower than 0.75, the water body
is defined to be limited by the corresponding nutrient for the corresponding phytoplankton group. A bioassay study by Turner
and Rabalais (2013) demonstrated that N limitation was more common than P limitation along transects C and F in June and
July 2010 (Fig. 4). All July samples were found to be N limited, while only some June samples along transect C were found
to be P limited with the rest to be N limited. The model mostly captured the dominated N limitation pattern along both transects.
As there was a lack of location information in this bioassay study, we could not pinpoint the location of the observed P
limitation in Fig. 4. However, our model indicated that the P limitation was more common around the Mississippi River mouth
for both phytoplankton groups. In June 2010, transect C, located at the boundary of the modeled N and P limitation, showcased
that the model could successfully capture the observed spatial pattern of nutrient limitation.

Dominated P limitation adjacent to the Mississippi River mouth was observed in other bioassay studies (e.g., Quigg et al.,
2011; Smith and Hitchcock, 1994; Sylvan et al., 2006, 2007) and was also captured by the model indicated by high percentage
occurrences over the simulation period (2007–2020) (Figs. 5b, 5e). N limitation was mostly found in the shallow parts of the
middle and western shelf during spring (Fig. 5a) and became more widespread offshore and eastward in July (Fig. 5d). This
pattern was also seen in earlier bioassay estimates (e.g., Quigg et al., 2011; Sylvan et al., 2007; Zhao and Quigg, 2014). The
Si limitation occurrence performed a distinct offshore gradient in spring (Fig. 5c). Bioassay studies have illustrated that Si
limitation occurred in the east shelf during spring (e.g., Quigg et al., 2011; Nelson and Dortch, 1996; Smith and Hitchcock,
1994). The gradient tilted westward in July, indicating a potential oligotrophic water intrusion from deep waters when the
circulation pattern changed during the summer months. However, there exists a knowledge gap regarding Si limitation over
the western shelf region, where no known bioassay studies have been conducted. We gather some clues from Dortch and
Whitledge's (1992) study of spring 1988 and summer 1987 in the Mississippi plume (mostly east of 90°W with depth >50 m),
where they found that Si had a higher potential as a limiting nutrient than N in summer at high salinity waters. Salinity in the
western shelf is usually high in July due to the changing predominant current system from westward to eastward or
northeastward. The low-saline and Si-rich plume waters can be replaced by deep waters with higher salinity and lower Si. We
expect a more Si-limited environment in the western shelf than in other parts during July, which, however, needs further
support from additional bioassay studies.

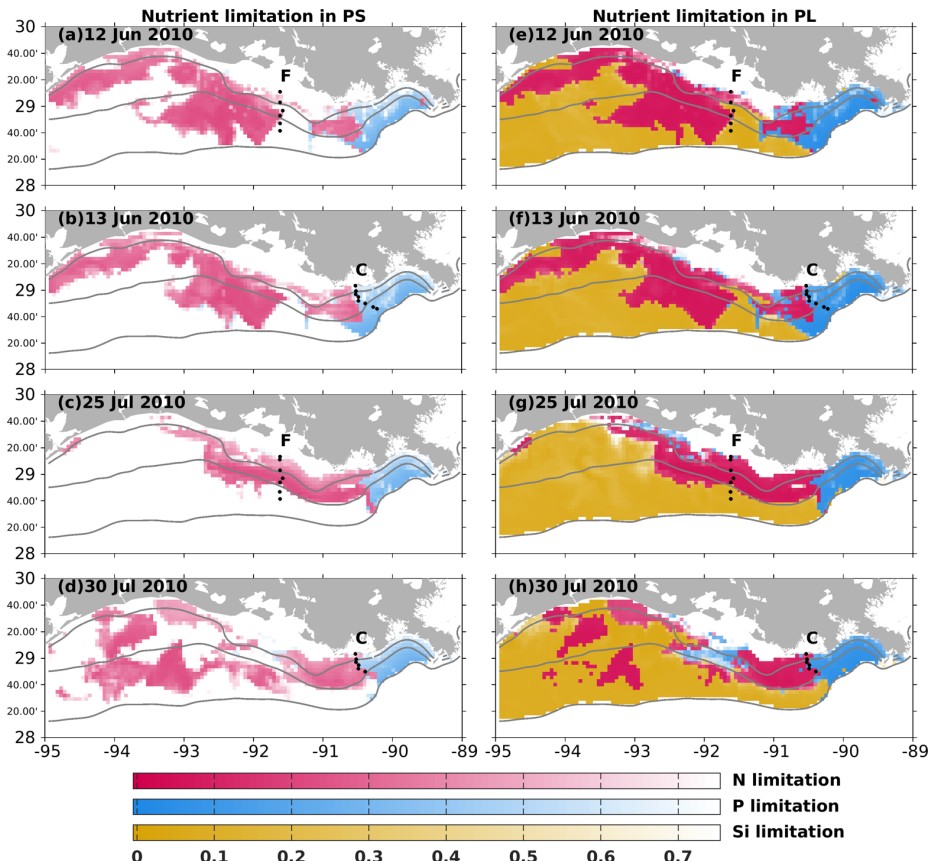


Figure 4. Comparisons of nutrient limitation patterns between model hindcast and a bioassay study (samples from 2010 mid-summer
shelf-wide cruises ) by Turner and Rabalais (2013) for June and July 2010. According to the bioassay study, in June, some samples
along transect C were limited by P, while all samples along transect F were limited by N; in July, all samples along both transects
were limited by N. Modeled nutrient limitation coefficients (for PS, left column; for PL, right column) are averaged over the surface
1 m. A lower coefficient indicates the corresponding nutrient is more limited.

346

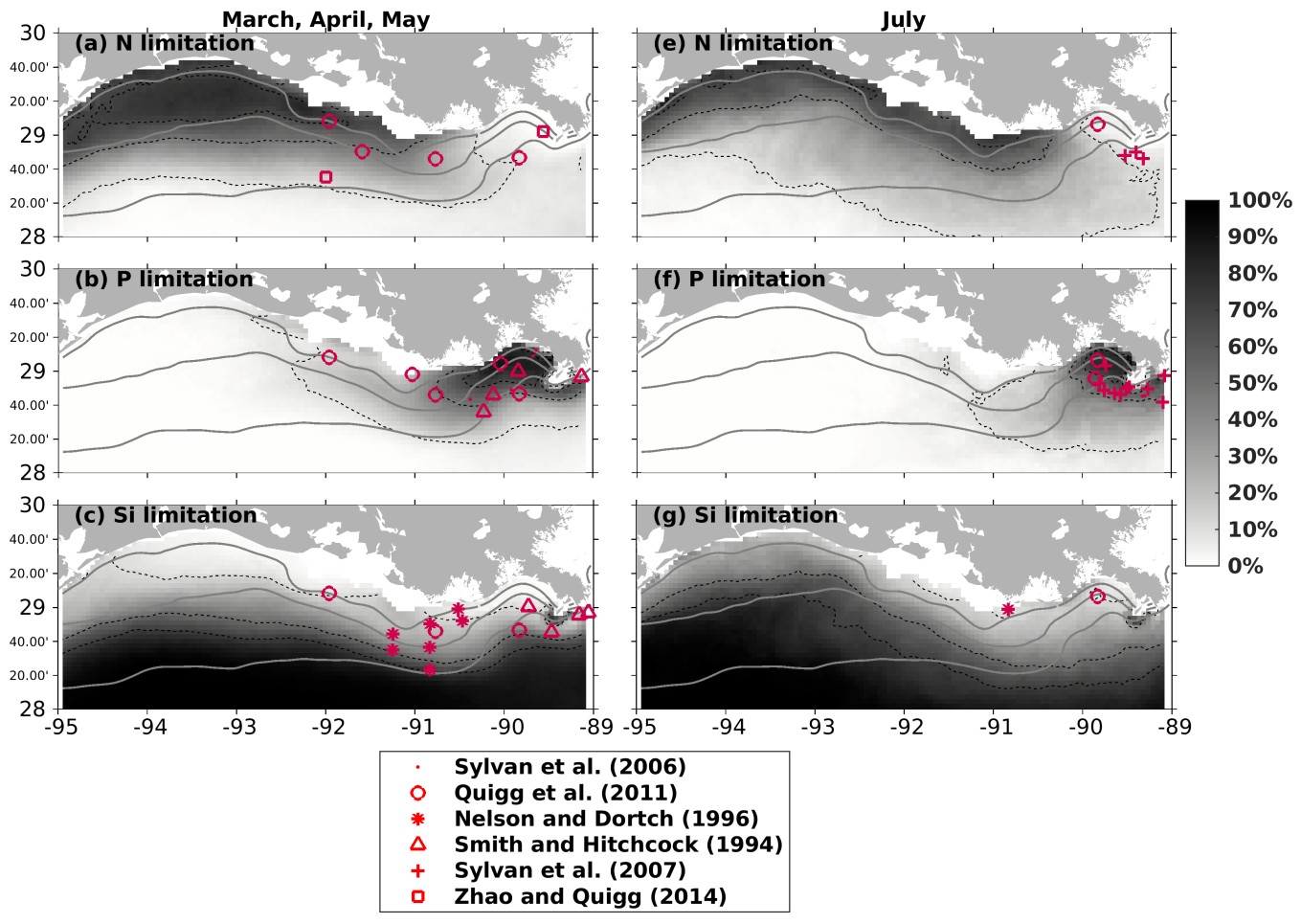

347

**Figure 5. Modeled nutrient limitation occurrences (in percentages) overlayed with locations of observed limited nutrients by bioassay studies in spring (left column) and July (right column). Modeled occurrences are obtained based on the entire simulation period (2007–2020). Solid grey lines indicate bathymetry of 10, 20, and 50 m, while black dash lines represent the contour lines of 10%, 50%, and 70%.**

## 3.4 Diatom ratios

Cruise observations confirmed that diatom is one of the dominated phytoplankton groups in the LaTex shelf (Schaeffer et al., 2012; Chakraborty and Lohrenz, 2015). When compared to the Schaeffer et al.'s (2012) measurements, vertical averages of PS and PL concentration over the surface 0.5 m at the sampled points (black dots in Fig. C2) were extracted from the model hindcast. Statistics of modeled diatom ratios were derived from the daily ratios at the selected locations over the cruise months in 2008. When compared to Chakraborty and Lohrenz's (2015) measurements, we only calculated the modeled diatom ratios at the surface, middle, and bottom layers. Statistics of modeled ratios were given based on the daily ratios at these layers over the cruise regions (polygons shown in Fig. C2) and during cruise months in 2009 and 2010. The modeled ratios reasonably reproduced the measured ones in magnitudes, monthly variability, and cross-shelf variability (Table 1). During the cruise periods in 2008, the range of modeled diatom percentage (59% to 87%) matched well with the measurements (71% to 86%)

except for May 2008, when underestimations were found. In 2009, our model results agreed well with the measurements in
inner shelf waters but overestimated the measurements in the mid-shelf regions, especially in the summer and fall of 2009.
The measured percentages exhibited salient monthly variations with higher values in winter and spring and lower ones in
summer and fall. In the cross-shelf direction, the phytoplankton community shifted from a highly diatom-dominated one in the
inner shelf waters to a less diatom-dominated one in the mid-shelf waters, especially in summer. It should be noted that a high
uncertainty was found in the diatom ratio from both hindcast and measurements (comparable standard deviation against mean
values). Therefore, model-measurement biases are expected when comparing statistics derived from a whole month (model
hindcast) and a few days (cruise measurements). Then, the biases should be acceptable as the magnitudes of modeled and
measured statistics are closed.

**Table 1. Comparison of simulated (mean ± 1SD) and measured (mean ± 1SD in parentheses) diatom percentage of the total**
**phytoplankton. Note that the statistics for the simulated percentages were conducted based on concentration values over the cruise**
**months and over regions that cover the cruise sampling locations (Fig. C2). The measured percentages by Schaeffer et al. (2012) (for**
**measurements in 2008) were calculated based on biovolume values, while those by Chakraborty and Lohrenz (2015) (for**
**measurements in 2009 and 2010) were given by chlorophyll *a* attributed to different phytoplankton groups.**

|  | Diatom/total phytoplankton × 100% | |
|---|---|---|
|  | Inner shelf | Mid shelf |
| February 2008 | 68±30 (71±47) | |
| April 2008 | 71±39 (71±17) | |
| May 2008 | 59±45 (80±24) | |
| June 2008 | 87±22(86±10) | |
| January 2009 | 46±36 (66±21) | 48±13 (47±14) |
| April 2009 | 46±37 (59±14) | 46±17 (33±29) |
| July 2009 | 63±31 (40±13) | 44±26 (13±16) |
| October–November 2009 | 53±35 (46±14) | 41±18(19±17) |
| March 2010 | 47±39 (50±14) | 50±24 (64±12) |


### 3.5 SOC rates

Modeled SOC rates were compared against a laboratory incubation study by McCarthy et al. (2013) at five shelf sites (location
see the Fig. 1 in that paper) using sediment and water samples collected during six cruises (i.e., July 2008, September 2008,
January 2009, August 2009, May 2010, and May 2011). The modeled SOC was averaged over the cruise months for four shelf
sites (i.e., F5, C6, B7, and MRM; Fig. 2b). Our model could well capture the SOC magnitude. The model generally

overestimated the SOC at all sites except for May 2010 at site C6, and August 2009 at sites MRM (Fig. 6). The largest overestimations were found in September 2008 when measurements were carried out shortly after Hurricanes Gustav and Ike. These measurements tended to provide a low SOC but a high water-column respiration, possibly induced by the mixing incurred by storms. Note that the model results shown in Fig. 6 were averaged over an entire month because no exact cruise date information was reported in McCarthy et al. (2013).

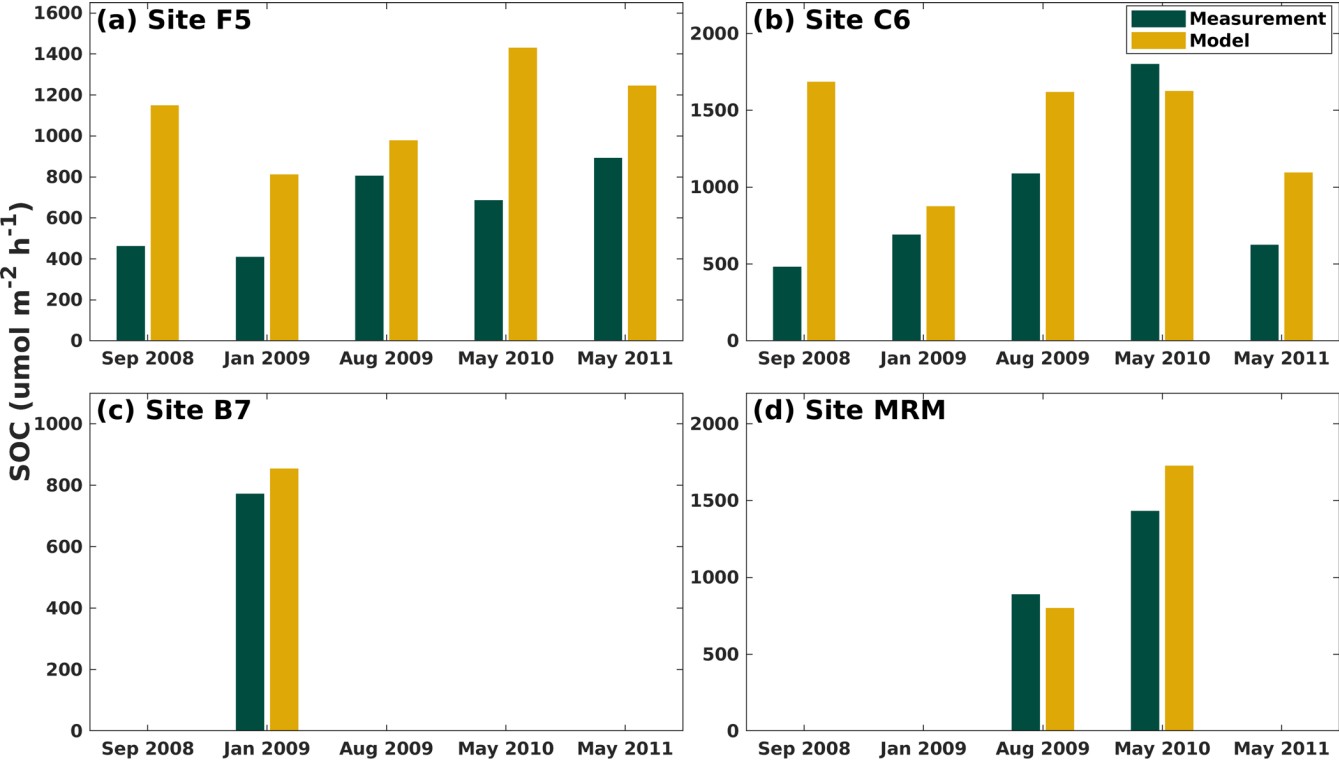

**Figure 6. Comparison of modeled and measured SOC (unit: µmol m$^{-2}$ h$^{-1}$) at four LaTex shelf sites (dark yellow dots in Fig. 2b). Note that the measurements are provided by McCarthy et al.'s (2013) incubation study and the modeled SOC for each sampled site is averaged over the specific months.**

### 3.6 DO profiles

Both the shelf-wide and SEAMAP cruise studies provide high-resolution measurements of DO profiles in the vertical direction, with the observed layers ranging from surface to bottom. The number of observed layers is close to or even more than that of the modeled layers. Therefore, the observed DO profiles were interpolated to the modeled layers using the nearest interpolation method for the one-to-one comparisons between modeled and observed DO profiles. Mean, median, and 25-75 percentile ranges of the model-observation differences were derived and compared against normalized depths ranging from -1 (bottom) to 0 (surface) (Fig. 7). Most of the biases were within ± 1 mg L$^{-1}$, indicating a robust model performance in reproducing DO profiles. We noticed the model tended to overestimate the shelf-wide observed DO by more than 1 mg L$^{-1}$ but less than 2 mg

L⁻¹ on average over the upper layers in shallow waters (Fig. 7a). When validating against the SEAMAP profiles, a wider range
of biases were also found at near-surface layers of the shallower water (Fig. 7d) than in deeper waters (Figs. 7e and 7f). On
the one hand, in shallow water, cruise measurements seldom resolved the vertical layers finer than the model where 36 layers
were designed, which introduced biases when interpolating the measured profiles to the modeled layers. On the other hand,
ROMS tends to overmix the water column in shallow water regardless of the vertical mixing parameterizations chosen
(Robertson and Hartlipp, 2017). Despite the slight overestimations of DO profiles, our model results performed better than
those of previous numerical studies. For example, DO concentration biases against profile measurements in Yu et al. (2015)
were mostly within 2 mg L⁻¹.

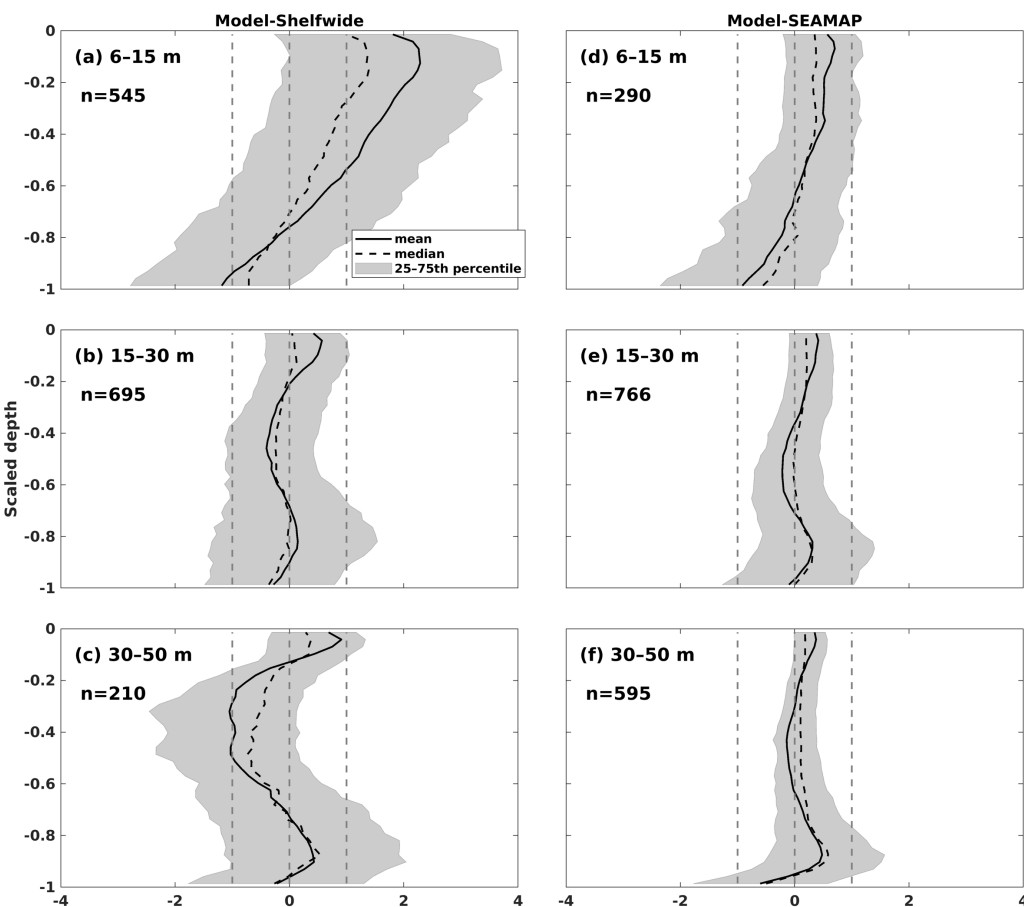


**Figure 7. Concentration difference statistics of DO profiles between model hindcasts and measurements by (a–c) NOAA's shelf-wide**
**cruises and (d–f) SEAMAP. The statistics are derived from one-to-one differences between hindcasts and measurements at specific**
**sampling locations and dates. The normalized depths of 0 and -1 represent the surface and bottom, respectively. The total counts (n)**
**of profiles within different depth ranges are shown in each panel.**

### 3.7 Spatial distributions of bottom DO and temporal variability of hypoxic area

As the annual NOAA shelf-wide cruises were conducted from the east shelf to the west in the summer, the model simulated bottom DO was resampled following the cruise periods. For example, if the westmost location of the cruise is 90°W on day 1, the simulated bottom DO concentration over the east of 90°W on that day is extracted. On the following day, if the westmost location of the cruise is 91°W, the simulation between 91°W and 90°W on day 2 is extracted, and so forth. All the extracted frames were blended to reconstruct the spatial distribution of simulated bottom DO concentration during the summer cruise period. Simulated results outside the LaTex shelf and over the deep (> 50 m) and shallow (< 6 m) water regions were excluded since observations were unavailable. Model results showed a good agreement with the observations in terms of interannual variability and spatial extent of bottom hypoxic waters (Fig. 8). The spatial distribution of the hypoxic regions varied over different summers. For example, the hypoxic area was small and was primarily restricted to nearshore (< 20 m) regions during the summers of 2007, 2009, 2012, 2014, and 2018. The size of the hypoxic zone was more prominent and extended offshore in 2008, 2011, 2013, and 2019. The spatial dispersion of hypoxic waters occurred mostly over the west of the LaTex shelf, where bathymetry gradients were gentle. Over the eastern shelf, the hypoxic water was mostly constrained within a narrow belt. These results suggested that the hypoxia development on the LaTex shelf was complex and generally followed the bathymetry and distances from the major river mouths.

The daily time series of the size of the hypoxic zone was calculated over the LaTex shelf (6–50 m; Fig. 9). There was a good agreement between simulated hypoxia zone size and that captured by the shelf-wide cruises in terms of variability and magnitude. The overall correlation coefficient (CC) was 0.69 over the 99% significant level (Table 2). The 10-year running CCs ranged from 0.66 to 0.76, surpassing at least the 95% significance threshold. Underestimations were found in 2007, 2008, and 2017 with a root-mean-squared error (RMSE) of 1693 $km^2$, while overestimates in other summers of interest with a RMSE=8084 $km^2$. The model performed apparent overestimation for 2019 summer. Nevertheless, biases in other summers were acceptable, considering the relative sporadic converges of cruise data.

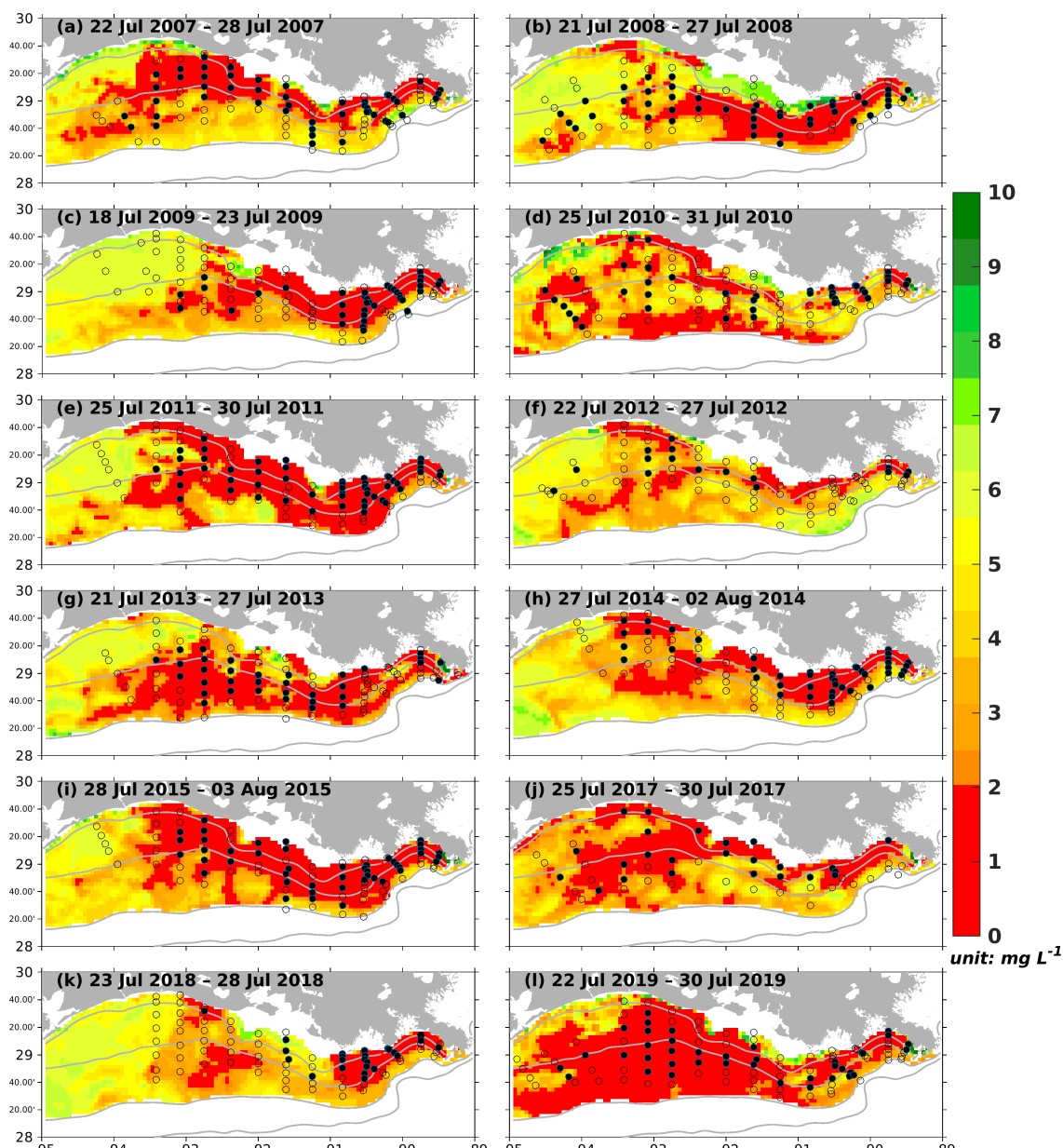


Figure 8. Modeled summer bottom DO concentration (colored patches) and NOAA's summer shelf-wide hypoxia observations (black dots and open circles). The black dots and the open circles are indicators of observed bottom hypoxia and normoxia, respectively. The solid grey lines indicate bathymetry of 10, 20, 50, and 100 m.

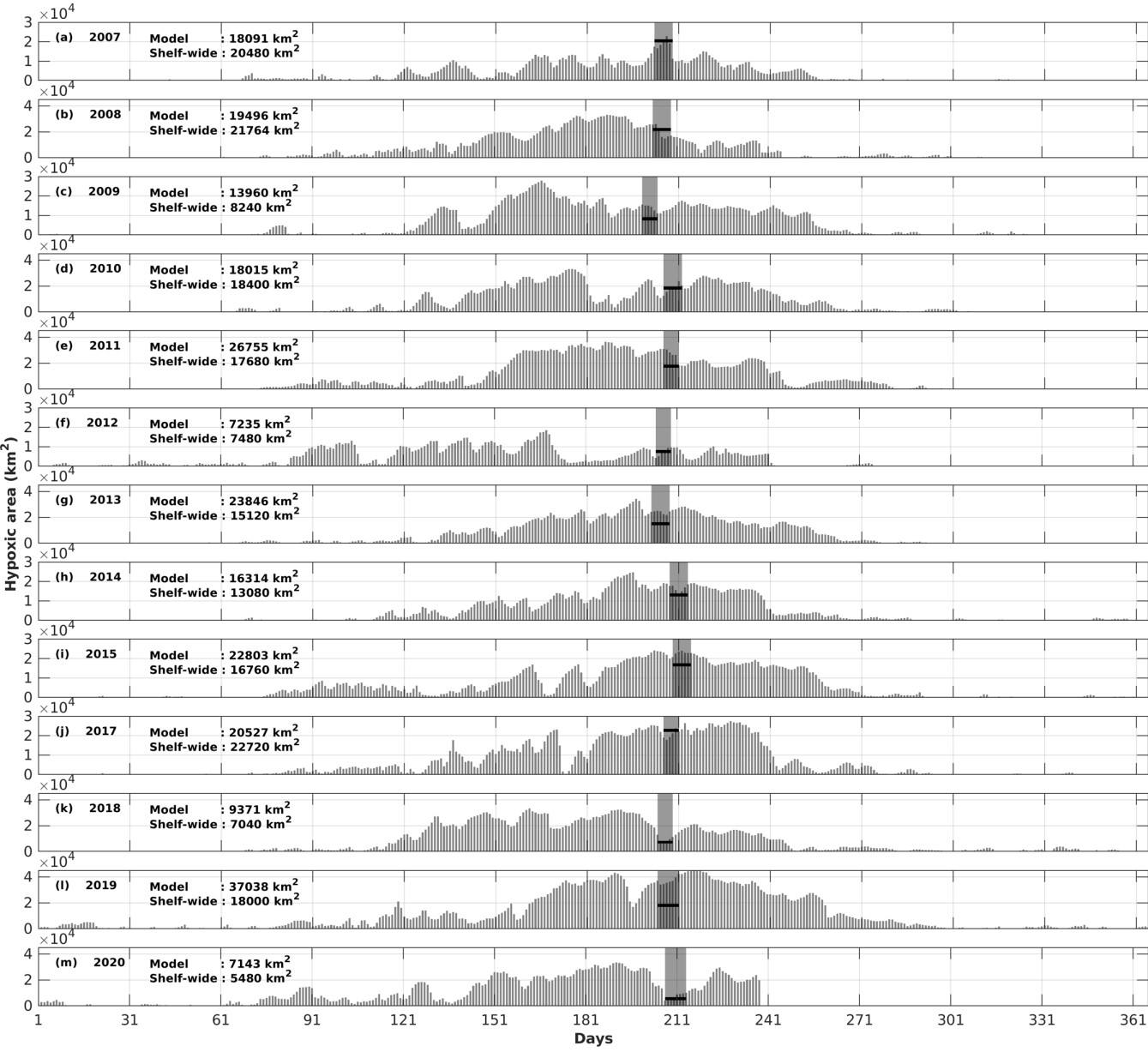

Figure 9. Comparison of the hypoxic area (in km²) between model simulations and shelf-wide cruise observations from 2007
to 2020 (except 2016). The grey patches denote the cruises periods while the solid black lines represent the measured hypoxic
area.

**Table 2. The overall (2007–2020) and 10-year running correlation coefficients (CCs) of summer hypoxic area between model simulations and shelf-wide measurements. Note that the comparison in 2016 is excluded due to the lack of measurement. Superscripts \* and \*\* indicate the corresponding CCs are above the 95% and 99% significant levels, respectively.**

| Year ranges | CC |
|---|---|
| 2007–2020 (overall) | 0.69** |
| 2007–2017 | 0.66* |
| 2008–2018 | 0.76** |
| 2009–2019 | 0.71* |
| 2010–2020 | 0.76** |

## 4 Results and Discussion

### 4.1 Nutrient limitation

In this study, the riverine nutrient loads from the Mississippi and Atchafalaya Rivers were calculated based on measurements from the USGS NWIS. During the investigated period (2007–2020), the riverine N:P ratio was higher than 16:1 during spring and reached its minimum in mid-summer to early fall (Fig. 10a). It indicated that P limitation in the shelf could be more severe in spring than in mid-summer and early fall (also seeing Fig. 5). Most riverine N:Si ratios fluctuated between 0.5 and 1 and were slightly higher in late spring and summer than in other seasons (Fig. 10b). The riverine N and Si loads were at a similar level when compared to the Redfield ratio of N:Si=1:1. However, recent studies have pointed out that marine diatoms require a lower N:P:Si ratio (16:1:20, Billen and Garnier, 2007; Royer, 2020), indicating that N may be more excessive over Si than previous thought. Riverine Si:P ratios were much higher than 16:1 and 20:1, suggesting that the major river systems transported excessive Si over P to the LaTex shelf. From the perspective of riverine supply, the plume's extent appeared to be more constrained by P availability (see Figs. 4–5) than by N and Si. The limitation effects of N and Si might be relatively similar, given that the N:Si ratio was around 16:20. However, the nutrient limitation is also related to the phytoplankton assimilation efficiency on nutrients (half-saturation coefficients for nutrient uptakes) and the water exchanges between the shelf and the adjacent waters.

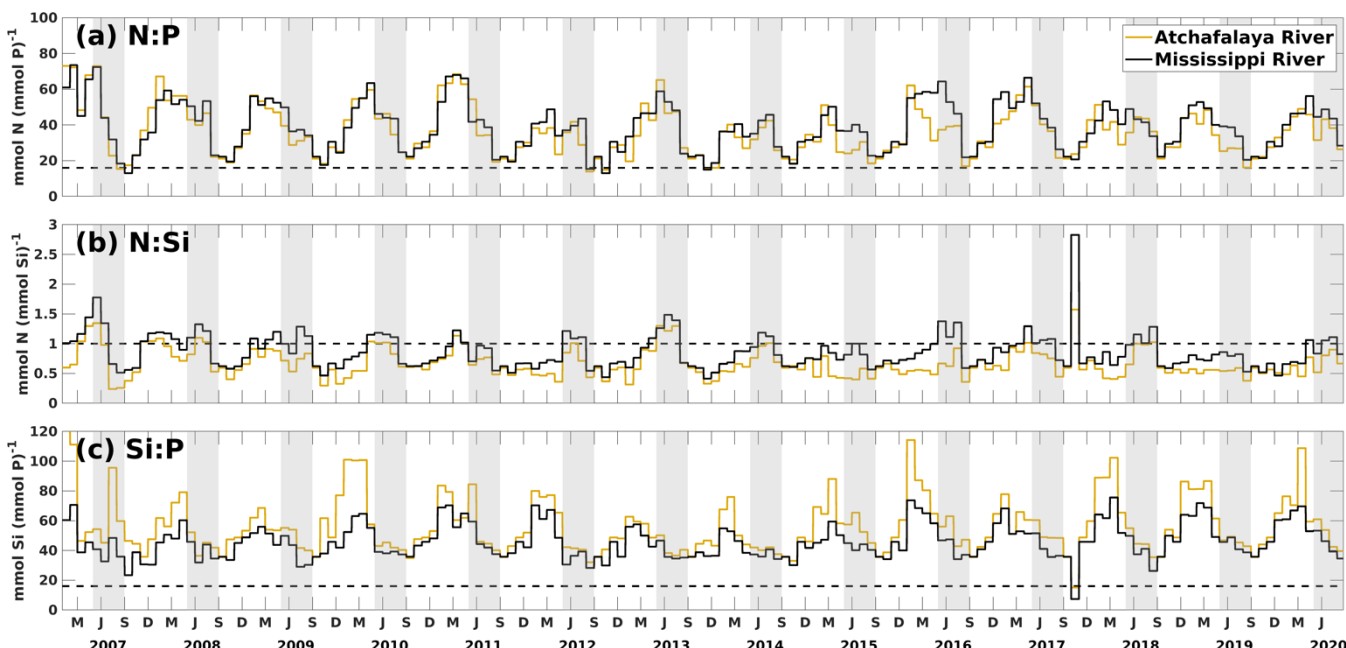

**Figure 10. Daily time series of ratios of nutrient loads from the Mississippi and Atchafalaya Rivers. The black dashed lines denote the nutrient ratios of 16:1, 1:1, and 16:1 in (a), (b), and (c), respectively. The gray patches indicate the late spring and summer (May– August) each year. The capitalized letters M, J, S, and D in the x-axis denote the first day of March, June, September, and December, respectively.**

The half-saturation coefficient for phytoplankton nutrient uptake is a critical factor associated with nutrient limitation. In our model, PL was parameterized to be more competitive than PS in nutritious waters with a higher half-saturation coefficient. The half-saturation coefficients for $NO_3$ and $NH_4$ used in this model study (Table B4) followed the parameterization in Shropshire et al. (2020). The half-saturation coefficients for $PO_4$ were designed as 0.03125 mmol P m$^{-3}$ for the PS and 0.1875 mmol P m$^{-3}$ for the PL, according to the Redfield stoichiometry of N:P=16:1. This parametrization method was also applied in Laurent et al. (2012) for discussion of P limitation effects in the LaTex shelf. The half-saturation coefficient for $Si(OH)_4$ ($K_{SiOH_4}$) was designed to be 6.0 mmol Si m$^{-3}$, mirroring the choice in Shropshire et al. (2020), although there was no discussion on how this parameter was determined. Uptake kinetic studies for different marine diatom species have suggested a wide range of $K_{SiOH_4}$ from 0.8 to 17.4 mmol Si m$^{-3}$ (Table 6). The average, median, first, and third quartile of the measured coefficients in Table 6 were 5.9, 4.5, 2.3, and 7.0 mmol Si m$^{-3}$, respectively. We opted for the average over the median coefficient in our model, considering the PL group as a representative marine diatom assemblage. However, the $K_{SiOH_4}$ for a diatom assemblage may shift given changing ambient silicate concentration. For example, as pointed out by Nelson and Dortch (1996), $K_{SiOH_4}$ for the sampled phytoplankton assemblage (dominated by diatom species) remained low from 0.48 to 1.71 mmol Si m$^{-3}$ when the ambient silicate concentration was low between 0.13 to 0.41 mmol Si m$^{-3}$, but increased to 5.29 mmol Si m$^{-3}$ as ambient silicate concentration was 4.72 mmol Si m$^{-3}$. Along Mississippi and Atchafalaya River plumes, which deliver silicate-rich waters to

the shelf (average concentrations are 118 ± 23 mmol m$^{-3}$ and 116 ± 21 mmol m$^{-3}$, respectively), the silicate concentration remains high, suggesting a high half-saturation coefficient. We acknowledged that a constant half-saturation coefficient cannot fully capture the dynamics of silicate and diatom outside the plumes, as indicated by Nelson and Dortch (1996). Further investigations and improvements in model parameterization for the dependency of $K_{SiOH_4}$ on silicate concentration are needed in future studies.

**Table 6. Half-saturation coefficient (unit: mmol Si m$^{-3}$) for silicate uptake by different diatom species according to multiple uptake kinetic studies.**

| Diatom species | $K_{SiOH_4}$ | Reference |
|---|---|---|
| *Cylindrotheca fusiformis* | 0.85 | Del Amo and Brzezinski (1999) |
| *Nitzschia alba* | 6.8 | Azam (1974) |
| *Nitzschia alba* | 4.5 | Azam et al. (1974) |
| *Phaeodactylum tricornutum* | 4.0, 9.2, 6.3 | Del Amo and Brzezinski (1999) |
| *Thalassiosira nordenskioeldii* | 2.8 | Kristiansen and Hoell, (2002) |
| *Thalassiosira pseudonana* | 7.04 | Thamatrakoln and Hildebrand (2008) |
| *Thalassiosira pseudonana* | 1.4 | Del Amo and Brzezinski (1999) |
| *Thalassiosira pseudonana* | 0.8, 2.3 | Nelson et al. (1976) |
| *Thalassiosira weissflogii* | 15.2, 17.4 | Milligan et al. (2004) |
| *Thalassiosira weissflogii* | 4.5 | Del Amo and Brzezinski (1999) |
| Average | 5.9 | |
| Diatom functional group (PL) | 6.0 | This study |

The changing coastal wind and current systems during summer can lead to significant changes in nutrient distribution, alternating the growth of phytoplankton and summer hypoxia development. Here, we show three snapshots in August 2019 (Fig. 11) when seasonal hypoxia reached its maximum (Fig. 9) to demonstrate the highly varying shelf hydrodynamics and the resultant nutrient dispersion patterns. During spring, the westward alongshore current system dominated the LaTex shelf, while in summer, currents shifted eastward and southward, forming a clockwise circulation in the middle and western shelf (Fig. 11a). This shift not only pushed the river plume eastward but allowed water intrusion from the west and deep gulf. Waters from the outer shelf were typically high in salinity and low in nutrient content with higher N:Si and lower Si:P ratios than local waters (Fig. 11c–11e). Although silicate concentration remained high and was usually excessive in the plume area, the intrusion of deep gulf waters led to an enlarging Si limitation domain in the west LaTex shelf (Fig. 11f–11g). The PL concentration and primary production (PS+PL) (Fig. 11h–11j) in the western shelf decreased pronouncedly after the intrusion of Si-limited waters. Pronounced declines in PON$_{sed}$ concentration (Fig. 11k) in the shallow western shelf were also detected five days after

the primary production decreased. The SOC was expected to decrease, which could relieve the summer bottom hypoxia in the
shallow western shelf.

We also noted that the upwelling system along the nearshore far western shelf (> 95°W) and the direct transport of PON from
the west could affect the evolution of bottom hypoxia on the LaTex shelf. In the northern hemisphere, the clockwise circulation
system was favorable for the development of coastal upwelling systems, which induced cooling at the surface along the coast
(Fig. 11b), and led to elevated concentrations of surface inorganic nitrogen, phosphate, and silicate along the nearshore western
shelf. Total surface primary production remained high roughly along the 20 m isobath, where the water column PON
concentration was also elevated. The clockwise circulation system carried the PON offshore and northeastward to the LaTex
shelf, inducing an increase in the $PON_{sed}$ pool (around 28°N; Fig. 11k) and SOC. The high alongshore production was limited
by N rather than Si or P. However, the N limitation band narrowed around the coastal upwelling zones. Such patterns—
including low-Si water intrusion, eastward transport of PON, and a narrow N limitation band in the upwelling zone—were
also found in other summer snapshots when the current system changed (e.g., Fig. C3).

Previous bioassay studies suggested the potential Si limitation on the LaTex shelf (Quigg et al., 2011; Nelson and Dortch,
1996; Smith and Hitchcock, 1994; Lohrenz et al., 1999). However, N and P limitations were reported more frequently than Si
limitations along the shelf. Part of the reason was that samples collected in previous studies were mainly from the eastern shelf,
where N and P typically appeared to be limited. Our understanding of potential nutrient limitations, particularly in the western
shelf during the recent decade, still needs to be completed. Nevertheless, this lack of *in situ* data should not hinder model
developments, as indirect evidence supports the potential Si limitation in the western shelf, especially during the summer. For
instance, a recent study using *in situ* incubations and laboratory experiments showed that the oligotrophic open gulf, generally
low in N, could also be Si-limited, as indicated by lower maximum growth rates of diatoms compared to other culture and
field measurements (Yingling et al., 2022). Additionally, earlier concentration measurements (Dortch and Whitledge, 1992)
showed that Si limitation sometimes overwhelmed the N limitation in the deep gulf waters (depth > 50 m). Water exchanges
between the LaTex shelf and adjacent deep waters become more pronounced in summer with changes in wind and current
systems. The intrusion of low-Si waters can promote the development and expansion of Si limitation, which in turn affects the
phytoplankton community and oxygen dynamics. Therefore, the accuracy of the boundary conditions along the LaTex shelf is
crucial in biogeochemical modeling. Indeed, earlier numerical studies (e.g., Fennel et al. 2013) emphasized the significance of
the correct physical boundary conditions for hypoxia modeling. Our results further illustrate that biogeochemical boundary
conditions, such as nutrient concentrations, are as critical as river forcings in influencing the shelf's nutrient distribution,
plankton, and oxygen dynamics. These effects have yet to be addressed in previous numerical studies of the LaTex shelf.

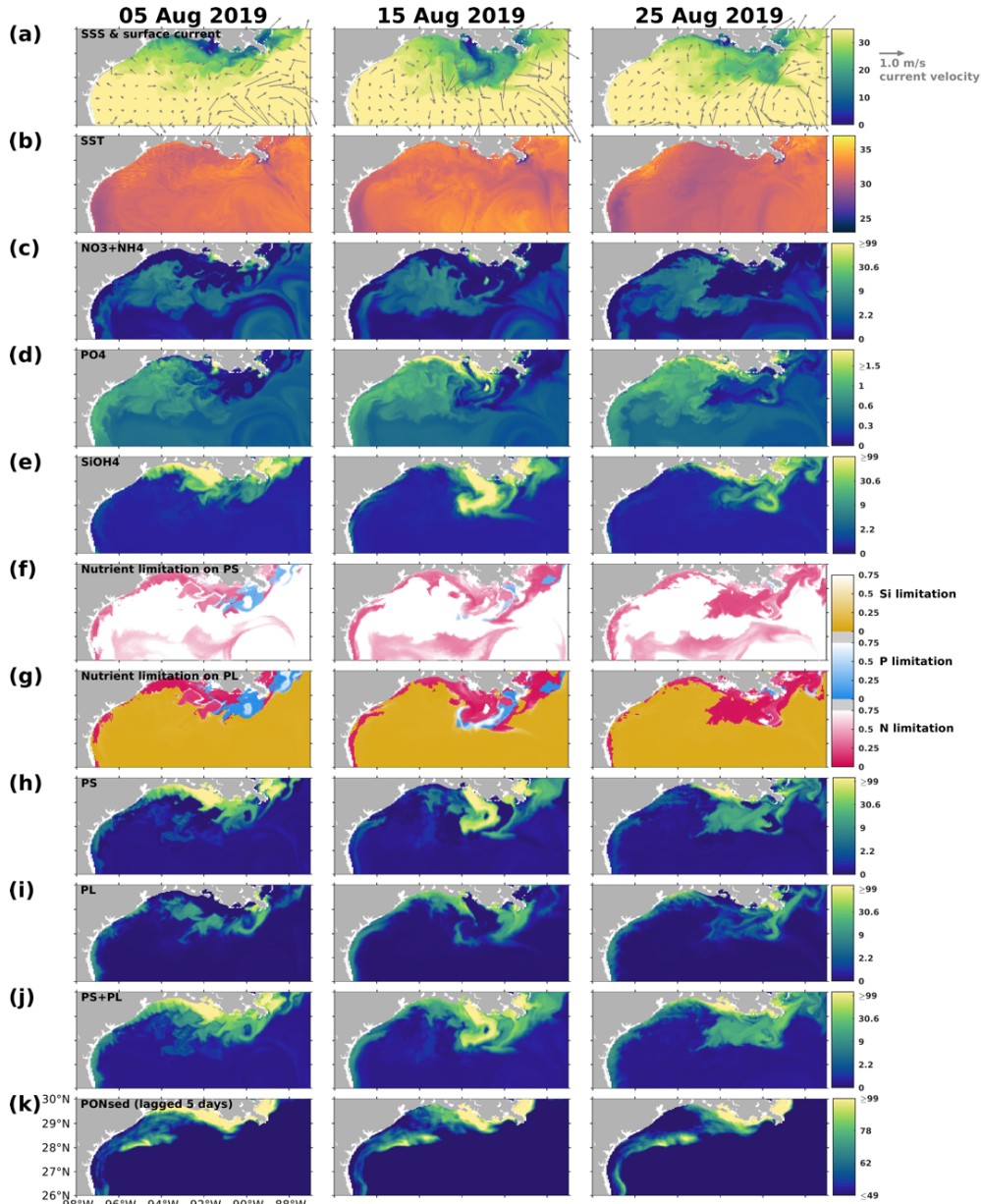

**Figure 11. Summer snapshots of (a) sea surface salinity (overlayed with surface current velocity), (b) surface temperature (°C), (c) surface total inorganic nitrogen concentration (mmol N m⁻³), (d) surface phosphate concentration (mmol P m⁻³), (e) surface silicate concentration (mmol Si m⁻³), (f–g) surface nutrient limitation coefficients, (h–i) surface phytoplankton concentration (mmol N m⁻³), and (k) PON$_{sed}$ concentration (mmol N m⁻³) with a 5-day lag in the nGoM. The nutrient, phytoplankton, and PON$_{sed}$ concentrations are displayed in the log10 scale.**

## 4.2 Plankton community interactions

On the LaTex shelf (Fig. 2b colored area), total production, primarily supported by the primary production (Fig. 12a), exhibited a bi-peak pattern in spring and summer (e.g., 2007, 2009, 2010, 2014, 2015, 2016, 2017, 2019, and 2020) with both peaks being of similar magnitude. This pattern was hardly captured by numerical models featuring a less complex plankton community (e.g., Fennel et al., 2011) and was seldom reported or discussed even in model simulations where this pattern appeared (see comparisons of modeled and satellite-derived chlorophyll *a* concentration in Gomez et al., 2018). Satellite-derived chlorophyll *a* concentration from multiple products, averaged over the LaTex shelf, also showed a bi-peak pattern from March to August (Fig. 12a), closely resembling the pattern observed in our hindcast primary production. A cruise study conducted in March, May, and July 2004 similarly depicted a higher chlorophyll *a* peak in May and a lower one in July (Quigg et al., 2011). The bi-peak pattern shown was attributed to the negative correlation between PS and PL time series, where a decrease in PS typically coincided with an increase in PL, and vice versa (Fig. 12b). For example, the peaks in primary production and chlorophyll *a*, observed from March to May 2019, coincided with the transition from a PS peak to a PL peak. The secondary peak, observed from June to July 2019, was attributed to sustained high PS biomass.

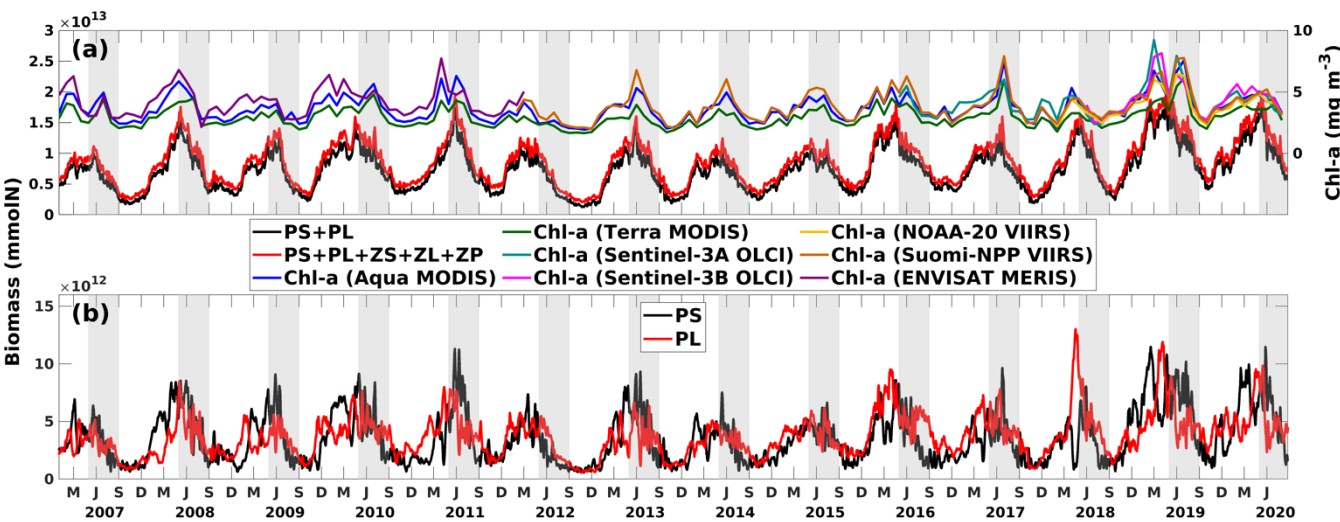

**Figure 12. Daily time series of (a) PS+PL and PS+PL+ZS+ZL+ZP biomass (represented by mmol N) and (b) PS, PL separated, integrated over the LaTex shelf (Fig. 2b colored area) and (a) monthly time series of regionally averaged (over the LaTex shelf) chlorophyll *a* concentration (in mg m⁻³) derived from multiple satellite products. The gray patches indicate the late spring and summer (May–August) period of each year. The capitalized letters M, J, S, and D in the x-axis denote the first day of March, June, September, and December, respectively.**

Competition for nutrients between PS and PL (bottom-up) and grazing pressure from zooplankton (top-down) jointly contribute to the differing fluctuation patterns of PS and PL and the bi-peak total primary production pattern. However, their effects are mostly non-linear and are not straightforward to explain. We sampled six snapshots around the primary production

peaks in the spring (early April) and summer (mid-June) of 2019 to illustrate the responses of both phytoplankton groups to
the changing nutrient environments and grazing pressure. Analysis was based on depth averages within the surface 1 m (Figs.
13–14).

In April 2019, a consistent westward current system dominated in the LaTex shelf, corresponding to an east-west elongated
river plume region, as indicated by the low sea surface salinity band (Fig. 13a). The spatial pattern of total primary production
(PS+PL) followed the plume, within which the PS concentration increased, and PL concentration decreased westward (Fig.
13e–13g). These patterns were associated with the nutrient distribution on the shelf (Fig. 13b–13d). Inorganic nutrients were
abundant around the riverine outlets and diluted and consumed westward following the currents. PL, having a greater half-
saturation constant for nutrients than PS, typically achieved higher growth efficiency or reached the maximum growth rate
more easily than PS when background nutrients were abundant. By contrast, PS could outcompete the PL when nutrient
supplies were low. In addition, a downwelling system was established along the shallow coast in the mid and western shelf,
leading to decreased nutrient concentrations and allowing PS to outcompete PL. The grazing pressure from the zooplankton
group appeared to be minor and did not significantly affect the distribution of PS and PL during these days (Fig. 13h–13j).

Pronounced bottom-up and top-down effects on the primary production were found around the biomass peak in June 2019,
coinciding with a shift in the coastal current system to a northward direction (Fig. 14a). The northward currents not only
constrained the river plume but also introduced oligotrophic deep water, as evidenced by the high surface salinity, to the inner
shelf. Note that the discharges of the Mississippi and Atchafalaya Rivers remained high from May to July 2019 (Fig. C1). A
distinct difference in the patterns of PS and PL was observed between 89 and 93°W and between 93 and 97°W (Fig. 14f–14g).
In the former region, where constrained river plumes and oligotrophic water intrusions were detected, PS exhibited a higher
nutrient uptake efficiency than PL. In contrast, PL concentration was slightly higher than PS concentration in the latter regions,
where the plume was pushed offshore. However, two areas of low PS concentration and corresponding high PL concentration
were identified between 93 and 96°W, nearshore stretching from southwest to northeast, and between 91 and 92°W, stretching
from nearshore to offshore. In these regions, the concentration of ZS, which grazes on PS only (Fig. 14h), was high, exerting
strong grazing pressure on PS but inversely allowing PL to bloom (Fig. 14h).

The results indicated that the responses in PS, PL, and PS+PL to the riverine nutrient loads were nonlinear due to the mixing
among the waters on the shelf, from the river, and intruding from the deep ocean. The riverine nutrient supplies were much
greater in June 2019 than in March–April 2019 (Fig. C1). A higher primary production and PL concentration in June would
have been expected if a nutrient-based linear relationship had been applied. However, as shown in the model and the satellite
products, primary production was higher in April than in June. This indicated that variations of phytoplankton concentration
are not only affected by riverine nutrient inputs but also the current system, which limits the expansion of river plumes,
pronounced upwelling or downwelling, and water exchanges with the oligotrophic open ocean. In the April and June 2019
snapshots, mesoscale eddies were found south of the Mississippi River outlets. The intensity and impact area of the June eddy
was greater than that of the April eddy, causing a more pronounced northward flow and more constrained river plumes along
the shelf in June. These eddy systems are known as Loop Current Eddy (LCE) systems, which can prorogate eastward and
interact with the LaTex shelf waters after the detachment from the GoM Loop Current (LC). A recent study indicated that LCE
has distinct bio-optical properties (e.g., temperature, salinity, density, DO concentration, and chlorophyll *a* concentration) from
the surrounding waters, highlighting the importance of open ocean dynamics to the shelf biogeochemical processes (Zhang et
al., 2023). Another recent study analyzing water samples from the LaTex shelf emphasized the significant impact of mesoscale
circulation features on the summer planktonic community composition (Anglès et al., 2019). This study revealed that between
20 and 25 June 2013, diatoms proliferated on the western shelf, where upwelling was detected, whereas the flagellate group
dominated within the river plumes. From 18 to 23 June 2014, diatom and flagellate bloomed in proximity to the Mississippi
River and Atchafalaya River outlets, respectively. In contrast, blooms on the western shelf were characterized by a mixture of
the two phytoplankton groups. Similar patterns were observed in our model results, as depicted in Fig. C4–C5.

In addition to the impacts of upwelling and LCE systems, direct advection of river outflow waters by coastal currents was also
found to be significant for phytoplankton community composition, carbon export, and the associated bottom DO conditions
based on other field studies in the nGoM (Chakraborty and Lohrenz, 2015) and northeastern GoM (Qian et al., 2003). Our
results suggested that the grazing pressure exerted by zooplankton groups can be variable, manifesting as significant in some
instances while remaining minimal in others. Laboratory experiments on surface water samples collected around the
Mississippi River outlets in May 1993 suggested significant grazing pressures by microzooplankton on the phytoplankton
growth (Strom and Strom, 1996). However, no salient grazer impact was found on phytoplankton growth according to bioassay
studies on the water samples collected around the plumes in April and August 2012 (Zhao and Quigg, 2014). Besides, other
unmodeled factors can also affect shelf primary production. For example, a reduction of chlorophyll *a* between 2011 and 2014
detected in the nGoM was attributed to the Deepwater Horizon oil spill disaster in 2010 (Li et al., 2019). Incorporating a
complex community into the model to address the nonlinear interactions among different plankton groups enhances our
understanding of the primary production variability and associated DO dynamics on the LaTex shelf (e.g., the bi-peak patterns
that were seldom discussed before).

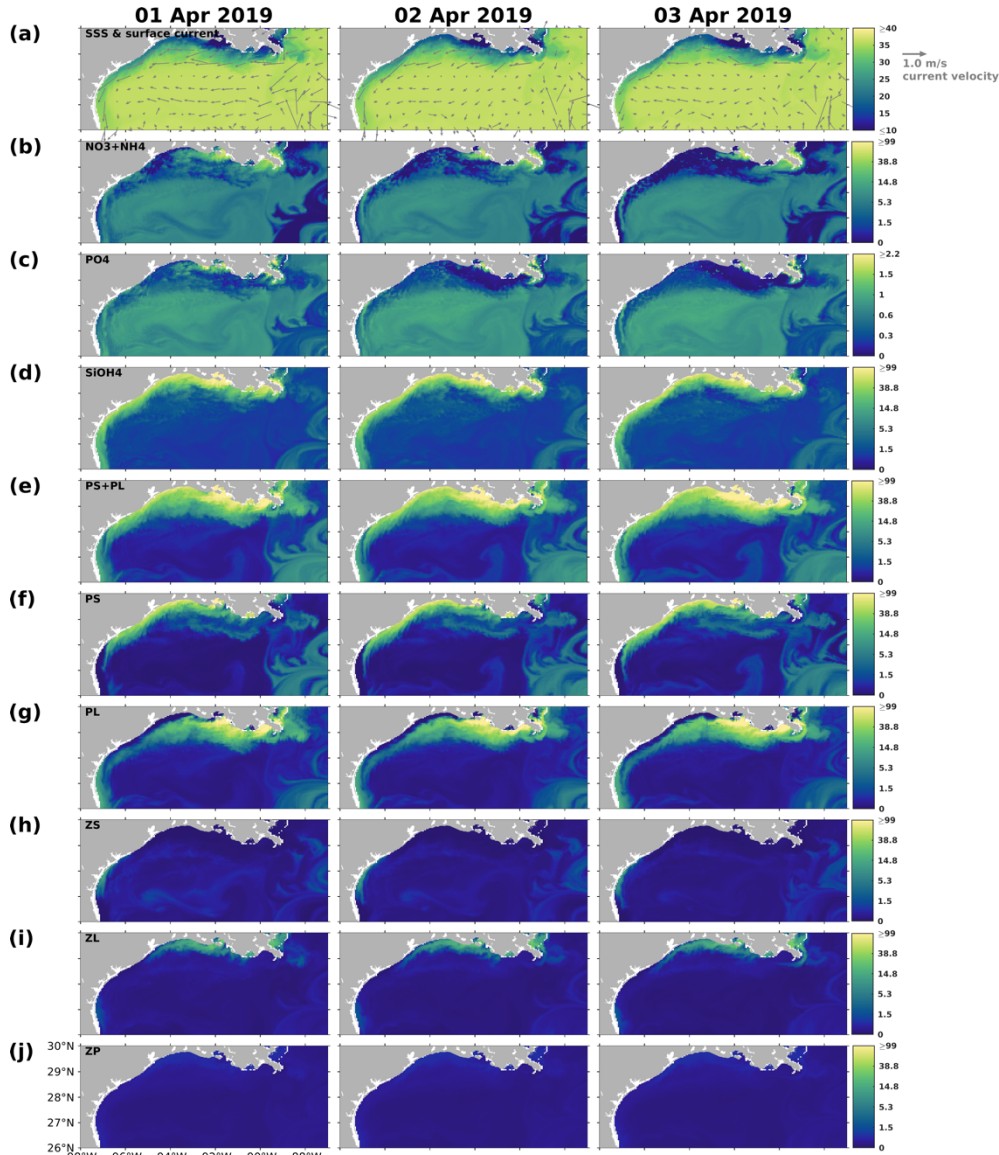

**Figure 13.** Snapshots of (a) sea surface salinity (overlayed with surface current velocity), (b) surface total inorganic nitrogen concentration (mmol N m$^{-3}$), (c) surface phosphate concentration (mmol P m$^{-3}$), (d) surface silicate concentration (mmol Si m$^{-3}$), (e–g) surface phytoplankton concentration (mmol N m$^{-3}$), and (h–j) surface zooplankton concentration (mmol N m$^{-3}$). The nutrient and plankton concentrations are displayed in the log10 scale.

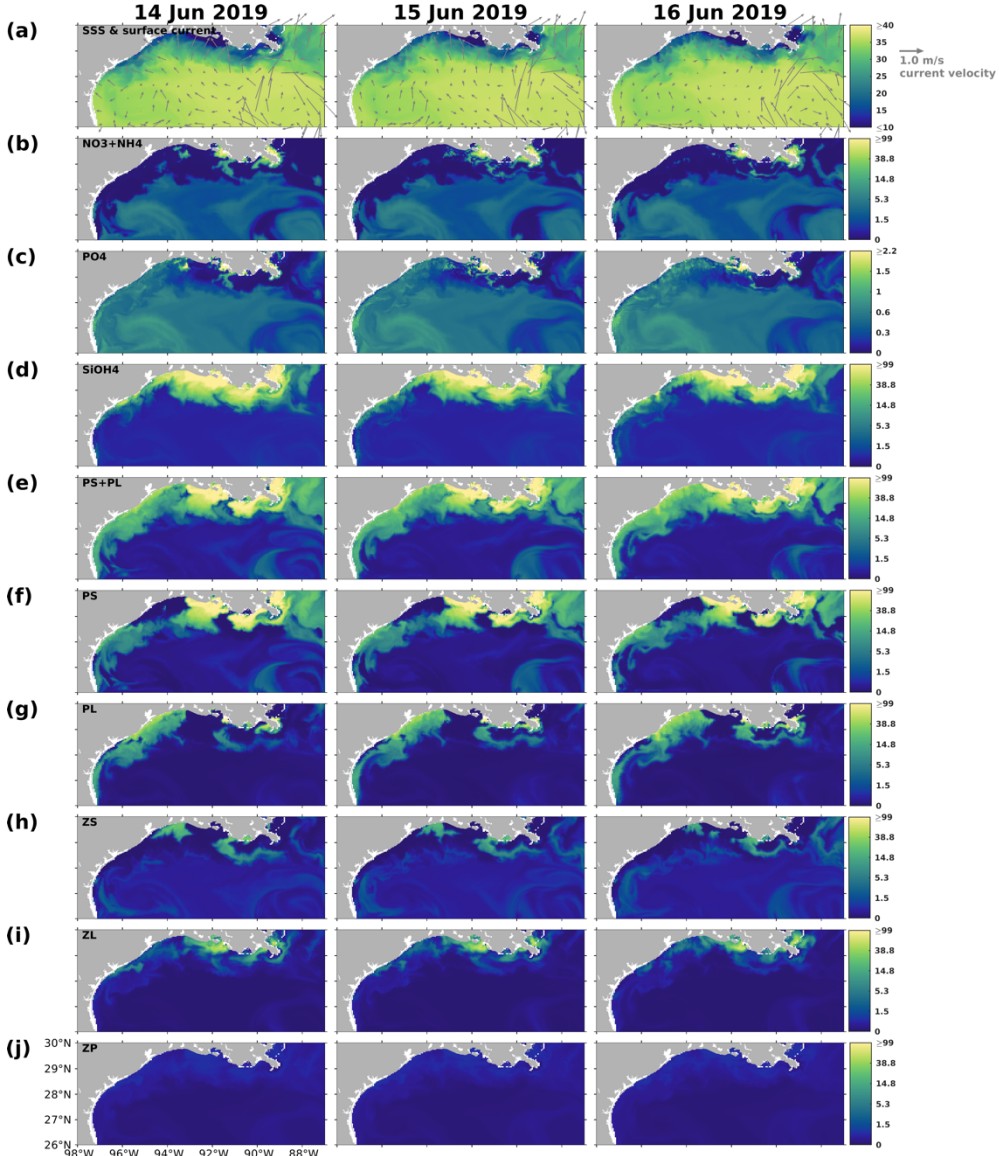

**Figure 14. Same as Fig. 13, but for snapshots from 14 June 2019 to 16 June 2019.**
**4.3 A re-examination of LaTex shelf DO dynamics**
In this section, we specified the bottom waters as the layers within 2 meters above the sea floor, while the upper waters
represented all layers above this 2-meter bottom layer. The purpose is to understand the contributions of different processes,
including water column biochemistry, air-sea flux (in upper layers), SOC (in bottom layers), and water transports
(advection+diffusion) to the daily variations of DO in the LaTex shelf during summers (May–August) of 2007–2020.

In the upper LaTex shelf, daily DO changes were primarily driven by shelf physics and local water column biochemistry (Fig. 15a), as reflected by their significant contributions to the variability and magnitude. The advection and diffusion terms together explained the greatest spatiotemporal variability of total DO changes. The ranges of the first and the third quartiles were closely shown in the total rate of changes (-124 to 107 mmol $O_2$ m$^{-2}$ day$^{-1}$) and changes by water transports (-117 to 72 mmol $O_2$ m$^{-2}$ day$^{-1}$). Detailed separation of the water transport terms indicated that horizontal advection of DO contributed the most to the variability of the physical terms. The water column biochemistry contributed the second largest to total DO variability, with a wide range of first and third quartiles (-41 to 96 mmol $O_2$ m$^{-2}$ day$^{-1}$). The phytoplankton groups contributed positively to the upper DO pool, with the majority contribution from the PS group. PS biomass was usually higher than PL biomass in summer when the allocation of nutrients was more favorable for the growth of PS. The net DO changes by water column biochemistry could be negative, indicating net metabolism, which was also reported by previous field studies demonstrating consistent net water column heterotrophy across the Louisiana shelf (e.g., Murrell et al., 2013). The air-sea interactions contributed negatively to the total DO changes and accounted for the least contribution. This indicated that the upper LaTex shelf was mostly a source of oxygen to the atmosphere during summer.

In the bottom layers, the DO variability was controlled by SOC and water transports (Fig. 15b). The SOC was steady (narrow range of quartiles), but major DO loss term (median= -32 mmol$O_2$ m$^{-2}$ day$^{-1}$, first quartile= -45 mmol$O_2$ m$^{-2}$ day$^{-1}$, and third quartile= -24 mmol$O_2$ m$^{-2}$ day$^{-1}$), driving the total rate of changes of DO to be negative at most shelf grids during summer (median= -8 mmol$O_2$ m$^{-2}$ day$^{-1}$ and first quartile= -32 mmol$O_2$ m$^{-2}$ day$^{-1}$, and third quartile= 11 mmol$O_2$ m$^{-2}$ day$^{-1}$). The advection and diffusion terms together acted as a major source of DO in the bottom layers (median=21 mmol$O_2$ m$^{-2}$ day$^{-1}$, first quartile=7 mmol$O_2$ m$^{-2}$ day$^{-1}$, and third quartile= 48 mmol$O_2$ m$^{-2}$ day$^{-1}$). However, they hardly offset the DO loss due to SOC. Such a positive contribution to DO by physical transports was mainly a result of steady and strong net DO supplies through vertical diffusion, as the variability and magnitude of DO changes due to total advection were less pronounced than those due to vertical diffusion. The vertical diffusion of DO is influenced by both water stratification and vertical DO concentration gradient. Water stratification results from multiple processes, including river plume dynamics, tidal dynamics, wind patterns, surface heating and cooling, etc. has been identified as an important indicator of bottom DO supply (Hetland and DiMarco, 2008; Bianchi et al., 2010; Fennel et al., 2011, 2013, 2016; Justić and Wang, 2014; Wang and Justić, 2009; Feng et al., 2014; Yu et al., 2015; Laurent et al., 2018). The variation of the vertical gradient was more related to the DO dynamics in the upper layers than in the bottom, as the DO variability is more pronounced in the upper layers (wider range in total rate of changes). Thus, while SOC and water stratification play crucial roles in DO changes in the bottom layers, DO changes in the upper shelf can affect the bottom DO through vertical diffusion.

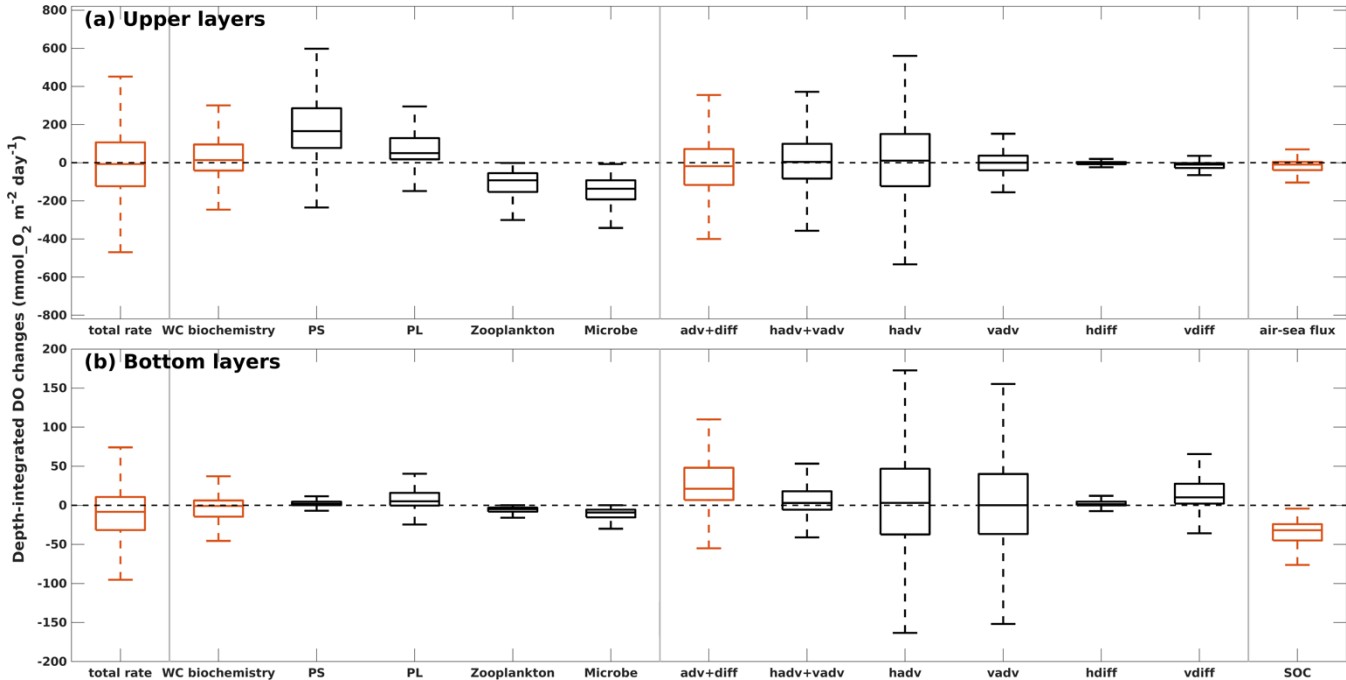

Figure 15. Depth-integrated rate of changes in DO due to different modeled processes in (a) the upper layers and (b) the bottom layers. The total rate of changes is the summation of DO sources/sinks by three groups of contributors (water column biochemistry, DO transports, and air-sea flux in upper layers or SOC in bottom layers) separated by vertical gray lines. In each group, DO changes by specific processes are illustrated by black boxes. Boxes represent the first and third quartiles, with lower and upper whiskers extending to the lowest and highest values within 1.5 interquartile range of the first and third quartiles, respectively. The median is indicated by a black line in the middle of the boxes. Statistics are summarized from the summers (May–August) records of 2007–2020 at all grid cells in the LaTex shelf.

The interactions within the plankton community (e.g., competition for nutrients and grazing pressure), which led to biomass differences, also resulted in different DO patterns at the bottom layer. Such impacts became more apparent when the DO contribution by water biochemistry outweighed that from transport processes in the upper ocean. For illustration, three summer snapshots of 14–16 June 2019 (Fig. 16 and 17) were sampled when widespread bottom hypoxia was detected. The water column biochemical processes contributed more than 50 % of total DO changes in most computational cells in the upper layers (Fig. 16a). First of all, the DO contribution by phytoplankton, zooplankton, and microbe exhibited distinct spatiotemporal patterns, complicating the net DO changes in the upper layers. Generally, the PS and PL groups enhanced DO levels, whereas zooplankton and microbes tended to deplete DO. During 14 June 2019, the DO losses by biochemical processes (Fig. 16b) in the shallow western shelf were mostly attributed to high ZS metabolism (Fig. 16e); the net DO gains between 91.5 and 92.5 °W reflected high PL concentrations (Fig. 14g) and the associated high DO supplies (Fig. 16d); the scattered DO losses over the shelf were primarily due to the homogenously high DO consumptions by microbes (Fig. 16h). During 15 and 16 June 2019, when DO supplies by PS and PL (Fig. 16c–16d) increased, net DO gains predominated in the shelf (Fig. 16b). However, the

net DO gains in the west (> 92.5 °W) and east (< 91.5 °W) shelf were mainly contributed by PS, while those in the middle
shelf by PL.

At the same time, changes in upper DO could affect the bottom DO through vertical diffusion, of which spatial patterns (mostly
positive; Fig. 17b) and daily variability aligned with biochemical DO alterations in the upper layers (Fig. 16b). However, water
column stratification, as indicated by the potential energy anomaly (PEA; Fig. 17a), resulted in noticeable spatial disparities
in the vertical diffusion of DO. On 15 June 2019, for example, the effects of vertical diffusion were weakened in areas that
featured strong stratification, as evidenced by high PEA values. In contrast, in regions of weak stratification, such as the
shallow waters between 90.5 and 92.5°W, vertical diffusion was markedly stronger. During the sampled period, among various
factors (i.e., total advection, horizontal diffusion, water-column biochemistry, and SOC), the vertical diffusion term
contributed the most to the total rate of changes in bottom DO, especially over the middle shallow shelf. As the rates of changes
were daily averaged and the bottom DO concentration was sampled at UTC 00:00 on each sampled day (Fig. 17i), the elevated
bottom DO level and relief of bottom hypoxia in the shallow middle shelf on 16 June 2019 were mainly due to the significant
vertical diffusion on the preceding day, driven by high PL-supported DO sources and weak water stratification. Thus, through
the interactions within the community in the upper ocean and DO diffusion processes between the upper and bottom layers,
the influence of planktonic community complexity on the bottom DO dynamics and the hypoxia evolution is evident.

The influence of SOC and water stratification on bottom hypoxia in the LaTex shelf has been well-documented. Yet, the role
of planktonic community complexity has received scant attention in prior numerical and observational studies. This study
devoted considerable effort to validating various factors, from nutrient dynamics (concentration and limitation types) to
phytoplankton composition (diatom ratio and temporal variations in total primary production) and oxygen variables (SOC, DO
profiles, and hypoxia patterns). Our findings illustrated how both bottom-up mechanisms (phytoplankton competition for
nutrients) and top-down effects (zooplankton grazing on phytoplankton) shape plankton composition, thereby influencing DO
levels in the upper water column and affecting subsequent changes in bottom DO and hypoxia patterns through physical
transports (e.g., vertical diffusion). The insights obtained suggest that the impacts of planktonic community complexity on
bottom DO and hypoxia patterns could be of high importance.

Nonetheless, incorporating a more complex plankton community in the model requires reasonable parameterizations for
different groups to represent their interactions. The large number of parameters can sometimes hamper the reliability of a
biogeochemical model due to the lack of support from in-situ observations or laboratory experiments. This is also a critical
reason why prevailing lower-trophic biogeochemical models are often "over-simplified". Even in complex models, the number
of plankton functional groups considered needs to be constrained to avoid over-parameterization. For example, there are two
phytoplankton and two zooplankton functional groups in PISCES (Aumont and Bopp, 2006) and CoSiNE models (Chai et al.,
2002), three phytoplankton and two zooplankton functional groups in PlankTOM5 model (Buitenhuis et al., 2010), and three
phytoplankton and one zooplankton functional groups in CCSM-BEC model (Moore et al., 2004).

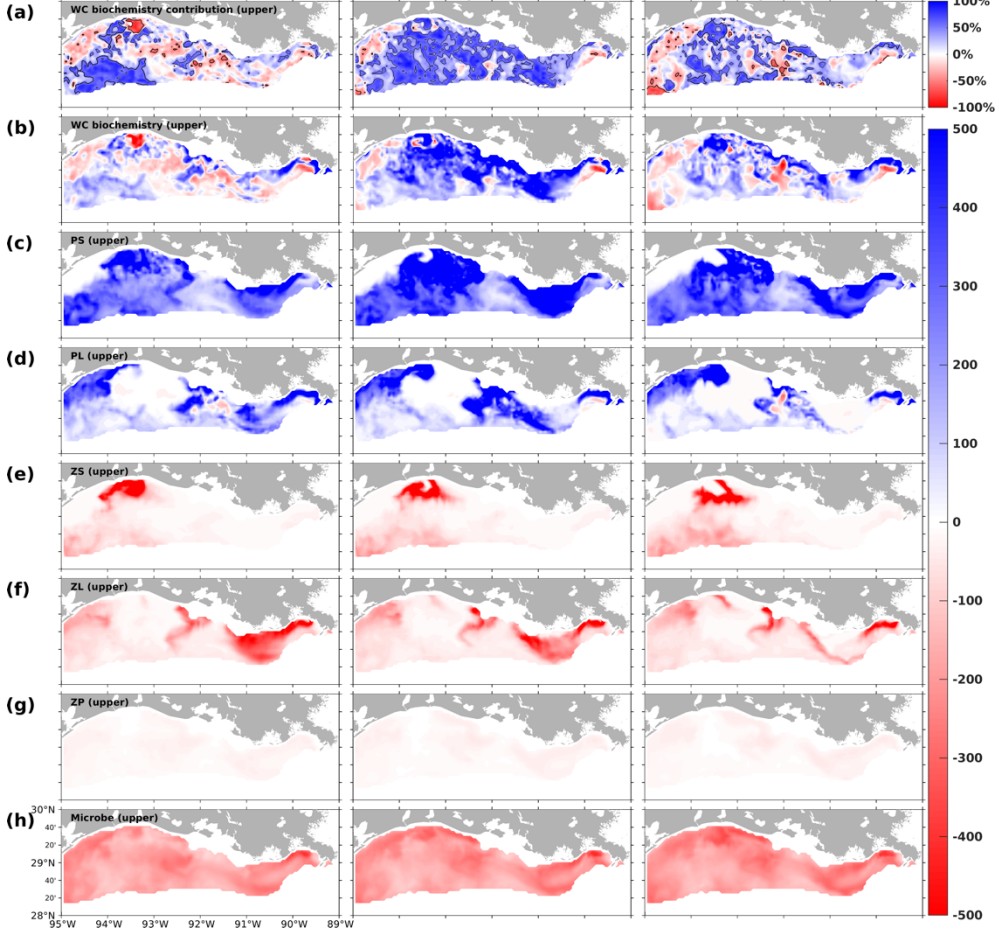

**Figure 16. Snapshots of DO contribution by the (a) water column biochemical processes (percentages) in the upper layers, DO gain/loss rates (mmol m$^{-2}$ day$^{-1}$) due to (b) water column biochemical processes, (c) PS, (d) PL, (e) ZS, (f) ZL, (g) ZP, and (h) microbe in the upper layers. The percentage contribution is related to the sum of absolute DO changes due to water column biochemical processes, water transports (advections and diffusions), and air-sea fluxes in the upper layers. The solid black lines in (a) indicate the -50% and 50% contour lines.**

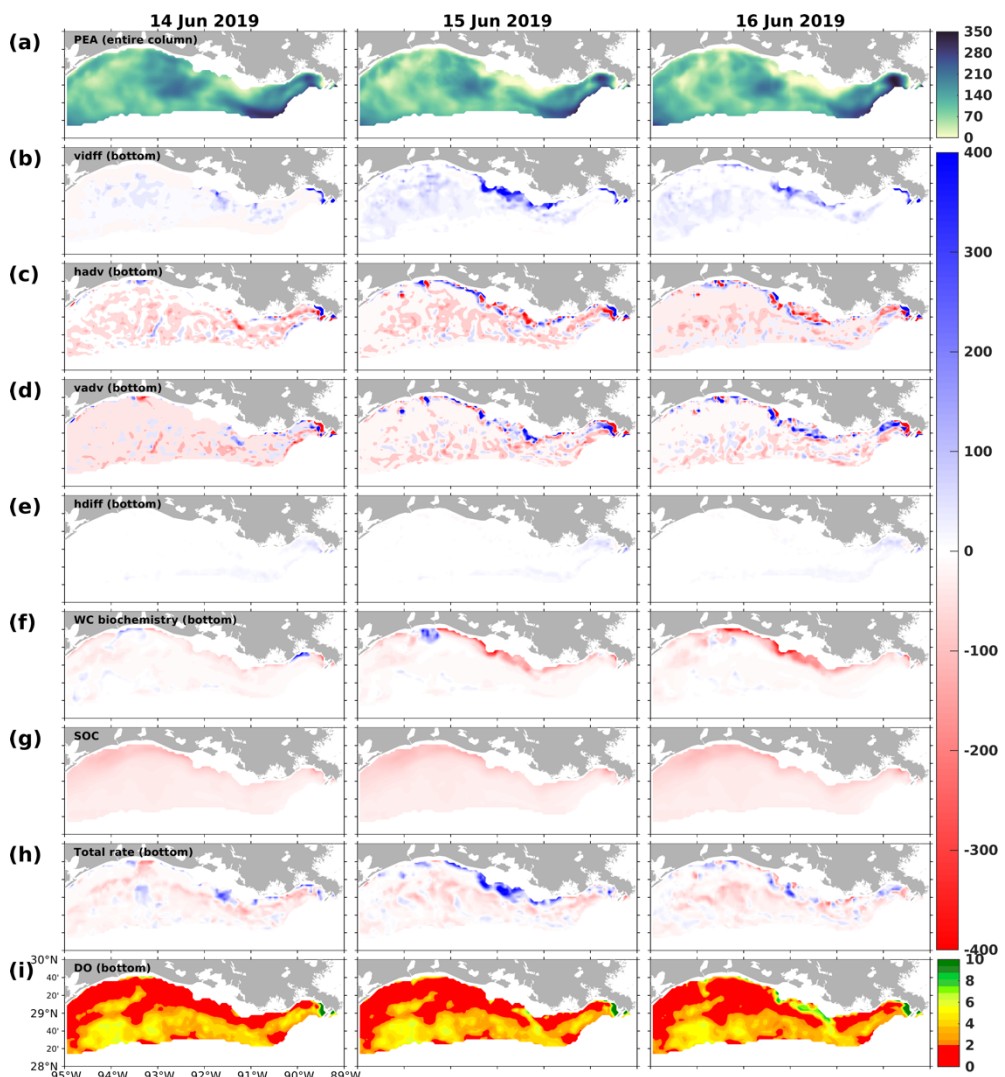

Figure 17. Snapshots of (a) potential energy anomaly (PEA; J m$^{-3}$), DO gain/loss rates (mmol m$^{-2}$ day$^{-1}$) due to (b) vertical diffusion (vdiff), (c) horizontal advection (hadv), (d) vertical advection (vadv), (e) horizontal diffusion (hdiff), (f) water column biochemical processes in the bottom layers, and (g) SOC, (h) total bottom DO gain/loss rates (mmol m$^{-2}$ day$^{-1}$), and (i) bottom DO concentration (mg L$^{-1}$). Rate snapshots are daily averages, while snapshots of state variables (i.e., PEA and bottom DO concentration) are extracted at UTC 00:00 on each sampled day.

## 5 Conclusions

In this study, we modified a three-dimensional coupled hydrodynamic–biogeochemical model (NEMURO) and adapted it to the GoM to investigate the mechanisms of bottom DO variability in the LaTex Shelf from 2007 to 2020. In addition to N and Si, a P flow was embedded into the NEMURO model to account for the impacts of P limitation on phytoplankton growth rates. Drawing upon the SOC scheme of the instantaneous remineralization developed by Fennel et al. (2006), a pool of sedimentary PON was added to capture temporal delays in SOC relative to the peak of plankton blooms. The model well reproduced the

surface inorganic nutrient concentration (i.e., nitrate, phosphate, and silicate), nutrient limitation patterns, the ratio of diatom
to total phytoplankton, and the magnitude of SOC. The model's robustness in DO simulation was affirmed via comparison of
the DO profiles against cruise observations from two different databases, comparison of spatial distributions of bottom DO,
and time series of the hypoxic area against the shelf-wide cruise observations.

Model results revealed that the changing dominated current system in summer can significantly alter the distribution of shelf
nutrients and types of nutrient limitations. While N and P limitation dominate the Mississippi and Atchafalaya River plume
area, Si limitation becomes pronounced as the coastal current system shifts from westward to eastward or northward,
facilitating the intrusion of low-Si waters from the west and the deep gulf. This effect, particularly evident on the western shelf,
has rarely been addressed in previous studies on nutrient limitation. Model results also indicated that under a westward
background current system, upwellings can enhance nearshore surface nutrient content, with the two modeled phytoplankton
functional groups, PS and PL, exhibiting distinct responses to the redistribution of surface nutrients.

Our findings underscore the importance of incorporating complex community dynamics and sophisticated nonlinear
interactions into biogeochemical models to capture the variability in primary production on the LaTex Shelf. The model
identified a bi-peak production pattern in spring and early summer, aligning with satellite-derived chlorophyll *a* variations – a
pattern not commonly reported in earlier research. We linked this bi-peak pattern to plankton community interactions,
including both bottom-up and top-down effects, as demonstrated in the sampled spring and summer snapshots. Changes in
nutrient distribution arising from interactions between the LaTex shelf and its adjacent waters, the passages of LCE, the
formation of upwelling or downwelling systems, and variations in river plume patterns are crucial in influencing plankton
interactions, highlighting the important role of open ocean dynamics and boundary conditions along the LaTex shelf in LaTex
biogeochemical modeling.

While the effects of SOC and water stratification on bottom hypoxia are well-recognized, our study illuminates how plankton
composition, influenced by bottom-up and top-down effects, can affect DO levels in the upper water column and lead to
changes in bottom DO and hypoxia patterns through physical transport processes, such as vertical diffusion. These insights
suggest the potential impacts of planktonic community complexity on bottom DO and hypoxia patterns, emphasizing the need
for future *in situ* and modeling efforts.

**Code/Data availability:** Model data is available at the LSU mass storage system and details are on the webpage of the
Coupled Ocean Modeling Group at LSU (https://faculty.lsu.edu/zxue/). Data requests can be sent to the corresponding
author via this webpage.

**Author contribution:** Z. George Xue designed the experiments and Yanda Ou carried them out. Yanda Ou developed the
model code and performed the simulations. Yanda Ou and Z. George Xue prepared the manuscript.
**Competing interests:** The authors declare that they have no conflict of interest.
**Acknowledgment:** Research support was provided through the Bureau of Ocean Energy Management (M17AC00019,
M20AC10001). We thank Dr. Jerome Fiechter at UC Santa Cruz for sharing his NEMURO model codes and Dr. Katja Fennel
at Dalhousie University for discussing model parameterization. The computational resource was provided by the High-
Performance Computing Facility (clusters SuperMIC and QueenBee3) at Louisiana State University.

**Appendix A: Expressions of processes terms modified in this study**
Detailed descriptions of related terms and parameters are listed in Appendix B.
**A1 Update gross primary production of PS and PL due to the additional phosphate limitation**
$GppPSn = GppNPS + GppAPS,$ (A1)
$GppPLn = GppNPL + GppAPL,$ (A2)
where,
$GppNPS = PSn\, V_{maxS}\, exp(K_{GppS}\, TMP)\left[1 - exp\left(-\frac{\alpha_{PS}}{V_{maxS}} I_{PS}\right)\right] exp\left(-\frac{\beta_{PS}}{V_{maxS}} I_{PS}\right) NutlimPS\, RnewS,$ (A3)
$GppAPS = PSn\, V_{maxS}\, exp(K_{GppS}\, TMP)\left[1 - exp\left(-\frac{\alpha_{PS}}{V_{maxS}} I_{PS}\right)\right] exp\left(-\frac{\beta_{PS}}{V_{maxS}} I_{PS}\right) NutlimPS\, (1 - RnewS),$ (A4)
$GppNPL = PLn\, V_{maxL}\, exp(K_{GppL}\, TMP)\left[1 - exp\left(-\frac{\alpha_{PL}}{V_{maxL}} I_{PL}\right)\right] exp\left(-\frac{\beta_{PL}}{V_{maxL}} I_{PL}\right) NutlimPL\, RnewL,$ (A5)
$GppAPL = PLn\, V_{maxL}\, exp(K_{GppL}\, TMP)\left[1 - exp\left(-\frac{\alpha_{PL}}{V_{maxL}} I_{PL}\right)\right] exp\left(-\frac{\beta_{PL}}{V_{maxL}} I_{PL}\right) NutlimPL\, (1 - RnewL),$ (A6)

$RnewS = \dfrac{NO_3}{(NO_3+K_{NO_3S})\left(1+\frac{NH_4}{K_{NH_4S}}\right)} \dfrac{1}{\frac{NO_3}{(NO_3+K_{NO_3S})\left(1+\frac{NH_4}{K_{NH_4S}}\right)}+\frac{NH_4}{NH_4+K_{NH_4S}}},$ (A7)
$RnewL = \dfrac{NO_3}{(NO_3+K_{NO_3L})\left(1+\frac{NH_4}{K_{NH_4L}}\right)} \dfrac{1}{\frac{NO_3}{(NO_3+K_{NO_3L})\left(1+\frac{NH_4}{K_{NH_4L}}\right)}+\frac{NH_4}{NH_4+K_{NH_4L}}},$ (A8)
$NutlimPS = min\left(\dfrac{NO_3}{(NO_3+K_{NO_3S})\left(1+\frac{NH_4}{K_{NH_4S}}\right)}+\dfrac{NH_4}{NH_4+K_{NH_4S}},\dfrac{PO_4}{PO_4+K_{PO_4S}}\right),$ (A9)
$NutlimPL = min\left(\dfrac{NO_3}{(NO_3+K_{NO_3L})\left(1+\frac{NH_4}{K_{NH_4L}}\right)}+\dfrac{NH_4}{NH_4+K_{NH_4L}},\dfrac{PO_4}{PO_4+K_{PO_4L}},\dfrac{SiOH_4}{SiOH_4+K_{SiOH_4L}}\right),$ (A10)
$I_{PS} = PAR\, frac\, exp\left\{z\, AttSW + AttPS \int_z^0 [PSn(\zeta) + PLn(\zeta)]d\zeta\right\},$ (A11)
$I_{PL} = PAR\, frac\, exp\left\{z\, AttSW + AttPL \int_z^0 [PSn(\zeta) + PLn(\zeta)]d\zeta\right\},$ (A12)
**A2 Update aerobic decomposition from PON to NH₄ and from DON to NH₄ due to the introduction of oxygen**
**dependency**
$DecP2N = PON\, VP2N_0\, exp(K_{P2N}\, TMP)\, \hat{r},$ (A13)
$DecD2N = PON\, VD2N_0\, exp(K_{D2N}\, TMP)\, \hat{r},$ (A14)
where,
$\hat{r} = max\left[\dfrac{max(0, Oxyg - Oxyg_{th})}{K_{Oxyg}+Oxyg-Oxyg_{th}}, 0\right],$ (A15)
**A3 Update water column nitrification due to the introduction of oxygen dependency and light limitation**
$Nit = Nit_0 exp(K_{Nit} TMP) LgtlimN \hat{r},$ (A16)
where,
$LgtlimN = 1 - max\left(0, \frac{I_N - I_0}{I_N - I_0 + k_I}\right),$ (A17)
$I_N = PAR\ frac\ exp\left\{z\ AttSW + max(AttPS, AttPL)\int_z^0 [PSn(\zeta) + PLn(\zeta)]d\zeta\right\},$ (A18)
**A4 Additional SOC term:**
$SOC = 8.3865\ PON_{sed}\ VP2N_0\ exp(K_{P2N}\ TMP),$ (A19)
**Appendix B: Descriptions of terms and parameters**
**Table B1. Descriptions of state variables**

| Terms | Description | Unit |
|---|---|---|
| $NH_4$ | Ammonium concentration | mmolN m$^{-3}$ |
| $NO_3$ | Nitrate concentration | mmolN m$^{-3}$ |
| $PO_4$ | Phosphate concentration | mmolP m$^{-3}$ |
| $DOP$ | Dissolved organic phosphorus concentration | mmolP m$^{-3}$ |
| $POP$ | Particulate organic phosphorus concentration | mmolP m$^{-3}$ |
| $SiOH_4$ | Silicate concentration | mmolSi m$^{-3}$ |
| $PSn$ | Small phytoplankton biomass concentration measured in nitrogen | mmolN m$^{-3}$ |
| $PLn$ | Large phytoplankton biomass concentration measured in nitrogen | mmolN m$^{-3}$ |
| $Oxyg$ | Dissolved oxygen concentration | mmolO$_2$ m$^{-3}$ |


**Table B2 Descriptions of related terms involved in the phosphorus cycle and nutrient limitation. Superscripts "*" and "+" denote**
**that the mathematic expressions of corresponding terms are the same as those in Kishi et al. (2007) and Shropshire et al. (2020),**
**respectively. Expressions of terms with no superscript are updated and reported in Appendix A.**

| Terms | Description | Unit |
|---|---|---|
| $DecP2N$ | Decomposition rate from PON to NH$_4$ | mmolN m$^{-3}$ day$^{-1}$ |
| $DecD2N$ | Decomposition rate from DON to NH$_4$ | mmolN m$^{-3}$ day$^{-1}$ |
| $DecP2D^{*+}$ | Decomposition rate from PON to DON | mmolN m$^{-3}$ day$^{-1}$ |
| $EgeZLn^{+}$ | Large zooplankton egestion rate measured in nitrogen | mmolN m$^{-3}$ day$^{-1}$ |
| $EgeZPn^{*+}$ | Predatory zooplankton egestion rate measured in nitrogen | mmolN m$^{-3}$ day$^{-1}$ |

| $EgeZSn^{*+}$ | Small zooplankton egestion rate measured in nitrogen | mmolN m$^{-3}$ day$^{-1}$ |
|---|---|---|
| $ExcPSn^{*+}$ | Small phytoplankton extracellular excretion rate to DON and is measured in nitrogen | mmolN m$^{-3}$ day$^{-1}$ |
| $ExcPLn^{*+}$ | Large phytoplankton extracellular excretion rate to DON and is measured in nitrogen | mmolN m$^{-3}$ day$^{-1}$ |
| $ExcZSn^{*+}$ | Small zooplankton excretion rate to NH$_4$ and is measured in nitrogen | mmolN m$^{-3}$ day$^{-1}$ |
| $ExcZLn^{+}$ | Large zooplankton excretion rate to NH$_4$ and is measured in nitrogen | mmolN m$^{-3}$ day$^{-1}$ |
| $ExcZPn^{*+}$ | Predatory zooplankton excretion rate to NH$_4$ and is measured in nitrogen | mmolN m$^{-3}$ day$^{-1}$ |
| $GppNPS$ | Small phytoplankton nitrate-induced gross primary production rate measured in nitrogen | mmolN m$^{-3}$ day$^{-1}$ |
| $GppAPS$ | Small phytoplankton ammonium-induced gross primary production rate measured in nitrogen | mmolN m$^{-3}$ day$^{-1}$ |
| $GppPSn$ | Small phytoplankton gross primary production rate measured in nitrogen | mmolN m$^{-3}$ day$^{-1}$ |
| $GppNPL$ | Large phytoplankton nitrate-induced gross primary production rate measured in nitrogen | mmolN m$^{-3}$ day$^{-1}$ |
| $GppAPL$ | Large phytoplankton ammonium-induced gross primary production rate measured in nitrogen | mmolN m$^{-3}$ day$^{-1}$ |
| $GppPLn$ | Large phytoplankton gross primary production rate measured in nitrogen | mmolN m$^{-3}$ day$^{-1}$ |
| $MorPSn^{+}$ | Small phytoplankton mortality rate measured in nitrogen | mmolN m$^{-3}$ day$^{-1}$ |
| $MorPLn^{+}$ | Large phytoplankton mortality rate measured in nitrogen | mmolN m$^{-3}$ day$^{-1}$ |
| $MorZSn^{+}$ | Small zooplankton mortality rate measured in nitrogen | mmolN m$^{-3}$ day$^{-1}$ |
| $MorZLn^{+}$ | Large zooplankton mortality rate measured in nitrogen | mmolN m$^{-3}$ day$^{-1}$ |
| $MorZPn^{*+}$ | Predatory zooplankton mortality rate measured in nitrogen | mmolN m$^{-3}$ day$^{-1}$ |
| $Nit$ | Nitrification rate | mmolN m$^{-3}$ day$^{-1}$ |
| $ResPSn^{*+}$ | Small phytoplankton respiration rate measured in nitrogen | mmolN m$^{-3}$ day$^{-1}$ |
| $ResPLn^{*+}$ | Large phytoplankton respiration rate measured in nitrogen | mmolN m$^{-3}$ day$^{-1}$ |
| $SOC$ | Sediment oxygen consumption rate | mmolO$_2$ m$^{-2}$ day$^{-1}$ |


**Table B3 Descriptions of other variables**

| Terms | Description | Unit |
|---|---|---|
| $I_{PS}$ | Photosynthetically available radiation for small phytoplankton | W m$^{-2}$ |
| $I_{PL}$ | Photosynthetically available radiation for large phytoplankton | W m$^{-2}$ |
| $I_N$ | Maximum photosynthetically available radiation | W m$^{-2}$ |
| $LgtlimN$ | Light inhibition on nitrification rate | no dimension |
| $NutlimPS$ | Nutrient limitation term for small phytoplankton | no dimension |
| $NutlimPL$ | Nutrient limitation term for large phytoplankton | no dimension |
| $PAR$ | Net short-wave radiation on water surface | W m$^{-2}$ |
| $\hat{r}$ | Oxygen inhibition on nitrification and aerobic decomposition rates | no dimension |
| $RnewS$ | The f-ratio of small phytoplankton which is defined by the ratio of nitrate uptake to total uptake of nitrate and ammonium | no dimension |
| $RnewL$ | The f-ratio of large phytoplankton which is defined by the ratio of nitrate uptake to total uptake of nitrate and ammonium | no dimension |
| $Thickness_{bot}$ | Thickness of the bottom water layer | m |
| $TMP$ | Water temperature | °C |
| $z, \zeta$ | Vertical coordinate which is negative below sea surface | m |


**Table B4. Descriptions and values of all model parameters. Superscripts "S", "L", "F06", and "F13" denote that the corresponding**
**parameters follow Shropshire et al. (2020), Laurent et al. (2012), Fennel et al. (2006), and Fennel et al. (2013), respectively.**
**Superscript "*" indicates the corresponding parameters are from this study.**

| Parameter | Description | Units | Values |
|---|---|---|---|
| | | Small phytoplankton | |
| $V_{maxS}$ | Small phytoplankton maximum photosynthetic rate at 0 ℃ | day$^{-1}$ | 0.4$^S$ |
| $K_{NO_3S}$ | Small Phytoplankton half saturation constant for nitrate | mmolN m$^{-3}$ | 0.5$^S$ |
| $K_{NH_4S}$ | Small Phytoplankton half saturation constant for ammonium | mmolN m$^{-3}$ | 0.1$^S$ |
| $K_{PO_4S}$ | Small Phytoplankton half saturation constant for phosphate | mmolP m$^{-3}$ | 0.03125 |
| $\alpha_{PS}$ | Small phytoplankton photochemical reaction coefficient, initial slope of P-I curve | m$^2$ W$^{-1}$ day$^{-1}$ | 0.1$^S$ |

| | | | |
|---|---|---|---|
| $\beta_{PS}$ | Small phytoplankton photoinhibition coefficient | $m^2 W^{-1} day^{-1}$ | 0.00045[S] |
| $Res_{PS0}$ | Small phytoplankton respiration rate at 0 °C | $day^{-1}$ | 0.03[S] |
| $Mor_{PS0}$ | Small phytoplankton mortality rate at 0 °C | $m^3 mmolN^{-1} day^{-1}$ | 0.002[S] |
| $\gamma_S$ | Ratio of extracellular excretion to photosynthesis for small phytoplankton | no dimension | 0.135[S] |
| $K_{GppS}$ | Small phytoplankton temperature coefficient for photosynthetic rate | $°C^{-1}$ | 0.0693[S] |
| $K_{ResPS}$ | Small phytoplankton temperature coefficient for respiration | $°C^{-1}$ | 0.0519[S] |
| $K_{MorPS}$ | Small phytoplankton temperature coefficient for mortality | $°C^{-1}$ | 0.0693[S] |
| Large phytoplankton | | | |
| $V_{maxL}$ | Large phytoplankton maximum photosynthetic rate at 0 °C | $day^{-1}$ | 0.8[S] |
| $K_{NO_3L}$ | Large Phytoplankton half saturation constant for nitrate | $mmolN\ m^{-3}$ | 3.0[S] |
| $K_{NH_4L}$ | Large Phytoplankton half saturation constant for ammonium | $mmolN\ m^{-3}$ | 0.3[S] |
| $K_{PO_4L}$ | Large Phytoplankton half saturation constant for phosphate | $mmolP\ m^{-3}$ | 0.1875 |
| $K_{SiOH_4L}$ | Large Phytoplankton half saturation constant for silicate | $mmolSi\ m^{-3}$ | 6.0[S] |
| $\alpha_{PL}$ | Large phytoplankton photochemical reaction coefficient, initial slope of P-I curve | $m^2 W^{-1} day^{-1}$ | 0.1[S] |
| $\beta_{PL}$ | Large phytoplankton photoinhibition coefficient | $m^2 W^{-1} day^{-1}$ | 0.00045[S] |
| $Res_{PL0}$ | Large phytoplankton respiration rate at 0 °C | $day^{-1}$ | 0.03[S] |
| $Mor_{PL0}$ | Large phytoplankton mortality rate at 0 °C | $m^3 mmolN^{-1} day^{-1}$ | 0.001[S] |

| | | | |
|---|---|---|---|
| $\gamma_L$ | Ratio of extracellular excretion to photosynthesis for large phytoplankton | no dimension | 0.135[S] |
| $K_{GppL}$ | Large phytoplankton temperature coefficient for photosynthetic rate | $°C^{-1}$ | 0.0693[S] |
| $K_{MorPL}$ | Large phytoplankton temperature coefficient for mortality | $°C^{-1}$ | 0.0693[S] |
| $K_{ResPL}$ | Large phytoplankton temperature coefficient for respiration | $°C^{-1}$ | 0.0693[S] |
| **Small zooplankton** | | | |
| $GR_{maxSps}$ | Small zooplankton maximum grazing rate on small phytoplankton at 0 ℃ | $day^{-1}$ | 0.6[S] |
| $\lambda_S$ | Ivlev constant of small zooplankton | $m^3\, mmolN^{-1}$ | 1.4[S] |
| $PS2ZS$ | Small zooplankton threshold value for grazing on small phytoplankton | $mmolN\, m^{-3}$ | 0.043[S] |
| $\alpha_{ZS}$ | Assimilation efficiency of small zooplankton | no dimension | 0.7[S] |
| $\beta_{ZS}$ | Growth efficiency of small zooplankton | no dimension | 0.3[S] |
| $Mor_{ZS0}$ | Small zooplankton mortality rate at 0 ℃ | $m^3\, mmolN^{-1}\, day^{-1}$ | 0.022[S] |
| $K_{GraS}$ | Small zooplankton temperature coefficient for grazing | $°C^{-1}$ | 0.0693[S] |
| $K_{MorZS}$ | Small zooplankton temperature coefficient for mortality | $°C^{-1}$ | 0.0693[S] |
| **Large zooplankton** | | | |
| $GR_{maxLps}$ | Large zooplankton maximum grazing rate on small phytoplankton at 0 ℃ | $day^{-1}$ | 0[S] |
| $GR_{maxLpl}$ | Large zooplankton maximum grazing rate on large phytoplankton at 0 ℃ | $day^{-1}$ | 0.3[S] |
| $GR_{maxLzs}$ | Large zooplankton maximum grazing rate on small zooplankton at 0 ℃ | $day^{-1}$ | 0.3[S] |
| $\lambda_L$ | Ivlev constant of large zooplankton | $m^3\, mmolN^{-1}$ | 1.4[S] |
| $PL2ZL$ | Large zooplankton threshold value for grazing on large phytoplankton | $mmolN\, m^{-3}$ | 0.040[S] |

| | | | |
|---|---|---|---|
| $ZS2ZL$ | Large zooplankton threshold value for grazing on small zooplankton | mmolN m$^{-3}$ | 0.040$^S$ |
| $\alpha_{ZL}$ | Assimilation efficiency of large zooplankton | no dimension | 0.7$^S$ |
| $\beta_{ZL}$ | Growth efficiency of large zooplankton | no dimension | 0.3$^S$ |
| $Mor_{ZL0}$ | Large zooplankton mortality rate at 0 ℃ | m$^3$ mmolN$^{-1}$ day$^{-1}$ | 0.022$^S$ |
| $K_{GraL}$ | Large zooplankton temperature coefficient for grazing | ℃$^{-1}$ | 0.0693$^S$ |
| $K_{MorZL}$ | Large zooplankton temperature coefficient for mortality | ℃$^{-1}$ | 0.0693$^S$ |

| Predatory zooplankton | | | |
|---|---|---|---|
| $GR_{maxPpl}$ | Predatory zooplankton maximum grazing rate on large phytoplankton at 0 ℃ | day$^{-1}$ | 0.1$^S$ |
| $GR_{maxPzs}$ | Predatory zooplankton maximum grazing rate on small zooplankton at 0 ℃ | day$^{-1}$ | 0.1$^S$ |
| $GR_{maxPzl}$ | Predatory zooplankton maximum grazing rate on large zooplankton at 0 ℃ | day$^{-1}$ | 0.3$^S$ |
| $\lambda_P$ | Ivlev constant of predatory zooplankton | m$^3$ mmolN$^{-1}$ | 1.4$^S$ |
| $PL2ZP$ | Predatory zooplankton threshold value for grazing on large phytoplankton | mmolN m$^{-3}$ | 0.040$^S$ |
| $ZS2ZP$ | Predatory zooplankton threshold value for grazing on small zooplankton | mmolN m$^{-3}$ | 0.040$^S$ |
| $ZL2ZP$ | Predatory zooplankton threshold value for grazing on large zooplankton | mmolN m$^{-3}$ | 0.040$^S$ |
| $\alpha_{ZP}$ | Assimilation efficiency of predatory zooplankton | no dimension | 0.7$^S$ |
| $\beta_{ZP}$ | Growth efficiency of predatory zooplankton | no dimension | 0.3$^S$ |
| $Mor_{ZP0}$ | Predatory zooplankton mortality rate at 0 ℃ | m$^3$ mmolN$^{-1}$ day$^{-1}$ | 0.12$^S$ |
| $K_{GraP}$ | Predatory zooplankton temperature coefficient for grazing | ℃$^{-1}$ | 0.0693$^S$ |

| | | | |
|---|---|---|---|
| $K_{MorZP}$ | Predatory zooplankton temperature coefficient for mortality | $°C^{-1}$ | 0.0693[S] |
| $\psi_{PL}$ | Grazing inhibition coefficient of predatory zooplankton grazing on large phytoplankton | $m^3\,mmolN^{-1}$ | 4.605[S] |
| $\psi_{ZS}$ | Grazing inhibition coefficient of predatory zooplankton grazing on small zooplankton | $m^3\,mmolN^{-1}$ | 3.01[S] |

| Light | | | |
|---|---|---|---|
| $AttSW$ | Light attenuation due to seawater | $m^{-1}$ | 0.03[S] |
| $AttPS$ | Light attenuation due to small phytoplankton, self-shading coefficient | $m^2\,mmolN^{-1}$ | 0.03[S] |
| $AttPL$ | Light attenuation due to large phytoplankton, self-shading coefficient | $m^2\,mmolN^{-1}$ | 0.03[S] |
| $frac$ | Fraction of shortwave radiation that is photosynthetically active | no dimension | 0.43[S] |
| $I_0$ | Threshold of light inhibition of nitrification | $W\,m^{-2}$ | 0.0095[F06] |
| $k_I$ | Light intensity at which light inhibition of nitrification is half-saturated | $W\,m^{-2}$ | 0.1[F06] |

| Water column nitrification and aerobic decomposition | | | |
|---|---|---|---|
| $Nit_0$ | Nitrification rate at 0 ℃ | $day^{-1}$ | 0.003[S] |
| $VP2N_0$ | Decomposition rate at 0 ℃ (PON→$NH_4$) | $day^{-1}$ | 0.01[S] |
| $VP2D_0$ | Decomposition rate at 0 ℃ (PON→DON) | $day^{-1}$ | 0.05[S] |
| $VD2N_0$ | Decomposition rate at 0 ℃ (DON→$NH_4$) | $day^{-1}$ | 0.02[S] |
| $VO2S_0$ | Decomposition rate at 0 ℃ (Opal→$Si(OH)_4$) | $day^{-1}$ | 0.01[S] |
| $K_{Nit}$ | Temperature coefficient for nitrification | $°C^{-1}$ | 0.0693[S] |
| $K_{P2D}$ | Temperature coefficient for decomposition (PON→DON) | $°C^{-1}$ | 0.0693[S] |
| $K_{P2N}$ | Temperature coefficient for decomposition (PON→$NH_4$) | $°C^{-1}$ | 0.0693[S] |
| $K_{D2N}$ | Temperature coefficient for decomposition (DON→$NH_4$) | $°C^{-1}$ | 0.0693[S] |

| | | | |
|---|---|---|---|
| $K_{O2S}$ | Temperature coefficient for decomposition (Opal→Si(OH)$_4$) | °C$^{-1}$ | 0.0693[S] |

| Other parameters | | | |
|---|---|---|---|
| $K_{Oxyg}$ | Oxygen concentration at which inhibition of nitrification and aerobic respiration are half-saturated | mmolO$_2$ m$^{-3}$ | 3.0[F13] |
| $Oxyg_{th}$ | Oxygen concentration threshold below which no aerobic respiration or nitrification occurs | mmolO$_2$ m$^{-3}$ | 6.0[F13] |
| $RPO4N$ | P: N ratio | mmolP mmolN$^{-1}$ | 1/16[L] |
| $RSiN$ | Si: N ratio | mmolSi mmolN$^{-1}$ | 1[S] |
| $rOxNO_3$ | Stoichiometric ratios corresponding to the oxygen produced per mol of nitrate assimilated during photosynthesis | mmolO$_2$ mmolNO$_3^{-1}$ | 138/16[F13] |
| $rOxNH_4$ | Stoichiometric ratios corresponding to the oxygen produced per mol of ammonium assimilated during photosynthesis | mmolO$_2$ mmolNH$_4^{-1}$ | 106/16[F13] |
| $setVPON$ | Sinking velocity of PON | m day$^{-1}$ | -5[*] |
| $setVOpal$ | Sinking velocity of Opal | m day$^{-1}$ | -5[*] |


**Appendix C: Supporting figures**

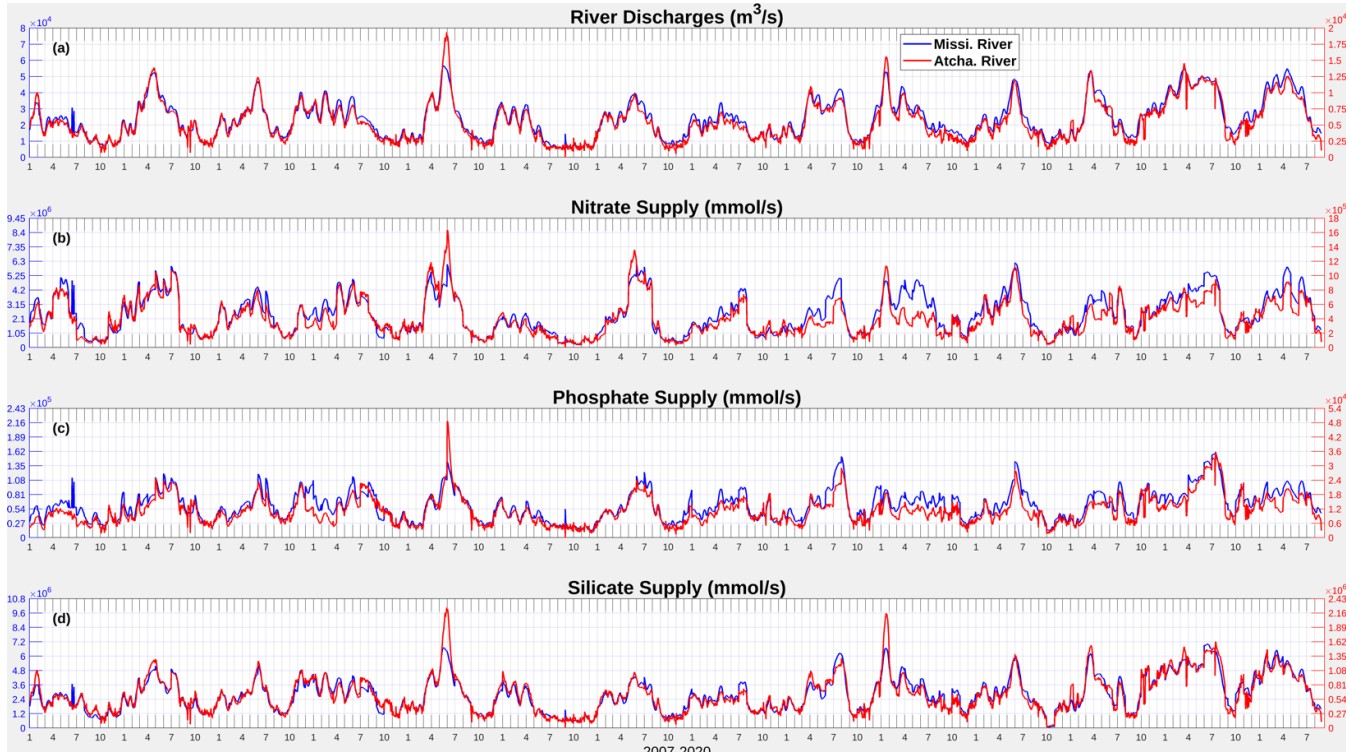


**Figure C1. Daily time series (2007–2020) of river discharges of freshwater, nitrate, phosphate, and silicate from the Mississippi and**
**Atchafalaya Rivers.**

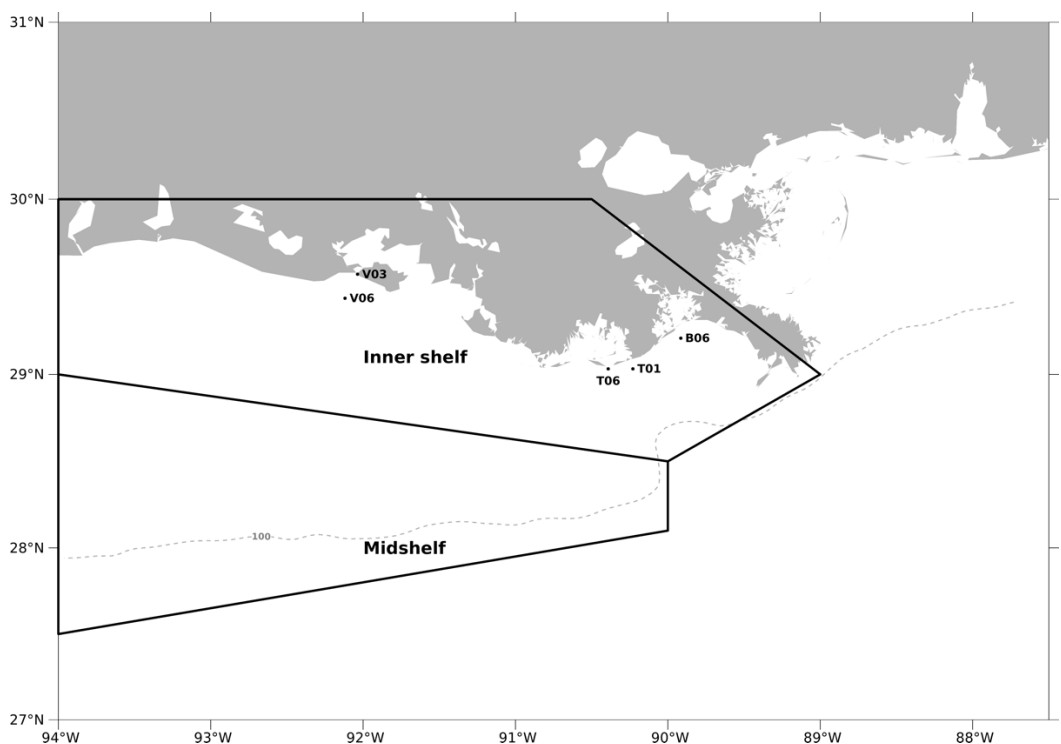

**Figure C2. The model computational meshes over which the regionally averaged diatom ratios are conducted for validation**
**purposes. Black dots indicate the sampling locations in Schaeffer et al. (2012), while the regions restricted by two black polygons are**
**two regions (i.e., inner shelf and mid-shelf) where samples were collected in Chakraborty and Lohrenz's (2015) study.**

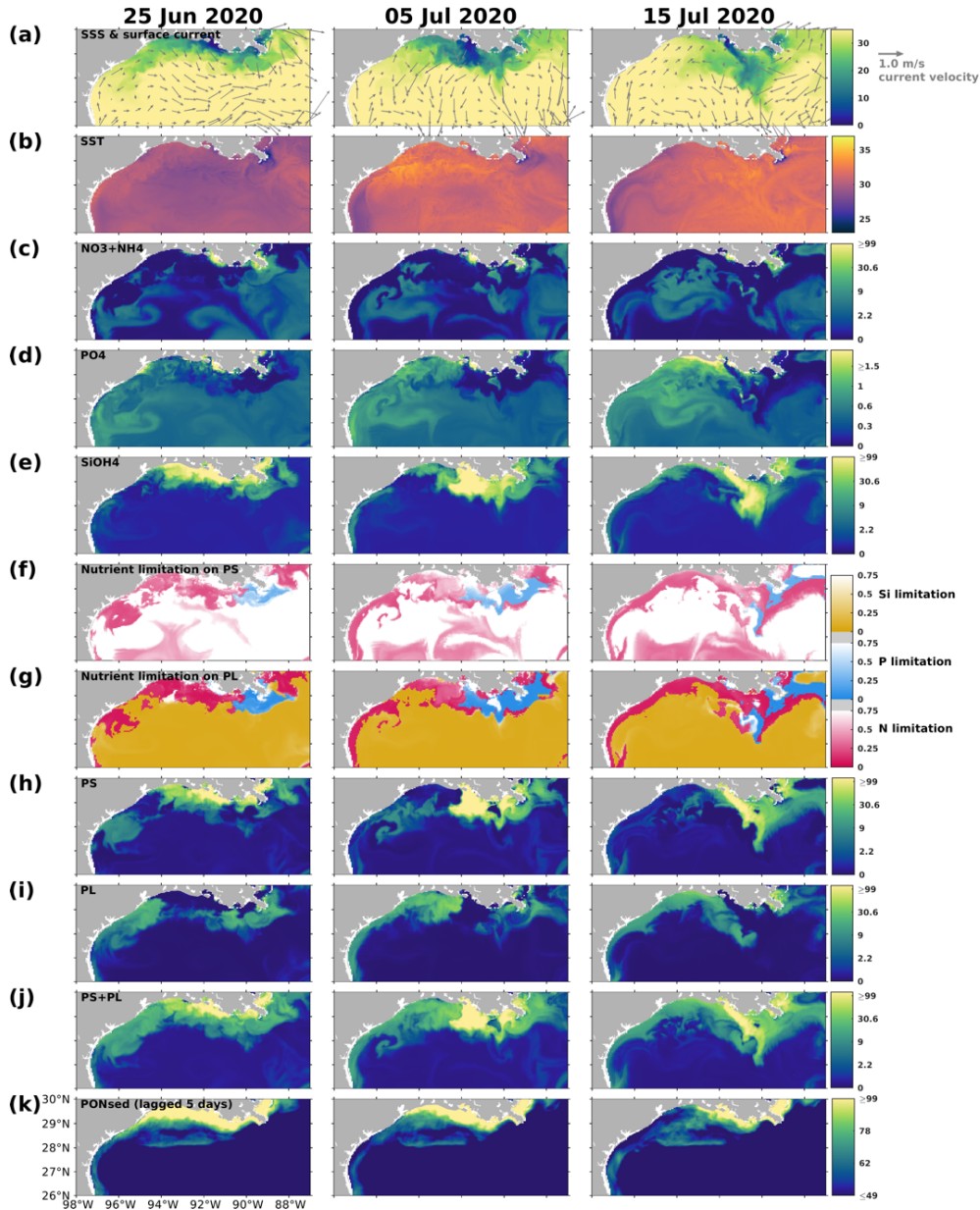


Figure C3. Summer snapshots of (a) sea surface salinity (overlayed with surface current velocity), (b) surface temperature (°C), (c) surface total inorganic nitrogen concentration (mmol N m⁻³), (d) surface phosphate concentration (mmol P m⁻³), (e) surface silicate concentration (mmol Si m⁻³), (f–g) surface nutrient limitation coefficients, (h–i) surface phytoplankton concentration (mmol N m⁻³), and (k) PON$_{sed}$ concentration (mmol N m⁻³) with a 5-day lag in the nGoM. The nutrient, phytoplankton, and PON$_{sed}$ concentrations are displayed in the log10 scale.

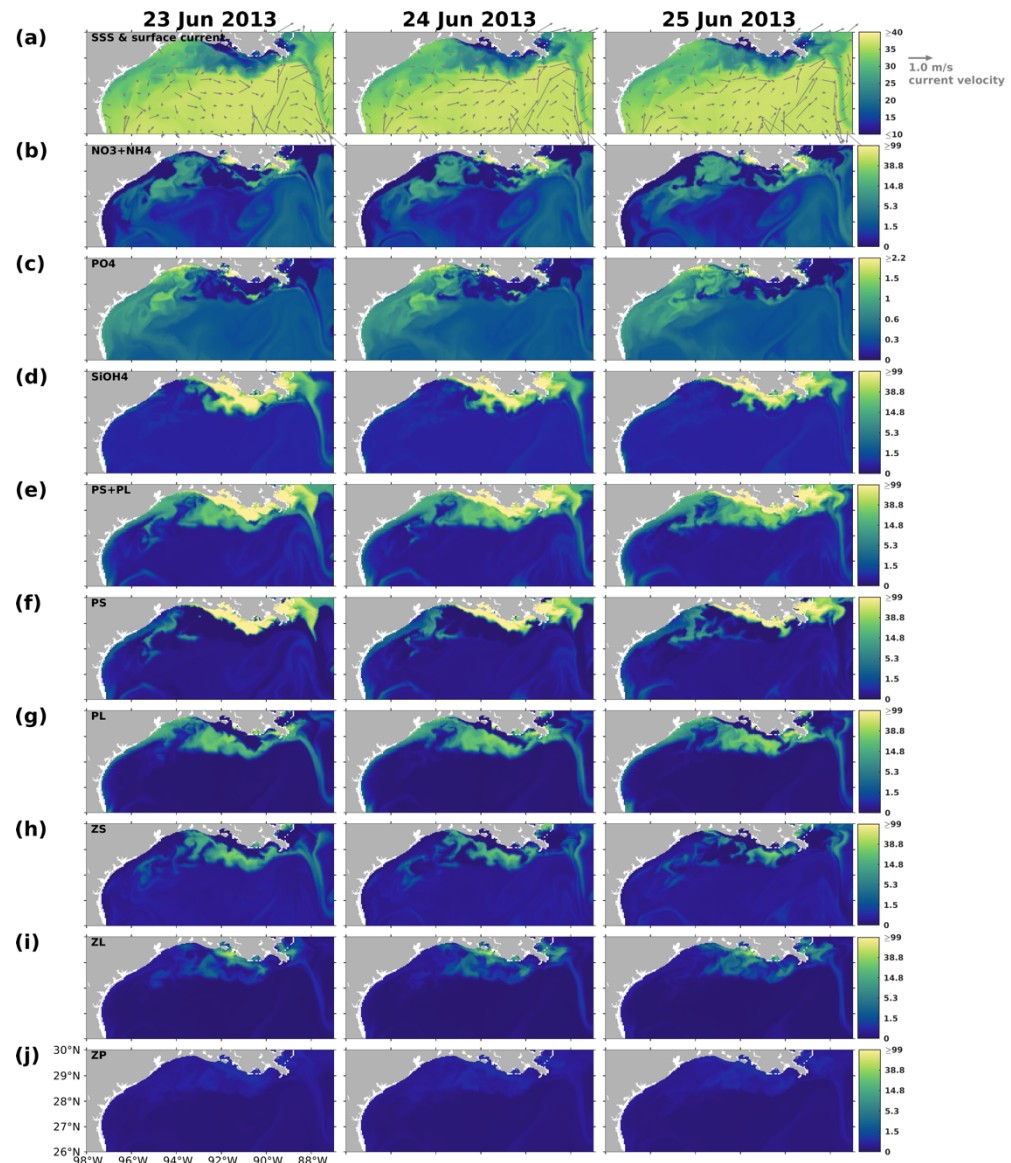

856

Figure C4. Snapshots of (a) sea surface salinity (overlayed with surface current velocity), (b) surface total inorganic nitrogen concentration (mmol N m⁻³), (c) surface phosphate concentration (mmol P m⁻³), (d) surface silicate concentration (mmol Si m⁻³), (e–g) surface phytoplankton concentration (mmol N m⁻³), and (h–j) surface zooplankton concentration (mmol N m⁻³). The nutrient and plankton concentrations are displayed in the log10 scale.

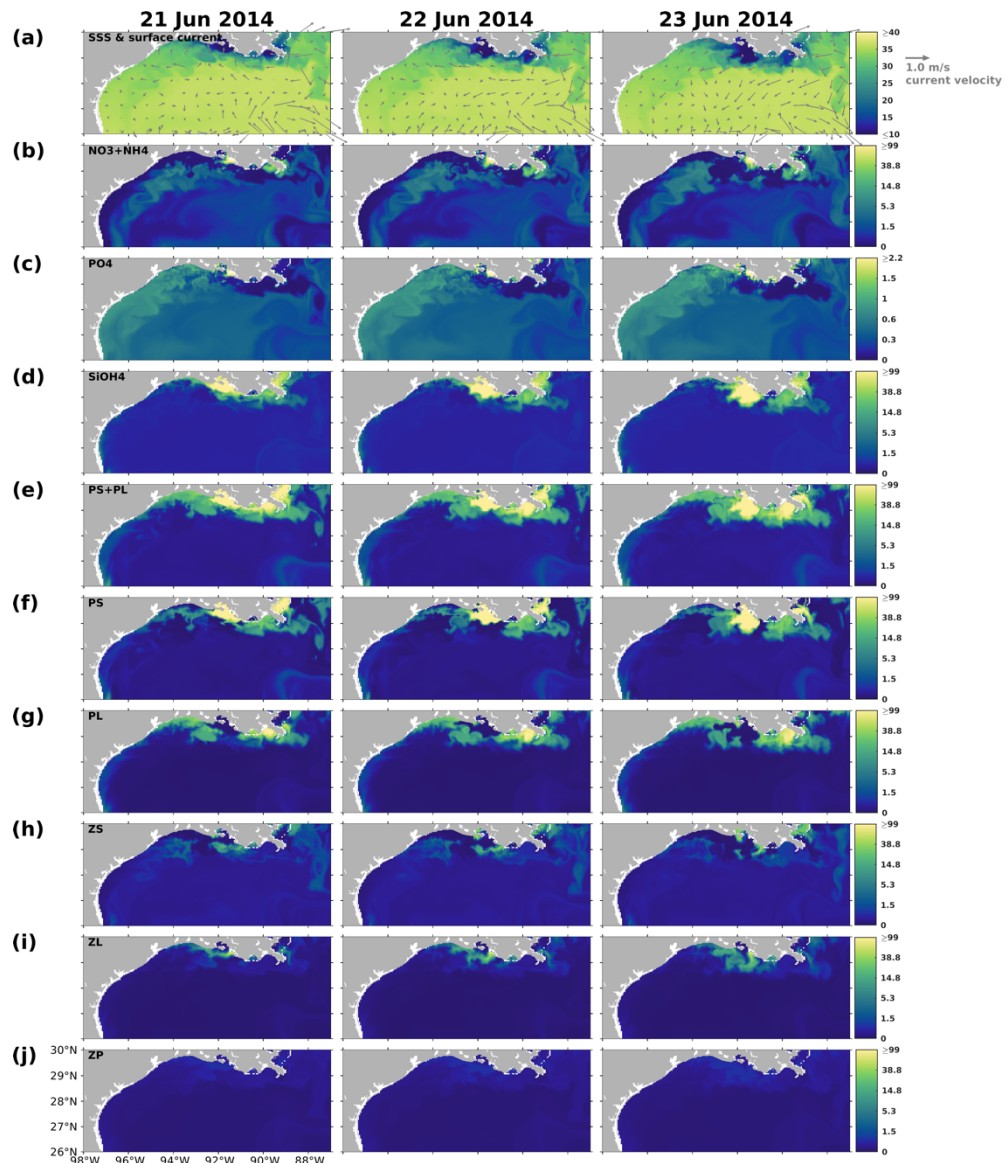


**Figure C5. Same as Fig. C4, but for snapshots from 21 June 2014 to 23 June 2014.**

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
