# Peer review of "Hydrodynamic and biochemical impacts on the development of hypoxia in the Louisiana–Texas shelf Part 1: roles of nutrient limitation and plankton community"

_Biogeosciences, 2022_

## Author Comment (AC1)

**Responses to Comments by Referee #1**

We thank Referee#1 by the detailed reviews he/she provided. Please find our detailed response below (*referee comments in Italic*).

**General comments:**

Overall: The manuscript describes the implementation of NEMURO in a ROMS-COAWST Gulf of Mexico model, including several new features that are targeted at studying hypoxia dynamics in the northern Gulf of Mexico. The main novelty is the inclusion of multiple phytoplankton and zooplankton functional types (from the *NEMURO* model), phosphorus, oxygen and a benthic layer that can accumulate PON. Using a 15 years simulation, the authors first carry out a validation of nutrient and oxygen, find that the model is able to reproduce the mid-summer hypoxic area and then analyze oxygen dynamics to show that 1) oxygen sinks in bottom waters are dominated by sediment oxygen consumption whereas the role of water column respiration is negligible, 2) hypoxia is controlled by SOC or PEA in the western and eastern part of the shelf, respectively, and 3) there is a quadratic relationship between the hypoxic volume and the hypoxic area, which can be used to predict hypoxic volume from the hypoxic area. My general assessment of the scientific content is that the manuscript lacks originality. There are some technical improvements from other models (see my technical assessment below) but the findings are mostly similar to previous studies using both observations and models, which are cited in the manuscript; the question is then what new knowledge does this study brings on the northern Gulf of Mexico hypoxia? This question should be central in the Introduction and in the Discussion.

**Response**: We acknowledge that new findings were not well presented in the first draft. For this study, the novelty of our model includes the incorporation, for the first time, of the silicate cycle in hypoxia simulation in the La-Tex shelf. Another novelty is the inclusion of multiple phytoplankton and zooplankton functional types. As the study region is dominated by the diatom community, we deem these two features of our model would provide new knowledge to the hypoxia research. We have been adding some sensitivity tests to further assess the importance of the silicate cycle in hypoxia development. Some previous results are discussed below and we will incorporate more relevant discussion during the revision stage.

Technical assessment: The model developed and used in this study seems appropriate, although I would like to discuss a few points that might need to be revised. These points are discussed in the specific comments below.

1) the main issue is the choice of a fast-sinking rate for the particulate organic matter. This choice results in the dominance of the sediment oxygen sinks, which is also a main conclusion of the study. The authors need to validate this part of the model (SOC versus water column respiration).

**Response**: To explore the sensitivity of sinking velocity in hypoxia development, we added two sensitivity tests with different sinking velocities, 1 m/day and 5 m/day, respectively. A most ideal selection of sinking velocity will be determined by the validation of SOC, the ratio of SOC and overlaying water respiration, bottom hypoxic area, and bottom hypoxic extent. Measured SOC and overlaying water respiration was reported by McCarthy et al., (2013), while the measured hypoxic area and extents are based on the Shelf-wide cruise. Following McCarthy et al., (2013), we extract the

daily SOC, and overlaying water respiration at sites F5, C6, B7, and MRM (Fig. 1 below) and averaged the observations by months.

Fig. 1 Map showing the location of the sampling site in the northern Gulf of Mexico in McCarthy et al., 2013.

Fig.2 indicates that a sinking velocity of 5 m/day provides the best estimate of SOC. The root-mean-squared errors (RMSEs) are 567  $\mu$ mol m-2 h-1, 713  $\mu$ mol m-2 h-1, and 452  $\mu$ mol m-2 h-1 for sensitivity tests with a sinking velocity of 15 m/day (used in the first draft), 1 m/day, and 5 m/day, respectively. The simulated (5 m/day) and observed SOC are generally in the same order of magnitude. The model results in general overestimate the SOC at sites F5 and C6 except for January 2009 and May 2010 at site C6, and underestimate SOC at sites B7 and MRM. Times series also reveals that the magnitude of simulated SOC by tests with a sinking velocity of 5 m/day is generally within the measured range (Fig. 3) over the entire year. The magnitude of simulated SOC by tests with a sinking velocity of 15 m/day is out of the upper measured bound especially in summers. Modeled SOC by the test with a sinking velocity of 1 m/day always yields a SOC below the measured ones.

Fig. 2 Comparison of observed SOC (in  $\mu$ mol m-2 h-1) by McCarthy et al., (2013) and simulated SOC by different sensitivity tests.

---

## Author Comment (AC2)

**Responses to Comments by Referee #2**

We thank Referee#2 for the detailed reviews he/she provided. Please find our detailed response below (*referee comments in Italic*).

**General comments:**

This manuscript by Ou et al. utilized a coupled physical-biogeochemical model to investigate the controlling factor of bottom hypoxia on the northern Gulf of Mexico and Louisiana-Texas Shelf. The authors added the phosphorus cycle and modified the sediment oxygen consumption module in an existing biogeochemical model NEMURO and coupled it with ROMS model. The coupled model was validated with observational data and then used to implement a 15-year hindcast simulation during 2006-2020. Then the authors explored the spatial variation of hypoxia development in the study area and found sediment oxygen consumption (SOC) and water column stratification are main factors to control the bottom oxygen in nearshore and offshore area respectively. Their model results also indicated separate hypoxia development schemes on the west and east Louisiana-Texas Shelf. Coastal deoxygenation is one of the most prominent environmental issues with important implications for marine ecosystem services. Although this paper made efforts to adopt a more sophisticated biogeochemical model with added phosphorus cycle and improved sediment oxygen consumption module, making contributions to investigate the spatial differences of dominant processes on hypoxia, it lacks original and novel aspects to explore the well-studied topic in this region, as well as comprehensive comparison with previous modeling study on the model performance, simulation results and conclusions, and address the question that how this new model stands out. The manuscript missed an advanced understanding and deep insight on the research topic of coastal hypoxia in a well-organized discussion section, thus this paper is a little thin on content. Although I see the value of this work, *I perceive that the publication is premature at this time.*

Response: Compared with existing modeling efforts, our model, for the first time, included a silicate cycle as well as multiple plankton functional groups, the importance of which has already been demonstrated in previous studies yet not included in hypoxia modeling efforts. We plan to include more results and discussion of the sensitivity tests focusing on the contribution of silicate limitation and the benefits of incorporating multiple plankton groups in the revision. The extensive model validation against nutrient and dissolved oxygen profiles confirmed the robustness of our model. In addition, another purpose of this model study is to provide the needed numerical solution for the development of a novel statistical model presented in paper Part II.

**Major comments:**

The hypoxia at the northern Gulf of Mexico has been well studied since the 1990s with increasing model studies in recent years. It ranged from a simple oxygen respiration model (Hetland&DiMarco, 2008) to a sophisticated coupled biogeochemical model (Laurent et al. 2012; Fennel et al. 2013). Including this study, they all generated similar conclusions that SOC is the controlling factor for hypoxia. In this sense, the improvement of complexity in the biogeochemical model does not make much sense. Also, the authors mentioned the additional work done on the NEMURO-based model filled gaps in phosphorus cycling and improved SOC representation. It's better to prove the advancement of the new model by validating with important variables, such as DO, Chla, PO4, NO3, with other model simulation studies.

**Response**: We carried out a series of sensitivity experiments to demonstrate the advance of incorporating the silicate cycle and will also provide more results to demonstrate the benefits of a more complicated plankton functional groups in DO simulation. We performed extensive model valuations against the DO (spatial distribution and vertical profiles) and nutrients (vertical profiles), and we will perform more validation for Chl-a.

We added some sensitivity tests to address the importance of silicate limitation. we plan to discuss silicate's impacts on hypoxia in two aspects: 1) the contribution of large phytoplankton (Pl, diatom); and 2) riverine silicate inputs. We were able to present some new model results with different riverine silicate inputs. The control run is symbolized as exp0 with the same setups as that in the first submission but with an updated sinking velocity changed to 5 m day-1 based on the above discussion (also see response below to comments "L195: SOC/THKbot is basically the oxygen consumption rate in the sediment. Why notintegrate SOC in the hypoxic area and get an overall integrated SOC?Any observational data validation on the newly added sediment and phosphorus module? In addition to the oxygen concentration validation?").

**Please note that the following contents regarding riverine silicate inputs are also included in our response to reviwer #1.**

We changed the riverine SiOH4 concentration from 0.2 to 2.0 with an increment of 0.2 in each experiment (exp1 through exp 9). For exp10 (ongoing), river silicate inputs were the same as that of the exp0, however, we removed the silicate limitation on large phytoplankton growth. More detail on these new experiment tests is listed in Table 1.

|                   | Simulation                | Sinking
velocity
(m day -
1 ) | Scale of
Riverine
SiOH4
concentration | Silicate
limitation
on Pl | #
phytoplankton
group | #
zooplankton
group |
|-------------------|---------------------------|--------------------------------------------------------------|------------------------------------------------|---------------------------------|-----------------------------|---------------------------|
| exp0
(control) | 1 Aug 2017–25
Aug 2020 | 5                                                            | 1                                              | Yes                             | 2                           | 3                         |
| exp1              | 1 Aug 2017–25
Aug 2020 | 5                                                            | 0.2                                            | Yes 2                           |                             | 3                         |
| exp2              | 1 Aug 2017–25
Aug 2020 | 5                                                            | 0.4                                            | Yes 2                           |                             | 3                         |
| exp3              | 1 Aug 2017–25
Aug 2020 | 5                                                            | 0.6                                            | Yes                             | 2                           | 3                         |
| exp4              | 1 Aug 2017–25
Aug 2020 | 5                                                            | 0.8                                            | Yes                             | 2                           | 3                         |
| exp5              | 1 Aug 2017–25
Aug 2020 | 5                                                            | 1.2                                            | Yes                             | 2                           | 3                         |
| exp6              | 1 Aug 2017–25
Aug 2020 | 5                                                            | 1.4                                            | Yes 2                           |                             | 3                         |
| exp7              | 1 Aug 2017–25
Aug 2020 | 5                                                            | 1.6                                            | Yes                             | 2                           | 3                         |
| exp8              | 1 Aug 2017–25
Aug 2020 | 5                                                            | 1.8                                            | Yes                             | 2                           | 3                         |
| exp9              | 1 Aug 2017–25
Aug 2020 | 5                                                            | 2.0                                            | Yes                             | 2                           | 3                         |
| exp10             | 1 Aug 2017–25
Aug 2020 | 5                                                            | 1.0                                            | No                              | 2                           | 3                         |

Table 1. Model setups of different sensitivity tests. Simulation of exp10 has not yet finished and thus updates will focus on results by testing exp0 through exp9.

The hypoxic area is estimated as the sum of the area of model grids when the bottom DO is less than 2 mg l-1. Percentage changes (Fig. 1) of the May–September hypoxic area are calculated between the sensitivity run (i.e., exp1–9) and the control run (exp0). The results of exp1 through exp9 indicate that the hypoxia area is mostly positively correlated with riverine silicate inputs. It is worth noting that the impact on the hypoxic area due to the changing riverine loads is not linear. The hypoxic area is more sensitive to elevated silicate supply especially when riverine input increases by more than 1.2. The average percentage increment of the hypoxic area ranges from ~35% to ~72% as the riverine silicate supply increase by 40% to 80%. In contrast, the hypoxic area decreases by ~19% to ~53% as the silicate supply decreases by 40% to 80%. When the changes in riverine silicate supply are less than 20%, the changes in the hypoxic area would be expected to be greater when riverine silicate is reduced (-19%) than increased (+12%).

We found three points that are new to coastal managers. Firstly, decreases in riverine silicate loads by 20% and 40% do not lead to significant differences in terms of hypoxic area reduction (by ~19% for both cases). Secondly, it is hard to meet the Gulf Hypoxia action plan goal of a 5,000 km2 hypoxic area by reducing the riverine silicate loads solely. The average summer hypoxic area from 2018 to 2020 is 15000 km2 which is comparable to that of 1985–2010 (14000 km2). Thus, to meet the Gulf Hypoxia action plan goal, the average percentage reduction of the hypoxic area should be ~67%. More discussion of the combined silicate and nitrogen reduction is needed and will be provided in the revision. Thirdly, as the range of 25th–75th percentile of hypoxic area changes enlarges as the riverine silicate load increases, an elevated riverine silicate input is likely to introduce much worse hypoxia events.

Fig. 1 Percentage differences of the simulated hypoxic area between sensitivity tests (exp1–9) and the control run (exp0). Statistics are based on the simulations in May– September from 2008 to 2020.

The differences in the spatial DO distribution between the riverine inputs sensitivity test (i.e., exp1–9) and the control run in August (2018–2020 average) are shown in Fig. 2. When riverine silicate is reduced, the low slope west shelf is more

sensitive to the changing silicate supply than the east shelf. In the cross-shelf direction, bottom DO between 10–50 m isobaths is more sensitive to the reduction of riverine silicate inputs than the rest regions. A slight decrease (by 20%; Fig. 2d) of silicate supply would lead to a maximum bottom DO increase of 2 mg O2 l-1 in this region. A 20% decrease in silicate supply can therefore easily induce a change from hypoxic to normoxic bottom waters in such regions. When riverine silicate inputs are increased, not much difference in the spatial distribution of DO reduction between the west and east part of the La-Tex shelf until the increase is more than 80%—then DO drops more in the west part of the shelf than the east part.

The above results show the impacts of riverine silicate loads on the bottom hypoxic area and bottom hypoxic water extent. We found that 1) the distribution of the hypoxic waters is more sensitive to the elevated riverine silicate loads with greater uncertainties than the reduced inputs; 2) a dual or triple silicate reduction is needed to meet the goal of the Gulf Hypoxia action plan; 3) the responses of bottom DO concentration is not spatially homogeneous along the shelf when riverine silicate loads are adjusted, and 4) the west shelf will suffer more from hypoxia conditions when riverine silicate is increased by more than 60%. We will provide further analysis and discussion on this topic in the revision including a recommendation of combined nitrogen and silicate reduction to meet the goal of the Gulf Action Plan.

Fig. 2 Differences in bottom DO (in mg O2 l-1) between experiments with different riverine inputs and the control run (August mean of 2018–2020).

The oxygen balance analysis is confusing and questionable. Although SOC is the dominant process in the bottom hypoxia generation (You et al. 2015), water column respiration (WCR) should not be orders of magnitude smaller than SOC, especially in the whole water column, as shown in Figure 15 and L455-456. Observational studies still showed varying evidence on SOC contribution (Murrell&Lehrter, 2011; Quiñones-Rivera, et al. 2010). More importantly, the reviewer has a sense that the authors did not understand and explain the oxygen dynamics well (Figure 10 and 15, section 4.2).

What is oxygen balance in the text? Based on L450-452, it should be water column respiration plus phytoplankton photosynthesis. This is a very confusing term and the physical transport of oxygen was totally missing. A lot of oxygen studies utilized standard oxygen budget analysis to separate dynamic terms in oxygen change (Li et al. 2014; Scully 2013; Yu et al. 2015). Please refer to those studies on the analysis and consider recalculating/rewriting this part.

**Response**: We agree with Reviewer#2 that the impact of WCR should not be neglected. In those figures, we mainly present the monthly climatology of model results which might make WCR less important. We plan to output all terms related to oxygen budget, including diffusivity, and advection, and calculate oxygen budget following the literature suggested by Reviewer#2.

Although this study employed sophisticated machine learning techniques to determine the controlling mechanisms on hypoxia in different regions. It could be actually achieved by oxygen budget analysis, with much clear representation in physical terms (advection and diffusion), rather than relying on stratification indicators. In addition, compared to the manipulating force on DO variability on a seasonal scale, the interannual variability is more of interest and worthy to look into.

**Response**: Following our responses to the above comments, we will provide a more comprehensive comparison of the DO balance including the local rate of changes, advection, and diffusion. According to both referees' comments, DO seasonality seems to be relatively well studied. We will look into interannual variability as well as the mechanism behind that in the revision.

The manuscript missed a comprehensive discussion section of advanced understanding of the study topic in-depth and in breadth. The overview of previous observational and model studies in this region, comparison with the current study, what are the agreements and differences, what are the causes, what are the defective aspects of this study, etc. are all important points to include. Expanding the implication to the global context is also valuable to discuss.

**Response:** We agree with reviewer #2 on this point. Our plan is to incorporate a series of sensitivity tests and more quantitative analysis regarding the contribution of silicate and the benefit of utilizing more c

---

## Author Response (AR1)

Dear Dr. Slomp:

Thanks again for giving us the chance to revise and improve our manuscript! After several months of revision and hundreds of more numerical experiments, we are pleased to let you know we successfully addressed the two reviewer's comments in round one of the submission. Here we are providing our point-to-point responses to the two referees.

Changes in model set-ups:
1. We performed a series of sensitivity runs to evaluate the model's robustness regarding different parameterizations of the sinking velocity of the organic matter (i.e., PON and Opal). And we conclude that a sinking velocity of 5 m/day is a reasonable prescription and updated relevant model results.
2. The riverine DO concentration was changed to be a constant of 258 mmol m$^{-3}$ assuming that the riverine DO was saturated at 25 °C under 1 atm.

Content removed:
1. The model validation for hypoxic thickness was removed since we focused more on the bottom DO dynamic.
2. We comprehensively revised the "Result and Discussion" section. Both reviewers pointed out that the 1$^{st}$ submission lacked originality. In this revision, we focus on 1) the contributions of different biogeochemical and hydrodynamic terms on bottom DO variability in different subregions, 2) the impacts of diatom and the complexity of lower-trophic community on the hypoxia dynamics, and 3) bottom DO's responses to the reduction of riverine nutrient loads with different reduction combination (percentage and nutrient type).

Content added:
1. The model validations of diatom ratios, sediment oxygen consumption (SOC), and ratios of SOC/ overlaying water respiration were added following the reviewers' suggestions.
2. Results and discussion on the factors to bottom DO variability was added (see section "4.1 Factors controlling subregion bottom DO variability" and "4.2 Stratification and Bottom DO Advection/Diffusion"). All terms that directly contribute to bottom DO changes were calculated and evaluated, which include horizontal advection, vertical advection, horizontal diffusion, vertical diffusion, the local rate of change (bottom DO), SOC, and DO changes due to water column biochemistry at the bottom layer. The summation of these terms contributes to the total changes in the bottom DO. A comparison of their contributions was given in different subregions of the LaTex shelf. We found that the most dominant factors to the bottom DO changes are the two advection terms, vertical diffusion term and SOC. The former three terms were associated with the changes in water column stratification. The strong linear correlations between PEA and the advection terms suggest that increased water stability in summer leads to fewer DO exchanges from advection processes. Nevertheless, the relationship between PEA and vertical diffusion of DO across the bottom layer appears to be non-linear.
3. Results from 16 sensitivity experiments for riverine nutrient reduction strategy were added (see section "4.3 Riverine nutrient reductions"). We found that the responses of summer hypoxia to the changing nutrient loads are not linear due

to the impacts of the complexity of the lower-trophic community. Nutrient reductions do not always guarantee a decrease in summer hypoxic areas; instead, due to the interactions among different plankton groups (e.g., competition on nutrients, grazing, and predation behaviors), the hypoxic zone could even increase under some nutrient reduction conditions. The most effective strategy is to simultaneously reduce the nitrogen, phosphorus, and silicon loads. A triple riverine nutrient reduction of 80% can help to fulfill the hypoxia reduction goal of 5000 $km^2$.

Sincerely,
Yanda Ou and Z. George Xue

**Responses to Comments by Referee #1**

*General comments:*
*Overall: The manuscript describes the implementation of NEMURO in a ROMS-COAWST Gulf of Mexico model, including several new features that are targeted at studying hypoxia dynamics in the northern Gulf of Mexico. The main novelty is the inclusion of multiple phytoplankton and zooplankton functional types (from the NEMURO model), phosphorus, oxygen and a benthic layer that can accumulate PON. Using a 15 years simulation, the authors first carry out a validation of nutrient and oxygen, find that the model is able to reproduce the mid-summer hypoxic area and then analyze oxygen dynamics to show that 1) oxygen sinks in bottom waters are dominated by sediment oxygen consumption whereas the role of water column respiration is negligible, 2) hypoxia is controlled by SOC or PEA in the western and eastern part of the shelf, respectively, and 3) there is a quadratic relationship between the hypoxic volume and the hypoxic area, which can be used to predict hypoxic volume from the hypoxic area. My general assessment of the scientific content is that the manuscript lacks originality. There are some technical improvements from other models (see my technical assessment below) but the findings are mostly similar to previous studies using both observations and models, which are cited in the manuscript; the question is then what new knowledge does this study brings on the northern Gulf of Mexico hypoxia? This question should be central in the Introduction and in the Discussion.*

**Response**: In this revision, we significantly changed the organization and presentation of the results. Some important new knowledge from our study includes 1)we quantified the physics (diffusion/advection) and biogeochemical control on hypoxia development in different subregions, 2) with the introduction of Si cycle and a complex plankton community, we found the non-linear relationship between riverine nutrient reduction and the changes of the size of the hypoxic zone. In details,

Changes in model set-ups:
1. We performed a series of sensitivity runs to evaluate the model's robustness regarding different parameterizations of the sinking velocity of the organic matter (i.e., PON and Opal). And we conclude that a sinking velocity of 5 m/day is a reasonable prescription and updated relevant model results.
2. The riverine DO concentration was changed to be a constant of 258 mmol m$^{-3}$ assuming that the riverine DO was saturated at 25 °C under 1 atm.

Content removed:
1. The model validation for hypoxic thickness was removed since we focused more on the bottom DO dynamic.
2. We rewrote the entire "Result and Discussion" section. Both reviewers pointed out that the manuscript lacked originality. In this revision we focus on 1) the contributions of different biogeochemical and hydrodynamic terms on bottom DO variability in different subregions, 2) the impacts of diatom and the complexity of lower-trophic community on the hypoxia dynamics 3) built on 2, we further access bottom DO's responses to the reduction of riverine nutrient loads with different reduction combination (percentage and nutrient type).

Content added:
1. The model validations of diatom ratios, sediment oxygen consumption (SOC), and ratios of SOC/ overlaying water respiration were added following the reviewers' suggestions.

2. Results and discussion on the factors to bottom DO variability was added (see section "4.1 Factors controlling subregion bottom DO variability" and "4.2 Stratification and Bottom DO Advection/Diffusion"). All terms that directly contribute to bottom DO changes were calculated and evaluated, which include horizontal advection, vertical advection, horizontal diffusion, vertical diffusion, the local rate of change (bottom DO), SOC, and DO changes due to water column biochemistry at the bottom layer. The summation of these terms contributes to the total changes in the bottom DO. A comparison of their contributions was given in different subregions of the LaTex shelf. We found that the most prevailing factors to the bottom DO changes are the two advection terms, vertical diffusion term, and SOC. The former three terms were associated with the changes in water column stratification. The strong linear correlations between PEA and the advection terms suggest that increased water stability in summer leads to fewer DO exchanges from advection processes. Nevertheless, the relationship between PEA and vertical diffusion of DO across the bottom layer appears to be non-linear.

3. Sensitivity experiments for riverine nutrient reduction strategy were added (see section "4.3 Riverine nutrient reductions"). We found the responses of summer hypoxia to the changing nutrient loads are not linear due to the impacts of the complexity of the lower-trophic community. Nutrient reductions do not always guarantee a reduction in summer hypoxic area, instead, due to the interactions among different plankton groups (e.g., competition on nutrients, grazing, and predation behaviors), the hypoxic area could even increase under some nutrient reduction conditions. The most effective strategy is to reduce the nitrogen, phosphorus, and silicon loads simultaneously. A triple riverine nutrient reduction of 80% can help to fulfill the hypoxia reduction goal of 5000 $km^2$.

*Technical assessment: The model developed and used in this study seems appropriate, although I would like to discuss a few points that might need to be revised. These points are discussed in the specific comments below.*
*1) the main issue is the choice of a fast-sinking rate for the particulate organic matter. This choice results in the dominance of the sediment oxygen sinks, which is also a main conclusion of the study.*

**Response**: Concerning the possibly improper sinking velocity applied in the model, we added two sensitivity experiments with different sinking velocities: 1 m/day and 5 m/day. A comparison of SOC against the observation by McCarthy et al., (2013) (Fig. RC1-1) suggests that the experiment with a sinking velocity of 5 m/day provides the best estimates (Fig. RC1-2). The root-mean-squared errors (RMSEs) are 567 $\mu$mol m$^{-2}$ h$^{-1}$, 713 $\mu$mol m$^{-2}$ h$^{-1}$, and 452 $\mu$mol m$^{-2}$ h$^{-1}$ for sensitivity tests with a sinking velocity of 15 m/day, 1 m/day, and 5 m/day, respectively. The model results (tests with sinking velocity = 5 m/day) generally overestimate the SOC at site F5 and C6 except for January 2009 and May 2010 at site C6, but underestimate SOC at site B7 and MRM. However, the simulated and observed SOC are generally in the same order of magnitude. Times series also reveals that the magnitude of simulated SOC simulated with a sinking velocity of 5 m/day is generally within the measured range (Fig. RC1-3) over the entire year. The magnitude of simulated SOC by tests with a sinking velocity of 15 m/day is out of the upper measured bound especially in summers. Modeled SOC by the test with a sinking velocity of 1 m/day is always below the lower measured bound.

We further compared the simulated ratio of SOC/overlaying water respiration against measurements (Fig. RC1-4). The test run with sinking velocity = 5 m/day

provides the best-simulated ratio with a low averaged RMSE of 4.23 over site F5, C6, and B7 compared with 4.58 (sinking velocity = 15 m/day) and 6.51 (sinking velocity = 1 m/day) derived by the other two tests. At site MRM, both the two tests with faster (5m and 15m/day) sinking velocity highly overestimate the ratio in August 2009. We ascribe such bias to the relative course bathymetry near the river mouths. Point sources are applied in the model for diverting momentum and concentration tracers from the river to the computational grid cells. The scheme can lead to an overshot of river water at the near-mouth grid cells, which, may further result in shorter residence time of organic matter and plankton.

We also compare the hypoxic area simulated by the three different experiments. The simulation with 5m and 15m/day settling velocity show a similar hypoxia zone, while the using 1m/day is generally greater than the former two (Fig. RC1-5). Both the former two estimations (5m and 15m/day) can reproduce the magnitude and interannual variability of the measured hypoxic area. Compared to the shelf-wide observations, the simulated bottom hypoxic extent (Fig. RC1-6– RC1-8) by the experiment with a sinking velocity of 5m/day produces less bias among the three experiments. Based on these results, we changed the sinking velocity of PON from 15 m day$^{-1}$ to 5 m day$^{-1}$ in the baseline simulations.

[Figure]

Fig. RC1-1 Map showing the location of sampling site in the northern Gulf of Mexico (McCarthy et al., 2013).

[Figure]

Fig. RC1-2 Comparison of observed SOC (in $\mu$mol m$^{-2}$ h$^{-1}$) by McCarthy et al., (2013) and simulated SOC by different sensitivity tests.

[Figure]

Fig. RC1-3 Daily average of simulated SOC by different sensitivity tests.

[Figure]

Fig. RC1-4 Comparison of observed ratio of SOC/overlaying water by McCarthy et al., (2013) and simulated ratio by different sensitivity tests.

[Figure]

Fig. RC1-5 Comparison of observed and simulated hypoxic area. Note that the horizontal red thick bars denote the shelf-wide cruise measurements.

[Figure]

Fig. RC1-6 Evolution of simulated bottom water dissolved oxygen concentration (unit mg l$^{-1}$) by the sensitivity experiment with a sinking velocity of 15 m day$^{-1}$. The black filled circles and open circles indicate the hypoxic site and non-hypoxic site, respectively, according to the Shelf-wide cruise observations. The grey curves denote bathymetry of 5, 10, 20, and 50 m.

[Figure]

Fig. RC1-7 Same as Fig. RC1-6 but for the sensitivity experiment with a sinking velocity of 1 m day$^{-1}$.

[Figure]

Fig. RC1-8 Same as Fig. RC1-6 but for the sensitivity experiment with a sinking velocity of 5 m day$^{-1}$.

*2) Looking at the results, it is not clear if the model is appropriately initialized/spun up. Hypoxia occurs in deep waters and a long-term deoxygenation trend occurs both inshore near the Atchafalaya and offshore. This seems to indicate that PON accumulate in the benthic layer nearshore throughout the simulation and that there is a drift in subsurface oxygen offshore.*

**Response**: We initialized the nitrate, phosphate, silicate, and dissolved oxygen based on the observations provided by the World Ocean Database and World Ocean Atlas. Other nutrients and plankton concentration terms were initialized spatially homogeneously as a small value. Physical terms were initialized using the HYCOM global analysis products. We did not find any "drift" in the sediment PON throughout the studied period. Sediment PON fluctuated between peaks and troughs and exhibited a salient annual cycle.

*3) the model does not include a light attenuation term from river sediment (near the river mouth). This could influence the timing and distribution of primary production over the shelf, and therefore affect the conclusions of the study.*

**Response**: The reason for the exclusion of a sediment module is to guarantee the model efficiency, which is also the practice of most of the hypoxia modeling efforts in this region. Our research group indeed published a paper regarding the impacts from sediment-induced light attenuation on shelf primary productivity (see Zang et al., 2020 Biogeosciences). Yet we found that introducing the sediment model into hypoxia simulation is computationally heavy. The model already needs 170 hours (~1 a week) with 500 CPU cores to finish a 15-year hindcast experiment. Nevertheless, based on extensive model validation (nutrient and DO profiles, hypoxia area distribution), we are confident that our current setup is capable of reproducing the general feature of hypoxia events.

*In term of model validation, model results are compared with many nutrients and oxygen data. However, the format of the model-observations comparison is questionable and does not result, in my opinion, in a satisfactory validation of the model.*

**Response**: We made significant changes in model validation in the revision. We replotted all the figures for model validation of nutrient and oxygen profiles (Figure 3 and Figure 6 in the manuscript). Probability histograms of nutrient differences between modeled and observed results were given for the upper 50 meters and upper 5 meters, respectively. Probability histograms of DO differences were given for the upper 50 meters.

For the validation of the diatom ratio, we compared the mean±1SD between simulated and observed values (Table 1 in the updated manuscript).

For the validation of SOC and the overlaying water respiration, we compared the mean SOC and the mean SOC/overlaying water respiration between simulations and observations using bar plots (Fig. 4 and 5 in the updated manuscript).

For the validation of the time series of hypoxic area, we provided the averaged modeled and measured value in the time series plot (Fig. 8 in the updated manuscript) and also the 5-year running $R^2$ (Table 2 in the updated manuscript).

*Manuscript assessment: both the Introduction and the Results/Discussion sections need some revisions. The Introduction review the literature of the northern Gulf of Mexico but does not assess what are the gaps in the knowledge. Rather, the authors propose technical improvements, which are welcomed but not sufficient. It is not clear, by the end of the manuscript, if using a more complex ecosystem model is an improvement over previous models. Although previous work is discussed relatively extensively in the Introduction, there is little discussion in the Results/Discussion section. Since similar studies have been carried out before, their results/findings should be compared. It would help to see what is the novelty of this study.*

**Response**:

In the section "1 Introduction", we added the following content in the last two paragraphs restating the main findings from the previous studies, the knowledge gaps, and the scientific questions we aimed to answer (Line 78-109). In this revision, we aimed to 1) understand the contributions of different factors in hypoxia development in different parts of the LaTex shelf and 2) to assess the outcomes of different riverine nutrient reduction scenarios regarding the reduction of the hypoxic zone within a complex lower-trophic ecosystem.

In the section "4.1 Factors controlling subregion bottom DO variability", our results indicated that the variability of bottom DO on the LaTex shelf was mostly controlled by four processes: horizontal advection, vertical advection, vertical diffusion, and SOC (Fig. 9b in the updated manuscript). Their impacts on bottom DO variability varied in different regions across a year, which, to our best knowledge, has not been reported by other studies. Although the importance of DO advection and SOC on bottom DO balance was also documented by Ruiz Xomchuk et al. (2021), vertical diffusion was proposed as a minor contributor in their study. Such a disagreement could result from the water layers investigated. Vertical diffusion of DO across the layer 10 m above the bottom was discussed in Ruiz Xomchuk et al. (2021), while here we estimated vertical diffusion of DO across the bottom layer.

In the section "4.2 Stratification and Bottom DO Advection/Diffusion", significant negative correlations were found between the PEA and the two absolute advection terms of bottom DO, while bottom DO flux due to vertical diffusion was found positively and moderately correlated to the PEA (Fig. 10 in the updated manuscript). Previous studies pointed out that water stratification affects the DO replenishment at the bottom layer, but did not provide evidence on how the stratification was correlated to the DO transport processes at bottom layer, especially for the non-linear relationship between PEA and DO vertical diffusion across the bottom layer (e.g., Hetland and DiMarco, 2008; Bianchi et al., 2010; Fennel et al., 2011, 2013, 2016; Justić and Wang, 2014; Wang and Justić, 2009; Feng et al., 2014; Yu et al., 2015; Laurent et al., 2018).

In the section "4.3 Riverine nutrient reductions", we provided 16 sensitivity experiments (Table 3 in the updated manuscript) which suggested that reductions in riverine nutrients loads would not guarantee a reduction of hypoxia area due to the impacts of interactions among different species (e.g., competition on nutrients, grazing, and predation behaviors) in a complex lower-trophic community (Fig. 11–15 in the updated manuscript). Scenarios of nitrogen reduction even illustrated an increase in hypoxic areas. The results were different from previous studies which were mostly built upon a highly simplified lower-trophic community model (e.g., Fennel et al., 2006, 2011, 2013; Fennel and Laurent, 2018; Justić and Wang, 2014) or statistic models based on the relationship between hypoxia and total nitrogen loads (e.g., Scavia et al., 2013; Obenour et al., 2015; Turner et al., 2012; Laurent and Fennel, 2019). According to our simulations, it is expected that the 3-year mean hypoxic area can reach the hypoxia goal

of 5000 km$^2$ if all nutrients (nitrogen, phosphorus, and silicon) are reduced by nearly 80% which is much more demanded than the recommended percentage indicated by previous model studies.

**Specific comments:**

*L25: The rationale/discussion to support your study is not very convincing and also quite vague, you need to provide better arguments that explain why you conducted this research*
**Response**: We restated the main findings from the previous studies, the knowledge gaps, and the scientific questions we aimed to answer in the Introduction section (see the above responses). We address the focus of this study as 1) the contributions of different factors in hypoxia evolution in different parts of the LaTex shelf; 2) the impacts of different riverine nutrient reduction scenarios on the variability of bottom DO and shelf hypoxia within a complex lower-trophic ecosystem.

*L33-34: this is true only in a dual reduction strategy*
**Response**: Yes. Have addressed it in L29-33 of the updated manuscript.

*L46-48: All of these authors agree that SOC depends on organic matter in the sediment but because sediment OM is unknown they use a relationship between bottom O2, bottom temperature and SOC. They assume oxic respiration, which is why they find a direct relationship between SOC and bottom O2. Justic and Wang (2014) use a sediment tracer that depends on the abundance of deposited OM and is the source for SOC.*
**Response**: We have rewritten this part as followed and updated it in L45–46:

*L52-53: I don't understand this sentence. SOC would be overestimated at the peak of bloom and underestimated during the post-bloom period. This is probably what you meant to say but this is not what I read*
**Response**: In the manuscript, we stated that in L49–L51,"However, the instantaneous parameterization tends to underestimate SOC at the peak of blooms yet overestimate SOC once the blooms started."

*L57-58: This is why the models cited previously used a relationship with T/O2 or instant remineralization. I think what you try to say here is that these earlier parameterizations are not satisfactory and you will try to do better. You should discuss how your SOC implementation will be better than Justic and Wang (2014) because this is the most similar.*
**Response**: Here, we want to address that we did not couple a sediment model for consideration of computation efficiency. The SOC scheme we adopted is based on Fennel's et al., (2006, 2011) scheme but with additional sediment, PON term to correct the misestimations by the instantaneous remineralization parameterization. Indeed, our SOC scheme is somewhat similar to Justic and Wang's (2014). But, in their model, the lower-trophic ecosystem supporting the modeled organic pool is simple. We added the comparison against their study in the manuscript (L98–103).

*L66-68: This is because diatom is the dominant functional group, e.g. Murrell et al. (2014), Lehrter et al (2017). Also, the fact that these models are not a true representation of the reality is not the main point. Here you should point out what these models are doing wrong because of their simple representation of the phytoplankton community and why adding more groups of phytoplankton (and zooplankton) would improve the representation of oxygen sinks and hypoxia on the shelf. More is not always better.*

*Murrell et al: Murrell MC, Beddick DL, Devereux R, Greene RM, Hagy JD, Jarvis BM, Kurtz JC, Lehrter JC, Yates DF (2014) Gulf of Mexico hypoxia research program data report: 2002– 2007. U.S. Environmental Protection Agency, Washington, D.C., EPA/600/R-13/257*
*Lehrter et al: 10.1007/978-3-319-54571-4_8*

**Response**: More is not always better. Yes, we strongly agree to it! In the Introduction section (L78–93), we added a summary of the previous studies and pointed out the knowledge gaps and the problems that a simple ecosystem model could cause.

In the Result and Discussion section, we provided sensitivity experiments to address the importance of the development of a more complex ecosystem in hypoxia simulation. Please refer to the above responses for the brief description and the manuscript for detailed discussion. In the validation section, we added validation for the diatom ratio (section "3.3 Diatom ratios"). We compared our simulated ratio of diatom (Table 1 in the updated manuscript) to two in-field studies by Schaeffer et al. (2012) and Chakraborty and Lohrenz (2015). The ratios provided by the former study are based on the biovolume of different phytoplankton groups, while those provided by the latter study are calculated by chlorophyll *a* attributed to different phytoplankton groups. We added a paragraph for the validation in L330–342.

The high contribution of the diatom group to the shelf phytoplankton community emphasizes that the inclusion of the diatom group in the numerical model is critical to well present the phytoplankton dynamics and the associated hypoxia events in the LaTex Shelf. Our model can reproduce well a diatom-dominated community on the shelf compared to cruise studies.

*L79-80: there are lots of discussions about the factors controlling bottom O2 in the papers you cited above.*
**Response**: We removed this sentence but stated the knowledge gap and the aim of this study in the Introduction section (L79–85 for knowledge gap and L103-106 for updated study objectives)

*L85: you did not discuss silicate above*
**Response**: We added the impacts of silicate on the bottom DO changes in the manuscript. Please see section "4.3 Riverine nutrient reductions" for details.

*L90: what is there to see in the accompanying paper?*
**Response**: The accompanying paper pointed out that statistical models were built based on the daily output by the coupled model in this study and successfully performed promising predictions of hypoxic area in the LaTex Shelf. Predicted hypoxic area showed a high agreement with the ROMS hindcast time series (RMSE=3256 km$^2$, $R^2$=0.7721). When compared to the shelf-wide cruise observations from 2012 to 2020, our prediction model provides a more accurate summer hypoxic area forecast than any existing forecast models with a high $R^2$ (0.9200); a low RMSE (2005 km$^2$); a low scatter index (15 %); and low mean absolute percentage biases for overall (18 %), fair-weather summer (15 %), and windy-summer (18 %) predictions. The accompanying paper has been published recently.

Ou, Y., Li, B., and Xue, Z. G.: Hydrodynamic and Biochemical Impacts on the Development of Hypoxia in the Louisiana–Texas Shelf Part II: Statistical Modeling and Hypoxia Prediction, Biogeosciences Discuss., 2022, 1–23, https://doi.org/10.5194/bg-2022-4, 2022.

*L98: do you have sediment transport in your model (since you are using COAWST) and if so, why not having sediment biogeochemistry as in Moriarty et al (2018)*

**Response**: Although we built our model on the COAWST, we did not include The sediment transport model with a consideration of model efficiency. For a long-term hindcast purpose, we simplified the sedimental processes. We wanted to demonstrate the advances of a simplified treatment of sediment model in a long-term hypoxia hindcast.

*L120: It is obvious why you want to add oxygen but you should discuss the addition of phosphorus, either here or in the introduction*

**Response**: We have added a discussion on hypoxia responses to the nutrient reductions on phosphorus in section "4.3 Riverine nutrient reductions" for details.

*L124: Can you develop? You mean phytoplankton and zooplankton are in N currency, but there is opal, DOP and DON*

**Response**: Nitrate, ammonium, phosphate, and silicate are the limiting nutrient in phytoplankton growth. Opal, DOP, and DON are recycled back to silicate, phosphate, and ammonium during decomposition processes.

*L126: can you provide a reference, a link to the observations? Would it be possible to get a time series of the observations in a supporting figure (for PO4, POP, DOP, silicate since they are new tracers? Also a map of all the gages would be useful since your model domain is quite large.*

**Response**: The river gages and nutrient concentration measurements can be found in the USGS National Water Information System (https://maps.waterdata.usgs.gov/mapper/index.html). We added the locations of all river point sources in Fig. 2a (in the updated manuscript). The time series of river discharges of freshwater, nitrate, phosphate, and silicate from the Mississippi and Atchafalaya rivers was also added in Appendix C (Fig. C1 in the updated manuscript).

*L129: I don't really understand what are your DOP and POP pools here (see next comment)*
*L138-139: These terms seem to be just*
*dDOP/dt = dDON/dt * RPO4N*
*dPOP/dt = dPON/dt * RPO4N*
*can you confirm? in this case you only have PO4 in your model*

**Response:** The inclusion of DOP and POP pools here is to complete the phosphorus cycle in the model. Therefore, the changes of DOP and POP pools due to biochemical processes were set to follow those of DON and PON pools, respectively. And it is also a reason why we only consider a sinking velocity for PON but not for POP. The ratio of riverine DON/DOP and PON/POP may not follow the Redfield ratio. However, the measurements of DOP and POP are usually rare, we thus assumed the Redfield ratio is valid when measurements of the riverine DOP and POP are missed and applied the Redfield ratio for the reconstruction of riverine DOP and POP according to the measurements of DON and PON.

*L161-172: please review this paragraph, the clarity could be improved*

**Response:** We have reviewed this paragraph and have rewritten some sentences. Please see L177–213 in the updated manuscript.

*L163-164: this is the opposite*
**Response:** The estimated SOC is smaller at the bloom peaks but is larger after the peaks. We quoted Fennel's et al., (2013) comments below:
*"An important limitation of this parameterization is that it neglects temporal delays in SOC which occur in nature and would result in smaller SOC at the height of blooms and larger SOC after bloom events in late summer and fall and further downstream from nutrient sources."*

*L164: Note for earlier that the formulations of Hetland and DiMarco (2008) and Lehrter et al (2011) include temperature, which mimics the delay because warmer water occurs after the peak of production*
**Response:** Yes. But neither study considered the organic matter in their SOC schemes. Although the organic matter can be neglected in the SOC scheme if the empirical function is good enough, this is not the way we consider the improvement of the instantaneous remineralization parameterization scheme of SOC. We aimed to provide a scheme that more represents reality. Thus, we mimic the delay by adding a sedimental PON pool in the SOC scheme.

*L180-187: this is a bit difficult to follow, could you make it easier?*
**Response:** We think the most difficult part shall be the derivation of the fraction of denitrification-produced $CO_2$ to the total $CO_2$ production, $x$. Some assumptions were made (1) that only a portion of $NH_4$ provided by the aerobic respiration according to Eq. (R1) is used as the source element in the following nitrification according to Eq. (R2) and (2) that all $NO_3$ produced by nitrification is used as the source element in denitrification according to Eq. (R3). Such a portion of $NO_3$ can be explicitly provided by the linear relationship of denitrification and SOC rate shown in Eq. (5). Indeed, we can set the $x$ as such a portion, but, by setting the $x$ as the fraction of denitrification-produced $CO_2$ to the total $CO_2$ production, we can simplify the calculation and can also have the linear relationship of denitrification and SOC rate applied. Once we have the $x$ determined (or the linear relationship of denitrification and SOC rate applied), we can derive the SOC and sedimental PON consumption as functions of $M$, the production rate of $NH_4$ by aerobic decomposition. We rewrote this part in the manuscript (L197–205).

*L181: How come M is expressed in m-3 since it represents the integrated OM decomposition in the sediment. If you express it in m-2 you can remove the THKbot terms which simplifies the equations*
**Response:** Yes. The production rate of $NH_4$ in sediment can be expressed in the real rate. However, we assumed that the oxygen consumed at the sedimentary layer contributes directly to the DO at the bottom water layer where DO is expressed in mmol m$^{-3}$. The THKbot is thus always needed in the equations. In the discussion, we aimed to compare the SOC rate, another rate of changes due to advection and diffusion, and the bottom DO total rate of changes, thus, we expressed these variables in the same volumetric unit.

*L192: Do you use the same expression for the water column respiration?*
**Response:** The expression for the water column respiration was slightly different since the concentration of PON used there was represented as mmolN m$^{-3}$. In other word, in water column, aerobic decomposition rate is:
$$M = PON \cdot VP2N_0 \cdot \exp(K_{P2N} \cdot TMP)$$

*L199: do you have anaerobic respiration occurring in this case and if not, why?*
**Response:** We did not consider the anaerobic respiration in the water column. Denitrification occurs as the oxygen level is low enough, however, the replenishment of DO in the water column is usually faster than in the sediment. Thus, although the water column could reach below hypoxia condition during some period, the fast replenishment would weaken the denitrification in the water column.

*L211: although this seems fine for the plume region, it seems very short for the entire GoM and may influence you results as the interior GoM is still adjusting during your analysis period. The fact that hypoxia occur>100m later on suggests that this is the case. Also you need to show that your sediment layer reach a seasonal steady state (later on it seems to accumulate throughout the simulation near the Atchafalaya). How was the benthic layer initialized? can you provide a time series of PONsed?*
**Response:** Our PONsed did reach a seasonal steady state starting from 2007 to 2020. We initialized the PONsed as 0 mmol m$^{-2}$ due to the lack of observation. We are confident that our model spin-up period is long enough. In previous ROMS applications to the shelf, the spin-up period was usually as short as a few months even though the model was initialized with the averaged climatological profiles (e.g., Hetland and DiMarco, 2008; Fennel et al., 2011, 2013, 2016). For comparisons, in our model, initial conditions for the physical terms were derived from the Hybrid Coordinate Ocean Model (HYCOM) global analysis products. Initial conditions for concentrations of $NO_3$, $PO_4$, and $Si(OH)_4$ were interpolated from measurements provided by the World Ocean Database (WOD, Boyer et al., 2018). Initial conditions for DO concentration were given by World Ocean Atlas (WOA, Garcia et al., 2018). Other biochemical tracers were initialized as 0.1 mmol m$^{-3}$ due to the lack of observations. With more realistic hydrodynamic initial conditions than previous ROMS applications, we are confident that a 5-month period for spin-ups is long enough. In addition, model validation for nutrient profiles between 2007 and 2009 illustrated our model was capable of reproducing the WOD profiles and was well spun up.

*L226-230: do you do any nudging toward HYCOM or any other climatological product?*
**Response:** We only nudged the salinity and water temperature toward HYCOM with inverse nudging coefficients decreasing from 1 day$^{-1}$ to 1/60 day$^{-1}$ from the open boundary to the interior. The inverse nudging coefficients for the interior (61% of grid cells including land cells) were set identically 1/60 day$^{-1}$.

*L240: can you also show the other rivers for completeness?*
**Response:** We have updated Figure 2a with the locations of all river point sources shown in the updated manuscript.

*L245-246: can you elaborate on this assumption?*
**Response:** The river discharges used as forcing in the model were measured not exactly at the river mouths, but at the lower reach of the main channel usually a few hundred kilometers from the mouths. Along the lowermost of the river channel, river discharges shall be greater than those where the measurements were conducted due to the water supplies from the adjacent watershed and lateral inflow of tributaries. Warner et al. (2005) took into account of such contributions by multiplying the discharges of measurements by 1.4 in their model study on the Hudson River. It is almost impossible to determine this factor for different rivers along the Gulf of Mexico since this factor is related to the distances from where the river gauges are deployed to the river mouths and lateral supplies by watershed and lateral inflow of tributaries. Such conditions vary

greatly among the rivers along the gulf. Factor 1.4 was not quite appropriate to all river systems but shall be considered in river systems like the Mississippi River and the Atchafalaya River systems which cover a large area of the watershed. For these two river systems, we chose 1.4 due to the lack of evidence showing the discharge ratio between the measurement spots and the river mouths.

*L250: it is indeed highly oversaturated. can you provide some context?*
**Response:** The riverine DO concentration was changed to be a constant of 258 mmol m$^{-3}$ assuming that the riverine DO was saturated at 25 °C under 1 atm.

*Figure 2c: The shelfwide surveys were not available prior to 2012? see here:*
*https://coastalscience.noaa.gov/project/integrated-ecosystem-modeling-causes-hypoxia/*
**Response:** We have downloaded the shelf-wide observation prior to 2012 according to the link provided and have expanded the validation accordingly. Plots were therefore updated (Fig. 2, 6 and, 7) in in the updated manuscript

*L296: this is not a good comparison, you should provide histograms for surface data is spring, summer, winter. A 1:1 comparison would also be more meaningful because it would show where the mismatch occur (at low, high concentrations? in the bottom, at the surface?)*
*Figure 3c,f,i: this pair comparison is a bit misleading because you mix all data. Subsurface NO3 should be relatively small, resulting in a good agreement, but there could be significant mismatch at the surface. It is at the surface that a good representation of NO3 is important because that is where primary production occur*
*L301: Same comment here, I don't think this is a proper way to validate the model. Also, what about chlorophyll?*
*L283: I don't understand your choice of model data comparison. Are you binning the profiles by bathymetry? This assumes that the variability occurs from shallow (north) to deep (south) regions whereas the variability should be from upstream (east) to downstream (west). Also looking at Figure 3b it looks like vertical profiles of nitrate are uniform even though high nitrate at the surface (within the plume) is expected. Another issue is that you are mixing all times together. Your observed nutrient dataset is relatively short so you could make a better comparison, surface and bottom maps for example at key periods of the year*

**Response:** We have updated all figures for profile validation. Please refer to the above responses. We did not provide validation for chlorophyll, instead, we provided validation for diatom (section "3.3 Diatom ratio" in the updated manuscript).

*Regarding PO4, high values are mainly found near the bottom, which suggest that the main source of PO4 is from resuspension events rather than from the river. Can you justify these patterns?*
**Response:** There is a lack of discussion of this issue in our manuscript. High PO$_4$ concentration at the near bottom was found both in the simulated and WOD profiles. We did not include the sediment module in our model, therefore, the high PO$_4$ concentration near the bottom should come from the recycling of DOP, POP, and sedimental organic matter (measured as PONsed). A similar phenomenon is found in the Si(OH)$_4$ profiles.

*L315: the data are available, see earlier comments. These data also include nutrients which could be used in complement of WOD*
**Response:** Comparisons of DO profiles between simulation and shelf-wide observations have been extended for the period prior to 2012 (Fig. 6 in the updated manuscript).

*Figure 4c,f,i: I assume that some differences are much larger than 50% because if the model is normoxic and the observation hypoxic (or inversely) the bias could be several hundred percent*

**Response:** Yes. We did have percentage differences greater than 50%. But there are already 72 % of WOD profiles, 66 % of shelf-wide profiles, and 92 % of SEAMAP profiles being misestimated by within 50%. Our model was capable of reproducing the DO profiles well.

*Figure 4h: Aren't SEAMAP cruise occurring in late spring rather than summer?*

**Response:** SEAMAP cruise occurred from June to July. But the measurements were not always carried out within the LaTex Shelf as shown in Figure 2c (red dots) in the updated manuscript.

*L335: I don't know why the model data <10m are not shown in Figure 6, these data should be available to the reader*
*L336: this is not true for the area off the Atchafalaya, observations are available there*

**Response:** The spatial distribution of bottom DO concentration was shown over depth from 6 to 50 m (see Fig. 7 in the updated manuscript).

*L337: 2017 as well. Can you comment on the occurrence of hypoxia around 100m (near the slope). Is that an issue in the model, i.e. does that influence hypoxia on the shelf?*

**Response:** The hypoxia around 100 m is due to the overestimation of the sinking velocity of the organic matter. We have rerun the model with a new parameter (sinking velocity=5 m day$^{-1}$). In this submission, we focused on the validation over the waters where shelf-wide measurements were conducted. According to the new results, we rewrote this part (L389–403).

*L349: why 10m? I agree that you should exclude the Atchafalaya Bay but you should include the coastal area. Also, you should have a more restrictive longitudinal extent because the observations are always <94.5W*

**Response:** The modeled hypoxic area (Fig. 8 in the updated manuscript) was re-estimated over the shelf with depth from 6 to 50 m (Fig. 2b in the updated manuscript).

*L349-353: In some years the model simulates a relatively large hypoxic area in June, sometimes also in May, do you think this is realistic? Are the SEAMAP data showing similar conditions?*
*Also, bottom waters don't always get fully reoxygenated in July-August in years with tropical storms/hurricanes, e.g. 2018-2020. Can you comment?*

**Response:** The simulated hypoxic area is affected by the studied area we chose (here waters with depths from 6 to 50 m). However, no matter for which area is selected, we still cannot provide a comparison for days without observations. We showed the daily time series of the hypoxic area here to emphasize that the hypoxic area can change during a short time period, which suggests that more cruise observations are needed to depict the whole picture of the shelf hypoxia.

We do not have much hypoxia observed by the SEAMAP, since the SEAMAP cruises cover a larger spatial area with less observation in the LaTex Shelf during summer. Bottom waters don't always get fully reoxygenated in July-August. It may result from our relatively large study area and features of hurricanes (e.g., tracks, intensity, and translation speeds). In mid-July 2019, hurricane Barry stroke coastal Louisiana as a category 1 level hurricane. It was a fast-moving and relatively weak hurricane which may not lead to fully reoxygenated bottom waters, especially after a

massive hypoxia event in early July. However, more discussions are needed on hurricane influences which are out of the scope of this study.

*Figure 6: 1) Another way to make this comparison would be to overlay the observations as scatter points over the model maps*
**Response:** We have updated this plot (Fig. 7 in the updated manuscript) accordingly.

*2) hypoxia varies rapidly and it might be better to show a mid-cruise map from the model rather than a ~1 week average*
**Response:** The model bottom DO shown for the comparison purpose was not a ~1-week average, but a ~1-week composite according to the cruise locations on different days. We added a description of the composite plot in the first paragraph of section 3.6 (Line 390–393).

*3) can you show the other years for completeness?*
**Response:** We have added comparisons in other years for completeness.

*L364: you use a mixed format for Results and Discussion but then you do not discuss much your results with respect to the literature*
**Response:** We have updated the section on Results and Discussion. Previous contents were replaced by the following section "4.1 Factors controlling subregion bottom DO variability", "4.2 Stratification and Bottom DO Advection/Diffusion", and "4.3 Riverine nutrient reductions". Please see our response above.

*L375: I don't quite follow this analysis, what does it mean?*
*It looks like there is a long term negative trend in the Atchafalaya plume and offshore. The 2 signals could be problematic: the Atchafalaya plume signal indicate that PONsed accumulates there during the simulation and the offshore signal seems to indicate that there is a drift in offshore subsurface O2 or that the offshore part of the model is still adjusting*
*Note: you don't have resuspension in your model. Can you justify your choice? this feature would be easy to implement and would provide a realistic distribution of SOC over the shelf. This may also prevent the accumulation of PONsed near the Atchafalaya.*
*L408: see earlier comment about the long term trend*
**Response:** We have removed this part. PONsed did not accumulate near the Atchafalaya nearshore regions. Please see the responses above for the model spin-up issue.

*L380-390: can you compare these patterns with the literature?*
**Response:** We have removed this part.

*L385: This is surprising that you find substantial hypoxia in a monthly climatology. This means that 1) hypoxia almost always occur at this location during that month (as shown on the right panels) and/or 2) bottom O2 concentrations are low at these locations, well below the hypoxia threshold.*
**Response:** We have removed this part.

*L450: also vertical diffusion and possibly horizontal advection, as well as SOC*
**Response:** See the updated section "4.1 Factors controlling subregion bottom DO variability" where we compared the contribution of bottom DO advection, bottom DO diffusions, and biochemical terms on the bottom DO changes.

*L456-457: you should compare your results with these. For that you should integrate respiration over the subsurface layer (or lower 4m for instance). You could also discuss your results with respect to other budgets, e.g. Yu el al (2015)*
**Response:** We have added the validation of SOC and ratio of SOC/ overlaying water respiration. We discussed the contribution of different controlling factors, in the section "4.1 Factors controlling subregion bottom DO variability".

*L477-478: this is not obvious*
*Figure 12a,c,e,g: I think the time series in Figure 10 were enough. I don't find these PEA maps very useful*
**Response:** We have removed this part.

*L498-511: this paragraph should go in the Methods section*
**Response:** We have removed this part.

*L513-527: other authors found that water column respiration is not dominant but not negligible either (Lehrter et al, Yu et al), can you comment on that? Is the large dominance of SOC in your model due to the set up of your model, high settling rate for instance?*
**Response:** We have removed this part. The high settling rate of organic matter indeed affects the results. We reduced the sinking velocity to 5 m day$^{-1}$ and provided a reasonable range of SOC and SOC/overlaying water column respiration.

L517: yes, this is where you find persistent hypoxia
**Response:** PONsed did not accumulate near the Atchafalaya nearshore regions. Please see the responses above for the model spin-up issue.

*L522: where is this shown? you speculate here*
**Response:** We have removed this part.

*L524: +10% would be a more conservative value and used for climate projections in the region, e.g. Lehrter et al 2017.*
**Response:** We have removed this part.

*L525: you speculate here*
**Response:** We have removed this part.

*L543: ah yes, that explains the very low water column respiration, see earlier comment.*
*In the Atchafalaya nearshore, PON settles instantly to the bottom and accumulate which explains SOC and hypoxia there. I think this is problematic as your model setup drives your conclusions. This brings up two points: 1) you should validate your choice of high settling rate. For instance if surface nutrients, surface chlorophyll, water column respiration and SOC compare well with the observations/literature then your choice is fine. If not then you may want to recalibrate your model. 2) if PON sinks rapidly to the bottom and water column respiration is not significant, then why do you have 3 functional types of zooplankton?*
*Note: with this type of model setup the predatory zooplankton tend to have a top-down control over primary producers, is this the case in your system and is this why the sinking rate is so high, to escape this control?*
**Response:** By sensitivity experiments, we have changed the sinking velocity to 5 m day$^{-1}$ as we have shown in this document. We introduced a complex ecosystem in this study aiming to test its impacts on hypoxia dynamics.

*L543-555: I don't get the point of this paragraph*
*Figure 15: I don't get the point of this figure*
**Response:** We have removed this part.

*L569-570 (see also earlier comment): Given your fast sinking environment it seems that a single functional group for phytoplankton (diatom) and zooplankton was enough in you study of the LATEX shelf. A more convincing argument for your model choice would be that it is needed for the open ocean part of your domain (if indeed it is)*
**Response:** In this study, we prescribed different phytoplankton and zooplankton functional groups to capture the system sensitivity to different nutrient limitation scenarios, and the benefit of utilizing different zooplankton groups has already been illustrated by Shropshire et al. (2020). Their study highlighted the NEMURO with a complex zooplankton community structure can reproduce well the zooplankton dynamic in the Gulf of Mexico. According to the updated simulation and discussion, the complexity of the ecosystem does affect the hypoxia responses to the different nutrient reduction scenarios suggesting that changes in hypoxia are not straightforward as what has been shown in previous studies.

We have changed our sinking velocity to 5 m day$^{-1}$ which provides a more reasonable range of SOC and SOC/overlaying water respiration.

Reference:
Shropshire, T. A., Morey, S. L., Chassignet, E. P., Bozec, A., Coles, V. J., Landry, M. R., ... & Stukel, M. R. (2020). Quantifying spatiotemporal variability in zooplankton dynamics in the Gulf of Mexico with a physical–biogeochemical model. *Biogeosciences*, *17*(13), 3385-3407.

*L571: P limitation: you did not show that either*
**Response:** We did not include the P limitation discussion in our study since P limitation has already been discussed and deemed to be important to the shelf hypoxia by Laurent and Fennel (2014) and Fennel and Laurent (2018). However, we provided sensitivity tests for a nutrient reduction on P and found that the percentage of reduction in the hypoxic area was close to the finding by Laurent and Fennel (2014). Please see the updated section 4.3 (L571–580).

*L572-573: this was the main novelty of this work. However, model tuning may be necessary to properly reproduce water column respiration (see also earlier comment)*
*L573: you did not show that, see earlier comments*
**Response:** We have validated the SOC and SOC/overlaying water respiration for the updated results with the sinking velocity of 5 m day$^{-1}$.

*L627-628: The model does not include a light attenuation factor for terrigenous material near the river (dependent on salinity for instance)? Light limitation is strong near the Mississippi and Atchafalaya River mouths but this light limitation effect is not included in your model. This lack of light limitation would result in high primary production near the river mouths and less production downstream, thereby influencing the timing and distribution of phytoplankton, respiration and bottom oxygen over the shelf Also (and L638), why is PAR different for small and large phytoplankton? shouldn't it be the same, each functional type having a different sensitivity to light? Looking at your parameter table I see that you are using the same value for both so effectively there is no difference in PAR*
**Response:** The inclusion of light limitation due to riverine sediments shall be an improvement of our model. We have validated the surface nutrients (0–5 m) concentration and found the observations can be reproduced well by our model (see above responses). For the light limitation due to phytoplankton self-shading effects, the

original codes in NEMURO were written with the AttPS and AttPL separated. We did not change this part of the codes but set the AttPS and AttPL the same assuming each functional type has the same sensitivity to light.

*L650: L650: did you mention how these parameters were chosen? were they calibrated to the Gulf of Mexico?*
**Response:** The parameters largely followed the set-ups by Shropshire et al. (2020), Laurent et al. (2012), Fennel et al. (2006), and Fennel et al. (2013). Details can be found in the updated Table B4 in the updated manuscript.

**Minor comments/typos:**

*L1: "impact" is not the right wording*

*L30: shrinking is not the right word, reduction is better. Please rephrase the sentence accordingly*

*L34: "shrinkage" is not the right word, may be "reduction"?*

*L34-35: replace with: "Transient phosphorus limitation on the shelf (Laurent et al 2012; Sylvan et al 2007) was deemed..."*
*Sylvan et al: 10.4319/lo.2007.52.6.2679*

*L35: "with the delayed onset and reduction of the hypoxic area"*

*L39: Conley et al 2009 is not related to the LATEX shelf*

*L56: "coupled"*

*L93: you could mention your main results here.*

*L123: I don't think you need this reference as this formulation is wide spread. However, you could mention that you use the same formulation as for the other nutrients*

*L162: please rephrase, the sentence is not complete*

*L332: I agree but you could mention the underestimation of the hypoxic layer*

*L345-346: you did not introduce Figure 7 yet*

*L377: this makes sense, the STDs are larger in the plume region where hypoxia occurs*

*L381: that seems normal since the hypoxic area is calculated from bottom O2*
*Figure 8e: the DO scale is a bit misleading*
*Why do you show bottom oxygen up to 100m in Figure 6 but then limit the output to 50m in Figures 8-9, 11-12?*

*L400: yes because the extent is a climatology (see comment above)*

*L414/446 (and elsewhere): "trough": minimum may be a better word (elsewhere as well)*
*Figure 9: can you show the results for the coastal area when you show maps?*

*L448: "also water stratification (Figure 10)."*

*L450: be more accurate, here you talk about water column processes*
*Figure 11: Since you don't compare modeled SOC with observations it would be easier to keep the original units*

*L468: Note that the maps show a nearshore/offshore gradient in PEA, following the bathymetry. This is due to the multiplier z in the PEA equation, which increases PEA with increasing bathymetry*

*L471: may be 1 reference is sufficient here?*

*L537: replace "low" by small*

*L568: "the NEMURO model"*

**Response:** We have corrected all the typos and inappropriate usage of words according to the comments.

**Responses to Comments by Referee #2**

*General comments:*
*This manuscript by Ou et al. utilized a coupled physical-biogeochemical model to investigate the controlling factor of bottom hypoxia on the northern Gulf of Mexico and Louisiana-Texas Shelf. The authors added the phosphorus cycle and modified the sediment oxygen consumption module in an existing biogeochemical model NEMURO and coupled it with ROMS model. The coupled model was validated with observational data and then used to implement a 15-year hindcast simulation during 2006-2020. Then the authors explored the spatial variation of hypoxia development in the study area and found sediment oxygen consumption (SOC) and water column stratification are main factors to control the bottom oxygen in nearshore and offshore area respectively. Their model results also indicated separate hypoxia development schemes on the west and east Louisiana-Texas Shelf. Coastal deoxygenation is one of the most prominent environmental issues with important implications for marine ecosystem services. Although this paper made efforts to adopt a more sophisticated biogeochemical model with added phosphorus cycle and improved sediment oxygen consumption module, making contributions to investigate the spatial differences of dominant processes on hypoxia, it lacks original and novel aspects to explore the well-studied topic in this region, as well as comprehensive comparison with previous modeling study on the model performance, simulation results and conclusions, and address the question that how this new model stands out. The manuscript missed an advanced understanding and deep insight on the research topic of coastal hypoxia in a well-organized discussion section, thus this paper is a little thin on content. Although I see the value of this work, I perceive that the publication is premature at this time.*

**Response**: We thank Reviewer #2's comments. Compared with existing modeling efforts, our model, for the first time, included a silicate cycle as well as multiple plankton functional groups in the modified biogeochemical model, the importance of which has already been addressed in previous studies yet not included in hypoxia modeling efforts. The extensive model validation against nutrient and dissolved oxygen profiles confirmed the robustness of our model. In addition, another purpose of this model study is to provide the needed numerical solution for the development of a novel statistical model presented in paper Part 2.

We agree that our findings presented in the first submission need some improvement. In this round, we performed a comprehensive revision of the manuscript. The main changes in the manuscript are listed below.

Changes in model set-ups:
1. We performed a series of sensitivity runs to evaluate the model's robustness regarding different parameterizations of the sinking velocity of the organic matter (i.e., PON and Opal). And we conclude that a sinking velocity of 5 m/day is a reasonable prescription and updated relevant model results.
2. The riverine DO concentration was changed to be a constant of 258 mmol m$^{-3}$, assuming that the riverine DO was saturated at 25 °C under 1 atm.

Content removed:
1. The model validation for hypoxic thickness was removed since we focused more on the bottom DO dynamic.

2. We rewrote the entire "Result and Discussion" section. Both reviewers pointed out that the manuscript lacked originality. In this revision, we focus on 1) the contributions of different biogeochemical and hydrodynamic terms on bottom DO variability in different subregions, 2) the impacts of diatom and the complexity of lower-trophic community on the hypoxia dynamics, and 3) bottom DO's responses to the reduction of riverine nutrient loads with different reduction combination (percentage and nutrient type).

Content added:
1. The model validations of diatom ratios, sediment oxygen consumption (SOC), and ratios of SOC/ overlaying water respiration were added following the reviewers' suggestions.
2. Results and discussion on the factors to bottom DO variability was added (see section "4.1 Factors controlling subregion bottom DO variability" and "4.2 Stratification and Bottom DO Advection/Diffusion"). All terms that directly contribute to bottom DO changes were calculated and evaluated, which include horizontal advection, vertical advection, horizontal diffusion, vertical diffusion, the local rate of change (bottom DO), SOC, and DO changes due to water column biochemistry at the bottom layer. The summation of these terms contributes to the total changes in the bottom DO. A comparison of their contributions was given in different subregions of the LaTex shelf. We found that the most prevailing factors to the bottom DO changes are the two advection terms, vertical diffusion term and SOC. The former three terms were associated with the changes in water column stratification. The strong linear correlations between PEA and the advection terms suggest that increased water stability in summer leads to fewer DO exchanges from advection processes. Nevertheless, the relationship between PEA and vertical diffusion of DO across the bottom layer appears to be non-linear.
3. Sensitivity experiments for riverine nutrient reduction strategy were added (see section "4.3 Riverine nutrient reductions"). We found the responses of summer hypoxia to the changing nutrient loads are not linear due to the impacts of the complexity of the lower-trophic community. Nutrient reductions do not always guarantee a reduction in summer hypoxic area, instead, due to the interactions among different plankton groups (e.g., competition on nutrients, grazing, and predation behaviors), the hypoxic area could even increase under some nutrient reduction conditions. The most effective strategy is to reduce the nitrogen, phosphorus, and silicon loads simultaneously. A triple riverine nutrient reduction of 80% can help to fulfill the hypoxia reduction goal of 5000 km$^2$.

*Major comments:*
*The hypoxia at the northern Gulf of Mexico has been well studied since the 1990s with*
*increasing model studies in recent years. It ranged from a simple oxygen respiration model (Hetland&DiMarco, 2008) to a sophisticated coupled biogeochemical model (Laurent et al. 2012; Fennel et al. 2013). Including this study, they all generated similar conclusions that SOC is the controlling factor for hypoxia. In this sense, the improvement of complexity in the biogeochemical model does not make much sense. Also, the authors mentioned the additional work done on the NEMURO-based model filled gaps in phosphorus cycling and improved SOC representation. It's better to prove*

*the advancement of the new model by validating with important variables, such as DO,*
*Chla, PO4, NO3, with other model simulation studies.*
**Response**: In the validation part, we added validation of SOC (L349–373, Fig. 4 in the updated manuscript), the ratio of SOC/overlaying water respiration (Fig. 5 in the updated manuscript), and the ratio of diatom/total phytoplankton against measurements (L329–348, Table 1).
We updated our results and focus on the direct impacts of biochemistry and hydrodynamics on the bottom DO variability (section "4.1 Factors controlling subregion bottom DO variability" and "4.2 Stratification and Bottom DO Advection/Diffusion" in the updated manuscript), the impacts of diatom and the complexity of lower-trophic community on the hypoxia dynamics including the impacts on the responses of hypoxia changes to the decreasing riverine nutrient loads (section "4.3 Riverine nutrient reductions")

*The oxygen balance analysis is confusing and questionable. Although SOC is the dominant process in the bottom hypoxia generation (Yu et al. 2015), water column respiration (WCR) should not be orders of magnitude smaller than SOC, especially in the whole water column, as shown in Figure 15 and L455-456. Observational studies still showed varying evidence on SOC contribution (Murrell&Lehrter, 2011; Quiñones-Rivera, et al. 2010). More importantly, the reviewer has a sense that the authors did not understand and explain the oxygen dynamics well (Figure 10 and 15, section 4.2). What is oxygen balance in the text? Based on L450-452, it should be water column respiration plus phytoplankton photosynthesis. This is a very confusing term and the physical transport of oxygen was totally missing. A lot of oxygen studies utilized standard oxygen budget analysis to separate dynamic terms in oxygen change (Li et al. 2014; Scully 2013; Yu et al. 2015). Please refer to those studies on the analysis and consider recalculating/rewriting this part.*
**Response**: We agree with Reviewer#2 that the impact of WCR should not be neglected. In the revision we validated the SOC and ratio of SOC/overlaying water respiration in the section "3.4 SOC and overlaying water respiration". For the factors controlling bottom DO variability (sections 4.1 and 4.2 in the updated manuscript), contributions of five hydrodynamic-related terms and two biochemical-related terms were discussed.

*Although this study employed sophisticated machine learning techniques to determine the controlling mechanisms on hypoxia in different regions. It could be actually achieved by oxygen budget analysis, with much clear representation in physical terms (advection and diffusion), rather than relying on stratification indicators. In addition, compared to the manipulating force on DO variability on a seasonal scale, the interannual variability is more of interest and worthy to look into.*
**Response**: We removed the machine learning part and rewrote the sections which discuss the controlling factors of hypoxia (please see sections 4.1 and 4.2 in the updated manuscript).

*The manuscript missed a comprehensive discussion section of advanced understanding*

*of the study topic in-depth and in breadth. The overview of previous observational and*
*model studies in this region, comparison with the current study, what are the agreements and differences, what are the causes, what are the defective aspects of this*
*study, etc. are all important points to include. Expanding the implication to the global context is also valuable to discuss.*

**Response:** In the updated section "4 Results and discussion", we moved our focus to 1) the contributions of different factors in hypoxia evolution in different parts of the LaTex shelf; 2) the impacts of different riverine nutrient reduction scenarios on the variability of bottom DO and shelf hypoxia within a complex lower-trophic ecosystem.

In the section "4.1 Factors controlling subregion bottom DO variability", our results indicated that variability of bottom DO on the LaTex shelf was mostly controlled by four processes: horizontal advection, vertical advection, vertical diffusion, and SOC. Their impacts on bottom DO variability varied over different subregions over the year. Such results have not yet been pointed out by previous studies. On the other hand, although the importance of DO advection and SOC on bottom DO balance was also documented by Ruiz Xomchuk et al. (2021), vertical diffusion was proposed as a minor contributor in their study. Such a disagreement could result from the water layers investigated. Vertical diffusion of DO across the layer 10 m above the bottom was discussed in Ruiz Xomchuk et al. (2021), while here we estimated vertical diffusion of DO across the bottom layer.

In the section "4.2 Stratification and Bottom DO Advection/Diffusion", significant negative correlations were found between the PEA and the two absolute advection terms of bottom DO, while bottom DO flux due to vertical diffusion was found positively and moderately correlated to the PEA (Fig. 10). Previous studies pointed out that water stratification affect the DO replenishment at the bottom layer, but did not provided evidence on how the stratification was correlated to the DO transport processes at bottom layer especially for the non-linear relationship between PEA and DO vertical diffusion across the bottom layer (e.g., Hetland and DiMarco, 2008; Bianchi et al., 2010;  Fennel et al., 2011, 2013, 2016; Justić and Wang, 2014; Wang and Justić, 2009; Feng et al., 2014; Yu et al., 2015; Laurent et al., 2018).

In the section "4.3 Riverine nutrient reductions", we provided 16 sensitivity experiments (Table 3, also see the above responses) which suggested that reductions on riverine nutrients loads would not guarantee a reduction of hypoxia area due to the impacts of interactions among different species (e.g., competition on nutrients, grazing, and predation behaviors) in a complex lower-trophic community. Scenarios of the nitrogen reduction even illustrated an increase in hypoxic area. The results were opposite to previous coupled numerical studies which were built upon a highly simplified lower-trophic community model (e.g., Fennel et al., 2006, 2011, 2013; Fennel and Laurent, 2018; Justić and Wang, 2014) and statistic models which were developed based on the relationship between hypoxia and total nitrogen loads (e.g., Scavia et al., 2013; Obenour et al., 2015; Turner et al., 2012; Laurent and Fennel, 2019). According to our simulations, it is expected that the 3-year mean hypoxic area can reach the hypoxia goal of 5000 $km^2$ if all nutrients (nitrogen, phosphorus, and silicon) are reduced by nearly 80% which is much more demanded than the recommended percentage indicated by previous model studies.

*Detailed comments:*
*Method*

*L105-106: are the new features of this biogeochemical model suitable for NGoM?*
**Response:** For the zooplankton modeling configurations, we followed Shropshire et al. (2020)'s works focusing on the zooplankton dynamics in the Gulf of Mexico, which use a similar model setup. According to that study, the zooplankton community could substantially affect the primary production in the study area.
Reference:
Shropshire, T. A., Morey, S. L., Chassignet, E. P., Bozec, A., Coles, V. J., Landry, M. R., ... & Stukel, M. R. (2020). Quantifying spatiotemporal variability in zooplankton dynamics in the Gulf of Mexico with a physical–biogeochemical model. Biogeosciences, 17(13), 3385-3407.

*L108: what is PL? should it be LP (large phytoplankton)?*
**Response**: We have corrected all typos related to abbreviations of plankton group, that is small phytoplankton (PS), large phytoplankton (PL), small zooplankton (ZS), large zooplankton (ZL), and predatory zooplankton (ZP).

*L120-122: no reactive, labile and refractory category in organic matter pool? In other words, is a single reaction rate enough?*
**Response**: We included a burial PON pool in the conceptual sedimental layer. PON settles down at the conceptual sedimental layer fuels the PONsed pool, which is a reactive labile organic sediment pool. After a portion of PONsed is decomposed during aerobic and anaerobic processes in sediment (see the SOC scheme), a certain portion of PONsed is burial and fueling the PONburial pool, which will be removed from the system.

*L156: What are ExcZS, ExcZL and ExcZP represented (I could not find those in the Appendix, and guess they should be zooplankton excretion rate to NH4?)? Why not include the zooplankton respiration term?*
**Response**: ExcZS, ExcZL and ExcZP are typos in the equation. They should be ExcZSn, ExcZLn, and ExcZPn, respectively as shown in Table B2. In our model, we combined zooplankton excretion and respiration. Thus, during excretion, zooplankton consumes oxygen.

*L158-159: How did oxygen inhibition on nitrification and aerobic decomposition rates were calculated? Using Michaelis–Menten formula?*
**Response**: The oxygen inhibition (Fennel et al., 2006; 2013) is considered as the maximum of 0 and an oxygen-dependent unitless term. It uses Michaelis–Menten formula. The inhibition term ($\hat{r}$) is described in A2 and A3 with the related parameter description in Table B4.

$$\hat{r} = max\left[\frac{max(0, Oxyg - Oxyg_{th})}{K_{Oxyg} + Oxyg - Oxyg_{th}}, 0\right]$$

Where

| Parameter | Description | Units | Values |
|---|---|---|---|
| $K_{Oxyg}$ | Oxygen concentration at which inhibition of nitrification and aerobic respiration are half-saturated | $mmolO_2\ m^{-3}$ | 3.0 |

| $Oxyg_{th}$ | Oxygen concentration threshold below which no aerobic respiration or nitrification occurs | mmolO$_2$ m$^{-3}$ | 6.0 |

Reference:

Fennel, K., Wilkin, J., Levin, J., Moisan, J., O'Reilly, J., and Haidvogel, D.: Nitrogen cycling in the Middle Atlantic Bight: Results from a three-dimensional model and implications for the North Atlantic nitrogen budget, Global Biogeochem. Cycles, 20, 1–14, https://doi.org/10.1029/2005GB002456, 2006.

Fennel, K., Hu, J., Laurent, A., Marta-Almeida, M., and Hetland, R.: Sensitivity of hypoxia predictions for the northern Gulf of Mexico to sediment oxygen consumption and model nesting, J. Geophys. Res. Ocean., 118, 990–1002, https://doi.org/10.1002/jgrc.20077, 2013.

*L164-166: how was the portion of sinking PON buried (PONburial) determined? How the initial sediment PON pool was calculated? Is there also an anaerobic layer? Is there any exchange between PONburial and PONsed?*

**Response**: The burial faction is determined using the scheme embedded in the original NEMURO model, where the burial faction is a function of the vertical flux of particulate organic matter. As the organic matter is buried, it will leave the system without returning to PONsed.

The PONsed is initialized as 0 due to a lack of available data to initialize the model. Our model does not include a sediment module, thus, the sedimentary PON pool is in an imaginary or conceptual sediment layer. In this layer, aerobic decomposition, nitrification, and denitrification occur simultaneously following the linear relationship between denitrification rate and total oxygen consumption rate (Eq. (5)). So, there is no specified anaerobic layer.

*L193: the description of THKbot is confusing. Is it the thickness of overlying water, or sediment layer?*

**Response**: THKbot is the thickness of overlying water or the thickness of the bottom layer of the ocean model. In our model, we do not separate overlaying water and bottom water. We consider THKbot since we assumed that oxygen consumption at the conceptual sediment layer directly contributes to decreases in oxygen concentration at the bottom water layer.

*L195: SOC/THKbot is basically the oxygen consumption rate in the sediment. Why not integrate SOC in the hypoxic area and get an overall integrated SOC? Any observational data validation on the newly added sediment and phosphorus module? In addition to the oxygen concentration validation?*

**Response**: We assumed that oxygen consumption at the conceptual sediment layer directly contributes to decreases in oxygen concentration (only) at the bottom water layer. It also implies that the oxygen consumed in the sediment is from the bottom water layer. For further comparison with other terms of DO changes (e.g., advection and diffusion) expressed in a volumetric unit, we transformed the areal SOC rate (mmolO$_2$ m$^{-2}$ day$^{-1}$) to a volumetric rate by dividing the THKbot.

For the newly added phosphorus module, we have added validation for PO$_4$ profiles against WOD observation in section "3.2 Nutrients concentration profiles". Comparisons showed that our model was capable of reproducing the measured PO$_4$.

For the validation of SOC, we added the section "3.4 SOC and overlaying water respiration" in the manuscript.

*L211: is 5 months enough for spin-up in this area? What is the initial condition (cold start or hot start)?*

**Response**: We are confident that our model spin-up period is long enough. In previous ROMS applications to the shelf, the spin-up period was usually as short as a few months even though the model was initialized with the averaged climatological profiles (e.g., Hetland and DiMarco, 2008; Fennel et al., 2011, 2013, 2016). For comparisons, in our model, initial conditions for the physical terms were derived from the Hybrid Coordinate Ocean Model (HYCOM) global analysis products. Initial conditions for concentrations of $NO_3$, $PO_4$, and $Si(OH)_4$ were interpolated from measurements provided by the World Ocean Database (WOD, Boyer et al., 2018). Initial conditions for DO concentration were given by World Ocean Atlas (WOA, Garcia et al., 2018). Other biochemical tracers were initialized as 0.1 mmol m$^{-3}$ due to the lack of observations. With more realistic hydrodynamic initial conditions than previous ROMS applications, we are confident that a 5-month period for spin-ups is long enough. In addition, model validation for nutrient profiles between 2007 and 2009 illustrated our model was capable of reproducing the WOD profiles and was well spun up.

*Biogeochemical model validations*

*The entire validation is qualitative rather than quantitative. Need statistic metrics to assess the overall model performance, i.e. taylor and target diagram.*

**Response**: We replotted all the figures for model validation of nutrient and oxygen profiles (Fig. 3 and Fig. 6 in the updated manuscript). Probability histograms of nutrient differences between modeled and observed results were given for the upper 50 meters and upper 5 meters, respectively. Probability histograms of DO differences were given for the upper 50 meters.

For the validation of diatom ratio, we compared the mean±1SD between simulation and observation (Table 1 in the updated manuscript).

For the validation of SOC and the overlaying water respiration, we compared the mean SOC and the mean SOC/overlaying water respiration between simulations and observations using bar plots (Fig. 4 and 5 in the updated manuscript).

For the validation of time series of hypoxic area, we provided the averaged modeled and measured value in the time series plot (Fig. 8) and also the 5-year running $R^2$ in Table 2.

*Figure 3: which cross-section was compared in Figure 2b? The difference histogram in (c)(f)(i) is vertically averaged or bottom value?*

**Response**: We have replotted Fig. 3. Please see the above responses.

*L287-288: both NO3 and PO4 were overestimated*

**Response**: Yes. But the biases are slight when compared to the riverine supplies.

*L295-296: this statement is a bit questionable that the high riverine nutrient concentration may not be the cause for the model-observation bias. Because the high concentration of PO4 and Si(OH)4 is at the bottom which indicates that it is nutrient regeneration, rather than the allochthonous source.*

*What are the causes for the hot points (with bottom high nutrient concentration) of PO4*

*and Si(OH)4?*

**Response**: We did not attribute the nutrient bias to the high riverine nutrient concentration, instead, we wanted to emphasize that such bias was acceptable regarding to the strong influences of high riverine nutrient loads.

We did not include the sediment module in our model, therefore, the high $PO_4$ and $Si(OH)_4$ concentration in the bottom layers could be a result of the recycled DOP, POP, sedimental organic matter (measured as PONsed), and opal. Specifically, the high peak of $Si(OH)_4$ concentration occurred at around 35 m depth was consistent with biogenic silica remineralization at lower water columns (Baronas et al., 2016).

*L303-304: model overestimates DO while also overestimating the recycled nutrient concentration. Usually, it is the opposite case since nutrient remineralization is associated with oxygen consumption. Any explanations?*
**Response**: Based on the sensitivity experiments, nutrient reductions do not always lead to increases in bottom DO due to the interaction among different species (e.g., competition on nutrients, grazing, and predation behaviors) in a complex lower-trophic community. In other words, the responses of DO changes to nutrient changes are not straightforward as what was suggested by previous numerical studies relying on highly simplified ecosystem models. Therefore, although the model overestimated the recycled nutrient concentration (i.e., $NO_3$ and $PO_4$) we would expect to have an overestimation in DO.

*L331-332: in section 3.4 model validation of oxygen, the result suggested that the model overestimated DO and hypoxia was more frequent in observed WOD profiles. Why here the modeled hypoxia thickness (<=4m) is greater than observed profiles?*
**Response**: We have removed the validation for hypoxic thickness since we did not provide a discussion on hypoxic volume but focused on the DO at the bottom water layer.

*L336-337: the model showed more offshore extension of hypoxia than observation. Any possible causes?*
**Response**: Using the updated model set-ups (sinking velocity of organic matter was changed to 5 m day$^{-1}$ from 15 m day$^{-1}$), such inconsistency seldom occurred. There is no model that can reproduce exactly the extension of the hypoxia area documented by ship-based observations. The model simulated more offshore extension (e.g., 2011) of hypoxia may result from the overestimated offshore transport of water and materiel due to a relatively coarse spatial model resolution (~5 km), parameterization of advection and diffusion processes, and the coarse spatial resolutions of atmospheric forcings. Compared to the existing modeling studies, we are confident that our model performed is pretty robust.

*L346-347: the hypoxia area was separated around 92.5W instead of 91W shown in the model simulation? This may reveal a certain defect in the dynamics of model simulation in oxygen.*
**Response**: Based on the updated simulations, we would note that the western and eastern hypoxic water were not always merged but were separated at around 91 °W (e.g., 2007, 2010, 2012, 2014, 2017, and 2018; Fig. 7 in the updated manuscript).

*L349: why not include hypoxia area in the water depth<10m?*

**Response**: We have replotted the spatial distribution of bottom DO (see Fig. 7 in the updated manuscript) covering depths from 6 to 50 m.

*L346: the order of figure citation is a bit messy; the figure should be numbered according to the order of citation, not the other way around. For example, the order of Figure 10 is not optimal for reference.*
**Response**: We have renumbered the figures according to the order how the results were shown.
*Figure 7: please adjust the x-axis as the other years for better comparison.*
**Response**: We have adjusted the x-axis according to the comment (see Fig. 8 in the updated manuscript).

*L351-352: this means no apparent bias of model simulation in the hypoxia area. How is this model performance compared to other model studies in this region?*
**Response**: Previous coupled models did not provide validation of hypoxic areas with a time record as long as ours. So, we did not add a comparison of hypoxic area time series against previous coupled models.

*Results*
*L432: use biogeochemical instead of biochemical throughout the manuscript*
**Response**: We did not use the term biogeochemical when discussing only the biological or chemical processes.

*L433: denitrification process should not consume oxygen*
**Response**: The "4 Result and Discussion" has been updated. The denitrification does not consume oxygen (Eq. (R3)). In our model, SOC was estimated considering the aerobic mineralization, nitrification, and denitrification in the sediment as a 1-step process with the linear assumption applied.

*L453-454: what does it mean by saying contributions are limited? I suggest showing the contribution in percentage. What is DO balance and how it was calculated?*
*L450-457: the entire description and calculation is misleading and confusing. Generally, all DO budget terms including physical terms, photosynthesis, SOC and WCR should be calculated. The summary of budget terms should match the change of DO. I think the authors did not understand and explain the oxygen dynamics well. Please refer to the model studies with oxygen budget analysis and rewrite this part.*
**Response**: This part has been removed and updated. In section "4.1 Factors controlling subregion bottom DO variability", we compared the contributions of bottom DO advection (horizontal and vertical), bottom DO diffusion ((horizontal and vertical), SOC, and water column biogeochemistry on the bottom DO variability. Our results indicated that the variability of bottom DO on the LaTex shelf was mostly controlled by four processes: horizontal advection, vertical advection, vertical diffusion, and SOC (Fig. 9b in the updated manuscript). Their impacts on bottom DO variability varied from in different subregions throughout a year.

*L455-456: does the biochemical process in this sentence represent water column respiration?*
**Response**: We have removed this part. But in section "4.1 Factors controlling subregion bottom DO variability", we calculated the changes in bottom DO due to water column biogeochemistry compiled processes of phytoplankton photosynthesis,

phytoplankton respiration, zooplankton metabolism, aerobic decomposition of PON and DON, and nitrification.

*L475-476: please indicate the change of PEA quantitively (e.g. in percentage).*
**Response**: We have removed this part. Instead, the influences of water stratification on bottom DO variability were addressed in section "4.2 Stratification and Bottom DO Advection/Diffusion" by investigating the relationships between PEA and bottom DO transports. We found strong linear correlations between PEA and the advection terms suggesting that increased water stability in summer lead to less DO exchanges from advection processes. Nevertheless, the relationship between PEA and vertical diffusion of DO across the bottom layer appears to be non-linear.

*L480-482: west-Mississippi nearshore did not show a change of current direction from*
*westward to southward, rather it pointed to northward.*
**Response**: We have removed this part.

*L498-499: please justify the choice of GBMs method.*
**Response**: We have removed this part.

*L498-511: Move detailed description of GBMs into method section.*
**Response**: We have removed this part.

*Figure 13(a) and Figure 10(a) conflicted in PEA contribution in nearshore West Mississippi?*
**Response**: We have removed this part.

*L540: what does this statement mean? Please clarify it.*
**Response**: We have removed this part.

*L543-544: how does it compare to other model studies? Is this parameterization better or not? Please add a more in-depth discussion here.*
**Response**: Concerning the possibly improper sinking velocity applied in the model, we added two sensitivity experiments with different sinking velocities: 1 m/day and 5 m/day. A comparison of SOC against the observation by McCarthy et al., (2013) (Fig. RC2-1) suggests that the experiment with a sinking velocity of 5 m/day provides the best estimates (Fig. RC2-2). The root-mean-squared errors (RMSEs) are 567 $\mu$mol m$^{-2}$ h$^{-1}$, 713 $\mu$mol m$^{-2}$ h$^{-1}$, and 452 $\mu$mol m$^{-2}$ h$^{-1}$ for sensitivity tests with a sinking velocity of 15 m/day, 1 m/day, and 5 m/day, respectively. The model results (tests with sinking velocity = 5 m/day) generally overestimate the SOC at site F5 and C6 except for January 2009 and May 2010 at site C6, but underestimate SOC at site B7 and MRM. However, the simulated and observed SOC are generally in the same order of magnitude. Times series also reveals that the magnitude of simulated SOC simulated with a sinking velocity of 5 m/day is generally within the measured range (Fig. RC2-3) over the entire year. The magnitude of simulated SOC by tests with a sinking velocity of 15 m/day is out of the upper measured bound especially in summers. Modeled SOC by the test with a sinking velocity of 1 m/day is always below the lower measured bound.

We further compared the simulated ratio of SOC/overlaying water respiration against measurements (Fig. RC2-4). The test run with sinking velocity = 5 m/day

provides the best-simulated ratio with a low averaged RMSE of 4.23 over site F5, C6, and B7 compared with 4.58 (sinking velocity = 15 m/day) and 6.51 (sinking velocity = 1 m/day) derived by the other two tests. At site MRM, both the two tests with faster (5m and 15m/day) sinking velocity highly overestimate the ratio in August 2009. We ascribe such bias to the relative course bathymetry near the river mouths. Point sources are applied in the model for diverting momentum and concentration tracers from the river to the computational grid cells. The scheme can lead to an overshot of river water at the near-mouth grid cells, which, may further result in shorter residence time of organic matter and plankton.

We also compare the hypoxic area simulated by the three different experiments. The simulation with 5m and 15m/day settling velocity show a similar hypoxia zone, while the using 1m/day is generally greater than the former two (Fig. RC2-5). Both the former two estimations (5m and 15m/day) can reproduce the magnitude and interannual variability of the measured hypoxic area. Compared to the shelf-wide observations, the simulated bottom hypoxic extent (Fig. RC2-6– RC2-8) by the experiment with a sinking velocity of 5m/day produces less bias among the three experiments. Based on these results, we changed the sinking velocity of PON from 15 m day$^{-1}$ to 5 m day$^{-1}$ in the baseline simulations.

[Figure]

Fig. RC2-1 Map showing the location of sampling site in the northern Gulf of Mexico (McCarthy et al., 2013).

[Figure]

Fig. RC2-2 Comparison of observed SOC (in $\mu$mol m$^{-2}$ h$^{-1}$) by McCarthy et al., (2013) and simulated SOC by different sensitivity tests.

[Figure]

Fig. RC2-3 Daily average of simulated SOC by different sensitivity tests.

[Figure]

Fig. RC2-4 Comparison of observed ratio of SOC/overlaying water by McCarthy et al., (2013) and simulated ratio by different sensitivity tests.

[Figure]

Fig. RC2-5 Comparison of observed and simulated hypoxic area. Note that the horizontal red thick bars denote the shelf-wide cruise measurements.

[Figure]

Fig. RC2-6 Evolution of simulated bottom water dissolved oxygen concentration (unit mg l$^{-1}$) by the sensitivity experiment with a sinking velocity of 15 m day$^{-1}$. The black filled circles and open circles indicate the hypoxic site and non-hypoxic site, respectively, according to the Shelf-wide cruise observations. The grey curves denote bathymetry of 5, 10, 20, and 50 m.

[Figure]

Fig. RC2-7 Same as Fig. RC2-6 but for the sensitivity experiment with a sinking velocity of 1 m day$^{-1}$.

[Figure]

Fig. RC2-8 Same as Fig. RC2-6 but for the sensitivity experiment with a sinking velocity of 5 m day$^{-1}$.

*L544-548: Figure 10 and Figure 15 looks very similar which is questionable to me. The*
*previous studies suggested that sediment oxygen consumption dominated the hypoxia in the study area, while the water column respiration was still notable.*
**Response**: We have removed this part.

---

## Author Response (AR2)

Responses to Comments by Referee #1

General responses:

   We sincerely thank the reviewer who provided us with such detailed comments and suggestions and also his/her patience with our prolonged responses.

   We reorganized our thoughts for this study and would like to focus on 1) the role of silicate cycling in the biogeochemical processes and bottom hypoxia development in the Louisiana-Texas shelf and 2) the impacts of a complex plankton community on the dissolved oxygen (DO) dynamics. Thus, we removed the original sections 4.1 and 4.2, which discussed the impacts of physics. Instead, we focused on the biogeochemical processes in DO dynamics.

   We also separated the Results and Discussion sections. In the Results sections, we tried to answer the following questions. 1) What are the limited nutrients for PS and PL, respectively? 2) What is the dominant plankton group on the shelf? 3) How do different plankton groups contribute to the source and sink of DO in water columns and sediments? In the Discussion section, we reran all sensitivity tests and expanded the 3-year (2018–2020) simulations to a 9-year (2012–2020) one. We aimed to discuss the responses of biomass and DO to the reduced riverine nutrient supplies. Please find our detailed point-to-point as follows,

*General comments:*
*Overall: This revision of Ou and Xue manuscript attempt to address some of the issues the original version that were pointed out by the reviewers. Indeed the authors provide more context in their introduction to justify their study as well as more validation that tend to demonstrate that the model agrees well with observations. The analysis is also better displayed. The use of Si limitation is also interesting.*

*That said, most of the analyses still repeat previous work (oxygen budget, effect of stratification, importance of SOC) and indeed the conclusions are similar. Apart from the model itself, Section 4.3. (nutrient load experiments including Si) is arguably the only novel part of the paper; I don't think Si has been included in such a way in nutrient management strategies/studies, although the potential for Si limitation has been discussed in various studies. That raises several questions that are critical to this investigation but that have not been or barely discussed.*

Responses: Sections 4.1 and 4.2 have been removed. We now focus on the contribution of different nutrients (N, P, Si) and the effects of a complex plankton community to hypoxia development.

*1) why was Si not included in previous models? the authors mention briefly in the Discussion section that previous work assume that Si is plenty and therefore not limiting. This is true but the authors do not provide strong evidence against these assumptions. Previous assumptions were based on observations. Also observations indicate that N (TN or NO3) is the main predictor for the mid-summer hypoxic area, which suggest that variations in N load control hypoxia.*

Responses: Previous statistical models suggested that there is a strong correlation between the reduction in riverine nitrogen and reductions in hypoxic areas. However, there have been several observational studies indicating the importance of silicate in the study area. In this study we also found significant correlations among riverine nitrogen loads, phosphorus loads, and silicon loads (see figure below and table in Appendix C). Strong correlations can also be found between

hypoxic area (or bottom DO) and phosphorus and silicon loads. In this sense, previous statistical studies might overestimate the effects of nitrogen. We added related discussions to the revised Discussion section.

[Figure]

**Figure C1. Daily time series (2007–2020) of river discharges of freshwater, nitrate, phosphate, and silicate from the Mississippi and Atchafalaya rivers.**

**Table C1. A correlation matrix of daily inorganic nutrient loads by the Mississippi River and the Atchafalaya River from 2007 to 2020. Correlation coefficients shown are all significant ($p<0.001$).**

|  | Mississippi nitrate+nitrite | Atchafalaya nitrate+nitrite | Mississippi phosphate | Atchafalaya phosphate | Mississippi silicate | Atchafalaya silicate |
|---|---|---|---|---|---|---|
| Mississippi nitrate+nitrite | 1 |  |  |  |  |  |
| Atchafalaya nitrate+nitrite | 0.9123 | 1 |  |  |  |  |
| Mississippi phosphate | 0.8328 | 0.7577 | 1 |  |  |  |
| Atchafalaya phosphate | 0.7517 | 0.7913 | 0.9155 | 1 |  |  |
| Mississippi silicate | 0.8583 | 0.7795 | 0.8759 | 0.7942 | 1 |  |
| Atchafalaya silicate | 0.7938 | 0.7956 | 0.8131 | 0.8148 | 0.9520 | 1 |

*2) Is this useful to include Si in an experiment to assess the effect of nutrient reduction strategies? can you provide examples of how this can be implemented? It might be more useful to focus on N and P reductions as in other studies (assuming constant Si, or extrapolating on future Si) and look in detail at the effects on the ecosystem. Mentioning that there are nonlinear effects to nutrient reductions is not enough.*

Responses: In the Discussion section, we carried out and analyzed the results of six sensitivity tests to cover the effects of different nutrient reduction strategies on hypoxia reduction. Our model shows that N is not always the only limited nutrient. P and Si limitations can be dominant in waters that are deeper than 20m. Please see the Discussion section for more details.

*3) the effect of nutrient management on phytoplankton community structure, trophic interactions, and ultimately organic matter deposition and hypoxia is very interesting. This should be the main focus of this manuscript instead of redoing previous work with a different model. For information, earlier models were validated against (bulk) phytoplankton biomass. Since section 4.3 this is an add on section at the end of the manuscript, these effects are only vaguely described. A thorough analysis of these effect would improve the manuscript significantly and make an important contribution.*

Responses: We strongly agree with this comment, and we thank the reviewer for helping us to point this out. We have shifted the focus of this study to the potential silicate limitation and the effects of a complex plankton community on hypoxia and DO dynamics.

*4) Are the simulated effects real? This is hard to believe that an 80% N load reduction will result in a 25% increase in hypoxia. By which mechanism? Si/P transport downstream? but then what is the source of NO3? Could you describe, through schematics the effects of the different load reductions? The model will always give results but the readers need to be convinced that those are realistic. Currently it makes me wonder if the biological model has been properly parameterized. How was the parameterization done after modifying the structure of the model? Using a predatory zooplankton without a proper parameterization may result in a top-down control of the system, which then lead to artificial nonlinearities in the response to decreasing nutrient loads. Zooplankton is often unconstrained due to the lack of observations, which may be why previous modelling studies used a more parsimonious approach to the model structure.*

Responses: Most of the parameters of our model followed previous model studies of Laurent et al. (2012) for phosphate-related parameterization, Fennel et al. (2006) for parameterization of light inhabitation on nitrification, Shropshire et al. (2020) for the rest.

A linear function of mortality was applied for PS, PL, ZS, and ZL, while quadratic mortality was used for ZP, accounting for the predation pressure of unmodeled predators, like planktivorous fish. We can see both bottom-up and top-down effects in the biomass responses to the different nutrient reduction strategies. Please see the updated Discussion section for more details.

As for the counterintuitive responses of hypoxia to the reduced N supply, we updated our discussion in the manuscript. We reran our sensitivity tests and focused on the riverine nutrient reductions of 60% only. Results indicate that 60% of N load reduction would lead to an increase in the bottom hypoxic area. We attributed such a counterintuitive response to the dominated limited nutrient and the maintenance of positive DO contribution (net production) by the plankton community at layers within the bottom 2 m.

The N is usually limited for the growth of PS and PL, mostly in the shallow middle and west shelf (10 – 20 m) during summers, while P and Si limitations are more commonly simulated by our model. The reduction in N supply only would lead to a slight decrease in photosynthesis. DO contribution by the plankton community maintains positive at both the upper (surface to 2 m above the bottom) and bottom water columns (layers 2 m above the bottom) as in the control run. As total production in both upper and bottom layers would decrease with less N supported, less DO production in the water column could be found. Although SOC would decrease due to less $PON_{sed}$, the total effects of the changes in the three DO source/sink terms are likely to result in a decrease in the bottom DO and a slight increase in the bottom hypoxic area. A detailed discussion has been added in the Discussion section.

*A less important but redundant issue is the use of "means" (or ratios) for validation and analysis. This is sometimes problematic because mean values, shelf wide for instance, are often not representative of the dynamics of the system.*

Responses: In the Validation section, we added plots of 1) nutrient concentration bias against distance to the Mississippi River mouth, 2) DO concentration profiles averaged for different ranges of depth.

As for the SOC validation, in McCarthy et al.'s (2013) study, cruise periods were only listed by months, and thus, we averaged the model results over the corresponding month and performed a model-data comparison. In the comparison of bottom hypoxic waters, the modeled results are not averaged outputs; instead, they are a composite of different bottom DO snapshots corresponding to the cruise date. In the Results and Discussion parts, in addition to the mean value, we added model medians, minimum, maximum, and quantiles for a better demonstration.

Specific comments:

*L134-135: you could mention Hetland and Dimarco parameterization as well.*
Responses: parameters added.

*L258-259: I don't understand this statement*
Responses: This part has been removed from the revised manuscript

*L268-270: you may want to reformulate this, you should say that complexity may be a factor instead of saying that all previous studies were wrong. So far there you did not provide evidence that this could be the case. Can you do that?*

Responses: We have reformulated this sentence.

*L349-351: this is not novel so you should provide evidence of why you think using more plankton groups would change previous findings. Otherwise you are only repeating previous work.*

Responses: We have rewritten this part and restated the focus and novelty of this study.

*L351-352: You should focus on your study here and mention that later in the discussion/conclusions*
Responses: We have removed this sentence.

*L414-418: was this model re-parameterized? if you change the structure of the model you will need to change your parameter set accordingly, e.g. manually or though optimization.*

Responses: The parameterizations of our model largely followed previous existing studies, i.e., Laurent et al. (2012) for addon phosphate parameterization, Fennel et al. (2006) for parameterization of light inhabitation on nitrification, Shropshire et al. (2020) for the rest. We have validated our nutrient, SOC, and DO concentration in the validation section. Comparisons indicated that our model provided an improvement in nutrient and DO simulations than previous model studies. Previous studies barely validated nutrient profiles. Our model-data comparison for DO profiles is better than that of existing studies. Please see the updated validation section for more detailed descriptions.

*L428-429: river P does not follow the Redfield ratio*
Responses: The Redfield ratio was only applied to fill the missing riverine measurements. As Fig. 8 shows (also attached here), riverine N:P did not follow the Redfield ratio indicating that the missing P measurements are rare for the Mississippi and Atchafalaya rivers.

[Figure]

**Figure 8. Daily time series of ratios of nutrient loads from the Mississippi and Atchafalaya Rivers and nutrient ratios averaged over the LaTex shelf (Fig. 2b) from the numerical results. Note that the latter ratios are derived based on the depth-integrated nutrient concentrations (in mmol m⁻²). The black dashed lines denote the nutrient ratios of 16:1, 1:1, and 16:1 in (a), (b), and (c), respectively. The gray patches indicate the late spring and summer (May–August) period of each year. The capitalized letters of M, J, S, and D in the x-axes denote the first day of March, June, September, and December, respectively.**

*L522-523: the way it is presented is confusing, you should say that it is coupled nitrification-denitrification, as in the Fennel et al model, which implies that NO3 produced in the sediment is used for denitrification*
Responses: The corresponding descriptions have been simplified.

*L564: what is the horizontal resolution in km?*
Responses: The horizontal resolution is about 5 km and has been addressed in the first sentence of this paragraph.

*L683: you could show that with biomass data*
Responses: The high diatom productivity could be found in Table 1. Instead of pasting the data here, we added a reference to Table 1.

*L689: you are mixing vertical and horizontal locations, a 35m peak can either be located at the bottom in 35m waters or in the subsurface in deeper waters. Similarly, a 15m peak can indicate deep observations in nearshore waters or subsurface conditions in deep waters*
Responses: We have removed this confusing statement as it would not provide evidence for our analysis. Here we aimed to provide validation for nutrient profiles rather than in-depth analysis.. Further, probability histograms were replaced by profile biases against distance to the Mississippi River mouth.

*L691: can you provide the average bias?*

Responses: Profile biases against distance to the Mississippi River mouth were added to Fig. 3.

*L775-778: what is the purpose of these numbers? Are these values varying seasonally?*
Responses: The probability histograms were replaced by profile biases against distance to the Mississippi River mouth. Thus, we removed this part but added a corresponding description for the average biases.

*L779: It depends where and when these differences occur. If it is close to the river source in Spring then yes, but if it is farther downstream in summer then this is a significant difference.*
Responses: Profile biases against distance to the Mississippi River mouth show observation biases are high only near the river mouths (<70 km for the Mississippi River mouth, ~250 km for the Atchafalaya River mouth). At other sites away for the mouths, biases are much lower. It supports that "The nutrient concentrations bias between simulations and observations is acceptable concerning the strong influences of high riverine nutrient loads on the shelf".

*Figure 3: are these averages for the area in Figure 2b? this should be mentioned. Also why averaging over this very large region with very heterogeneous conditions?*
Responses: The averages were not for the area in Figure 2b, instead, we extracted the modeled nutrient profiles at the locations shown as blue crossings (WOD-derived measurements) in Figure 2c on the date of the measurements and performed averages of these modeled profiles.

*L787: see previous comment*
Responses: The modeled averages were performed for the inner shelf and midshelf, respectively, according to the Fig. C2 in Appendix C. In Schaeffer et al. (2012) and Chakraborty and Lohrenz (2015), locations of sample sites were not provided (Figure 1 in both studies). We then restricted the lon/lat range of the inner shelf and mid-shelf according to the figures shown in their studies.

*Table 1: can you add columns with biomass?*
Responses: Schaeffer et al. (2012) provide biovolume, and Chakraborty and Lohrenz (2015) provided chlorophyll *a* of different plankton groups. Neither study provided plankton biomass data.

*Section 3.4, Figure 4: SOC observations are often heterogeneous and it is not expected that your model exactly match observations at a particular location given the simple representation of the sediment. However, it would be useful to show how your sediment layer behave (since it is a new addition) with time series of PON accumulation/respiration and SOC at several representative locations of the shelf. In comparison you can compare with the measurements of McCarthy et al and others (Murrel and Lehrter?, e.g. 10.1007/s12237-010-9351-9).*

Responses: We strongly agree with this comment. Model-data comparison suggested that our model can capture well the SOC magnitude. We have removed the comparison of SOC/overlaying water respiration. In McCarthy et al.'s (2013), the overlaying water was the layer ~20 cm above the bottom. In our model, as a sigma vertical coordinate system was used with 36 vertical layers designed, the thickness of the bottom layer is usually ~ 1 m. We have

added a time series of integrated sedimentary PON (Fig. 11 in the updated manuscript; also attached here) over the LaTex shelf to illustrate the behavior of the PON accumulation.

[Figure]

**Figure 11. Comparisons between daily PON_{sed} and plankton biomass (i.e., (a) PS, (b) PL, and (c) secondary production). All biomass matrices were integrated over the entire water column and over the LaTex shelf.**

*Figure 5: it would be better to show a comparison of water column respiration instead*
Responses: We did not provide a comparison of water column respiration here. On one hand, in the McCarthy et al.'s (2013), only samples at 4 layers of the water columns were collected (i.e., surface, middle, bottom, and overlaying). In contrast, there were 36 vertical layers designed in our model. Thus, there should be a great bias between the estimated depth-integrated respiration provided by the McCarthy et al. (2013) and our estimates. On the other hand, as the literature did not provide exact sample dates, biases are introduced between monthly averages of simulations and the cruise measurements. The model is not expected to capture well with the depth-integrated water respiration.

*Figure 6. comparing average profiles of DO doesn't make much sense at the scale of the shelf. The envelope, which represents actual observations) indicates here that hypoxia tends to be found in shallower waters in the model with somewhat more severe conditions. The large differences in panel (d) is probably the result of this mismatch in space.*
Responses: We updated this figure (also attached here), showing averaged profiles against the normalized depth for different depth ranges. Despite some overestimations (~ 1 mg L$^{-1}$) of DO profiles, our model results, in general, provided similar and even better performance than previous numerical studies. For example, DO concentration biases given by Yu et al. (2015) were within 2 mg L$^{-1}$. And to our best knowledge, no existing study has ever tried to provide one-to-one comparison between model-simulated DO profiles and observed ones.

[Figure]

**Figure 5. Comparisons of DO profiles between model hindcasts and measurements by (a–c) NOAA's shelf-wide cruises, and (d–f) SEAMAP. The normalized depths of 0 and 1 represent surface and bottom, respectively.**

*Section 3.6: can you also provide a mid cruise comparison in the appendix/supporting material? Can you also mention in the Methods (may be I missed it) how is the bottom sediment layer was initialized?*

Responses: The model results shown in Figure 6 is not averaged results during the cruise periods. Instead, they were composites of different DO snapshots spanning over the cruise period. The cruises were always conducted from the east to the west, we thus "sampled" the modeled snapshots from the east to the west following the cruise dates. For example, on day 1, if the cruise reached 91W, model DO over the east of 91W was "sampled" as the 1st snapshot and was added to the composite first. On day 2, as the cruise reached 92W, model DO between 91W to 92W was "sampled" as the 2nd snapshot and was added to the composite, and so forth. We added relevant descriptions in the first paragraph of section 3.6.

The sedimentary PON, burial PON, sedimentary Opal, and burial Opal were initialized as 0.1 mmol m$^{-3}$. We have added it in the Method (Line 236).

*Section 4.1. This belong in the Results section*
Responses: We have removed this section.

*L1052-1063: The way it is formulated it sounds novel but this type of regional budget has been done previously, you should mention that you are doing the same type of budget but with your new model.*
Responses: We have removed this content.

*L1065-1066: It make sense that water column biogeochemistry does not have a significant effect because you look at the bottom DO and therefore you only include respiration within this (thin) layer. However water column BG influence the entire water column and is relevant to bottom DO in the subsurface layer. At this scale you may find that water column BG is as important as SOC for bottom DO.*

Responses: Our new results indeed showed that the biochemical processes in the water column (both bottom 2 m and layers above) are as important as SOC for bottom DO. Please see the updated discussion section.

*L1122-1124: You point out the issue with this analysis (i.e. previous comment), it would have been more relevant to look at the bottom (lower 5 or 10m) or subsurface layer. Vertical diffusion might have shown a seasonal cycle then. Also, looking at the bottom layer only, you artificially increase the contribution of SOC on DO.*

Responses: In the updated version, we focused on the depth-averaged DO concentration at layers within the bottom 2 m rather than the DO at bottom layer. We also removed the analysis of this part.

*L1160-1161: isn't this expected?*
Responses: We have removed this part.

*1163-1459: not clear. It is difficult to interpret such pattern with shelf averages*
Responses: We have removed this part.

*Section 4.3: 3 years simulation including 2 years (2018, 2020) with mid summer wind events. Is this enough? What were the nutrient loads, river discharge during these years?*
*did you initialize your model from your long run?*
*you didn't show the dynamics of the sediment PON pool, that would be an interesting addition*
Responses: We extended the length of simulations to 9 years (2012-2020). The river water discharges were kept the same as those in the control run. The riverine nutrient concentrations were reduced by 60% with different reduction strategies (i.e., -60%N, -60%P, -60%Si, -60%(N+P), -60%(N+Si), and -60%(N+P+Si)). Thus, the nutrient loads were decreased with the shelf hydrodynamic unchanged. The sensitivity runs were initialized from the long-term (15-year) control run. We have added relevant discussion about the $PON_{sed}$ pool to the revised manuscript.

*L1495: here you say that this is a mean for mid summer hypoxia and the below it becomes a multi year summer mean, which is very different. The good metric to show the effect of nutrient reduction is to take either a season mean or seasonally integrated hypoxia (the time integral of the hypoxic area)*
Responses: We have updated this figure. Previous comparisons were all based on the mid-summer (during the shelf-wide cruise) mean matrices. In the updated version, we focused on the statistic matrices conducted for the May-August period of each year.

*Figure 11: % change from what value? in Figure 8 you show that mid summer hypoxia is very small in 2018 and 2020(see general comments) These results are hard to believe. You need to be*

*more convincing. How do you explain that N80, P80 result in an increase of the diatoms, which are the dominant phytoplankton in the river plume. What is their source of nitrogen? River N load has been shown to be well correlated with mid-summer hypoxia. Your results suggest that Si load is the best predictor for hypoxia.*

Responses: We have updated this figure. New results indicate that diatoms would not change much under the N60 scenario. But PS would experience a slight decrease. It was due to that the shallow parts of the mid and west shelf were usually limited by N in summer. However, in other parts of the shelf, P and Si limitations were more common. Previous studies showed that riverine N loads are well correlated with mid-summer hypoxia. However, riverine P and Si loads are both highly correlated to riverine N loads (see below table), which indicates that riverine P and Si loads are both well correlated with the mid-summer hypoxia as well. Here we would like to investigate, by using a mechanistic model rather than a statistical one, whether the introduction of the Si cycle and a more complex plankton community would provide different results from previous model studies. We agree that the Si limitation in the LaTex shelf was rarely shown in published studies. Yet our new model results indicate that it is worth conducting more data collection, including the possible contribution from Si to hypoxia development.

**Table C1. A correlation matrix of daily inorganic nutrient loads by the Mississippi River and the Atchafalaya River from 2007 to 2020. Correlation coefficients shown are all significant ($p<0.001$).**

|  | Mississippi nitrate+nitrite | Atchafalaya nitrate+nitrite | Mississippi phosphate | Atchafalaya phosphate | Mississippi silicate | Atchafalaya silicate |
|---|---|---|---|---|---|---|
| Mississippi nitrate+nitrite | 1 |  |  |  |  |  |
| Atchafalaya nitrate+nitrite | 0.9123 | 1 |  |  |  |  |
| Mississippi phosphate | 0.8328 | 0.7577 | 1 |  |  |  |
| Atchafalaya phosphate | 0.7517 | 0.7913 | 0.9155 | 1 |  |  |
| Mississippi silicate | 0.8583 | 0.7795 | 0.8759 | 0.7942 | 1 |  |
| Atchafalaya silicate | 0.7938 | 0.7956 | 0.8131 | 0.8148 | 0.9520 | 1 |

*L1556-1558: you talk about nonlinear response but you don't provide the mechanisms. please explain the mechanisms, diagrams would be useful to support these explanations*
Responses: In the updated manuscript, we focus on the explanations of the nonlinear responses, starting from nutrients to biomass and then to DO dynamics.

*L1572: what about phytoplankton biomass?*
Responses: Please see our response to comments for Table 1.

*L1640-1648: This is the novel part of the study but I have a hard time to believe these results, they do not make sense ecologically. The authors should show the mechanisms, nonlinearities that explain their results.*

Responses: Please see our response to general comments 4). We have provided related discussion in the revised Discussion section.

*L1650: does that follow the BG policy or should the results be available from a repository?*
Responses: We will deposit our model data in a public repository.

Minor comments/typos:

L258: Fennel et al is cited twice (later in the Discussion as well)
Responses: Have corrected.

L347: typo, 3->2 and 2->3
Responses: Have corrected.

L352-354: this is not necessary
Responses: We have removed this sentence.

Figure 7: can you show 2020 as well?
Responses: We could not find the source data of 2020 shelf-wide cruise.

L1506: for mid summer or for whole summer?
Responses: Previous results were based on the mid summer statistics. In the updated version, we focused on the statistic matrices conducted for May-August period of each year.

L1144: sediment biogeochemistry
Responses: Have corrected.

Figure 12: your color scale is a bit counter intuitive
Responses: We have removed this figure.

Responses to Comments by Referee #2

General responses (we also included this in our response to Referee#1):

      We sincerely thank the reviewer for providing us with such detailed comments and suggestions and also his/her patients for our prolonged responses as we carried out a series of new experiments and reshaped the article.

      We reorganized our thoughts for this study and would like to focus on 1) the role of silicate cycling in the biogeochemical processes and bottom hypoxia development in the Louisiana-Texas shelf and 2) the impacts of a complex plankton community on the dissolved oxygen (DO) dynamics. Thus, we removed the original sections 4.1 and 4.2, which discussed the impacts of physics. Instead, we focused on the biogeochemical processes in DO dynamics.

      We also separated the Results and Discussion sections. In the Results sections, we tried to answer the following questions. 1) What are the limited nutrients for PS and PL, respectively? 2) What is the dominant plankton group on the shelf? 3) How do different plankton groups contribute to the source and sink of DO in water columns and sediments? In the Discussion section, we reran all sensitivity tests and expanded the 3-year (2018–2020) simulations to a 9-year (2012–2020) one. We aimed to discuss the responses of biomass and DO to the reduced riverine nutrient supplies. Please find our detailed point-to-point reponse to your comments as follows,

*This version of manuscript by Ou et al. improved the previous version of manuscript somewhat. The reviewer appreciated the modification on the bottom DO budget analysis and additional sensitivity runs on the sensitivity runs to evaluate the model's robustness regarding different parameterizations. The contributions of different biogeochemical and hydrodynamic processes on bottom DO, the bottom DO's response to the reduction of riverine nutrient loads, and the impact of diatom on hypoxia dynamics were better analyzed. Although I see improvement of the manuscript, there are still some issues that needs to be addressed, which I think is significant and necessary. In particular, this paper missed real discussion section including the comparison with previous work, and advanced understanding on the topic. In other words, although much work has been done for this research, this paper is still organized like a technical report of result, lack of the understanding of both physical and biogeochemical mechanisms. My major comments and concerns are listed below.*

Responses: We have separated the Results and Discussion sections. We added comparisons with previous studies in the Discussion section (please refer to the revised manuscript).

*(1) "Compared with existing modeling efforts, our model, for the first time, included a silicate cycle as well as multiple plankton functional groups in the modified biogeochemical model, the importance of which has already been addressed in previous studies yet not included in hypoxia modeling efforts." As the authors indicated in the reply, the highlight of this paper should be adding the silicate cycle as well as multiple plankton functional groups contribute to the hypoxia simulation and understanding in this area. However, although the authors validated the new model well with nutrient profile, SOC and diatom percentage, the bottom DO as well as subsurface DO was significantly overestimated (Figure 6), and the hypoxia area was underestimated correspondingly. Moreover, the interannual variability of SOC was actually not well captured by the model (shown by Site F5 and C6). Therefore, there might be systematic defect with the model, either hydrodynamics or biogeochemical processes, that need to be at least thoroughly discussed.*

Responses: We change the focus of this study to the potential contribution of Si cycling in hypoxia development and the effects of a complex plankton community on DO dynamics.

SOC observations are often heterogeneous. Model-data comparisons suggested that our model can, in general, capture the SOC magnitude. As the model results were monthly averages, it would be hard for the model to fit perfectly with the measurements.

We updated Figure 6 and reordered it to Figure 5 (see below), showing averaged profiles against the normalized depth for different depth ranges. Despite the overestimations (~ 1 mg L$^{-1}$) of DO profiles, our model results provided similar and even better performance than previous numerical studies. The DO concentration biases in Yu et al. (2015) were within 2 mg L$^{-1}$. And the bias of our model outputs is less than 1 mg/ L$^{-1}$, which gives us confidence about the model performance.

[Figure]

**Figure 5. Comparisons of DO profiles between model hindcasts and measurements by (a–c) NOAA's shelf-wide cruises, and (d–f) SEAMAP. The normalized depths of 0 and 1 represent surface and bottom, respectively.**

*(2) The authors applied oxygen budget analysis on the bottom DO, tried to discern the major contributors for bottom DO. This part of manuscript did not make much contribution to the study and what have been shown were mostly covered by previous studies, i.e. SOC was the dominant term especially in the nearshore area. Also, the physical mechanisms between stratification and advection terms were not explained well. Therefore, the reviewer suggests removing or shorten this part and focus on the nutrient and lower tropic community impact on bottom DO.*

Responses: The sections 4.1 and 4.2 have been removed following the reviewer's suggestion.

*(3) Following my previous comments, this paper still lacked comprehensive discussion and*

*comparison with previous modeling study on the model performance, simulation results and conclusions, specify and address what were the agreement, what were the disagreement, and what was new. The authors claimed that the bias of overestimation on DO was acceptable, however, the quantitative comparison should be provided with other model works in this area, like RMSE, bias, correlation coefficient, etc., to prove the improvement of this new model. Even if the new model underperformed compared to previous models, discussions on the potential causes were needed.*

Responses: We separated the Results and Discussion sections and added the comparison between our model results and previous simulations in the Validation section (also see responses above).

*Detailed comments:*

*Abstract*

*Modify "biochemical" to "biogeochemical" in the title and throughout the entire manuscript. The biogeochemical cycle links the living biomass in the water column to the sediment. Biochemical is only part of the processes discussed and less used for the topic.*

Responses: Have corrected. But we would like to keep "biochemical" in our title as this is the PartI of our duo-paper, the PartII paper about applying Machine Learning in hypoxia prediction is published in 2022.

Ou, Y., Li, B., & Xue, Z. G. (2022). Hydrodynamic and biochemical impacts on the development of hypoxia in the Louisiana--Texas shelf -- Part 2: statistical modeling and hypoxia prediction. *Biogeosciences*, *19*(15), 3575–3593. https://doi.org/10.5194/bg-19-3575-2022

*L13-15. 16-17: that's basically all the terms… which are more important in the physical terms and why in the aspects of hydrodynamics?*

Responses: The sections 4.1 and 4.2 have been removed. The Abstract has been updated accordingly.

*L20: add period after the bracket*

Responses: corrected.

*L20: how about water column and sediment nutrient recycling associated with change in oxygen condition? (Kemp and Testa et al, 2012)*

Responses: In this study, we focus on how the complexity of the plankton community will lead to different DO responses. Please see the updated Discussion section.

*Introduction*

*L38-40: clarify the terms: overlying water, surface water*
Responses: The overlying water mentioned in McCarthy et al. (2013) was the water layer about 20 cm above sediments. We did not mention surface water here.

*L52, L80: modify "biochemical" to "biogeochemical"*
Responses: please see our earlier response.

*L57-65: this paragraph is suggested to move after paragraph L67-76; sentences should be added in this paragraph about how phytoplankton species affect hypoxia size*
Responses: Have modified.

*L80: these is no such term as sedimentary biochemical…consider biogeochemical processes or other technical terms*
Responses: The term "sedimentary biochemical" has been replaced by "biogeochemical processes at sediment layers".

*L80-81: remove (Fennel et al., 2016)*
Responses: Corrected.

*L85-91, 102-103: The necessaries of using higher level representation of plankton community should be highlighted and further specified in the introduction. Reviews of previous work on the defects (for example, poor model performance on reproducing observational Chl-a) of using simplified representation on trophic level should be expanded in addition to L57-65. It was noted by the authors that the influence of the community complexity can be reflected in the SOC and eventually in the bottom DO variability. The goal of this study should prove model performance improvement by adding additional plankton groups by providing better model validation compared to previous works.*
Responses: We have specified these in Lines 84–91 where we also posted our focuses of this study. We also compared our model performance with previous in the updated Validation section. For example, DO concentration biases against profile measurements were found within 2 mg $L^{-1}$ in Yu et al. (2015) but were found within 1 mg $L^{-1}$ in our study.

*L97, 102: please check how many groups of phytoplankton and zooplankton carefully.*
Responses: Sorry for the typos. The numbers of phytoplankton and zooplankton groups should be two and three, respectively.

*Biogeochemical model validations*
*L303-304, Figure 3: which are the profiles shown? If it is statistics of multiple profiles, please normalize the vertical depth.*
Responses: It is statistics of multiple profiles. We aimed to provide nutrient profile validation in this section rather than to provide any in-depth analysis. In this revision, we replaced probability histograms with profile biases against distance to the Mississippi River mouth. Please see the updated section 3.2.

*L323, 324: plot nutrient concentration bias according to the distance of observation stations to the river mouth would help to discern the influence of river load and model itself*
Responses: The plot has been updated according to the suggestion (updated Fig. 3).

*L354: how thick is the bottom water column of overlying water? Overlying water is a very confusing term, please describe it with depth range or other more accurate way*
Responses: In the incubation study by McCarthy et al. (2013), the overlying water layer above sediments (depth 20 cm above sediments) was isolated for a separate incubation. Respiration rates at the overlying water were then measured. However, in our model, there was no overlying water layer added. In our model, as a sigma vertical coordinate system was used with 36 vertical layers designed, the thickness of the bottom layer is usually ~ 1 m. It is hard to expect the model can reproduce well enough the measured overlaying water respiration with the simulated bottom water respiration. So, we removed the comparison of SOC/overlaying water respiration.

*The interannual variability of SOC was actually not well captured by the model (shown by Site F5 and C6). The authors tried to explain the bias with river point sources diverting at the computational grids. However, if that was the case, the interannual variability should have been captured although there might be a discrepancy on the magnitude. Therefore, there might be a systematic defect with the model, either hydrodynamics or biogeochemical processes, that need to be discussed. This can be found in Figure 8 for the overall model underestimation of HV, especially for the year 2007, 2010, 2012*
Responses: It is hard for monthly mean simulations to capture exactly the measured SOC. We acknowledge the overestimation of DO profiles, especially at lower layers, which may be ascribed to the coarser vertical resolution near the bottom and the vertical mixing parameterization applied. We have posted relevant discussion in the updated section 3.2 and 3.4.

*L379-381, Figure 6: why (a) showed model significantly overestimated DO compared to the measurement from subsurface to bottom layer, while histogram (d) did not show any bias? The model showed consistent overestimation of surface DO, suggesting there might be issues with model air-sea oxygen flux calculation or phytoplankton production dynamics.*
*Suggest validating DO profile data by season (spring, summer, winter) to discern the probable causes.*
Responses: We have posted relevant discussion in the revised section 3.5.

Results and Discussion

*L434-439: please specify the oxygen budget equation applied, the integration area, depth, and method, etc. in the method section or result section. Use equation to represent all the terms, including the individual terms in the water column respiration. When you mention bottom DO concentration balance, what is the depth of water column?*

Responses: The section 4.1 and 4.2 have been removed. The revised manuscript focuses on the biogeochemical term only (DO source by photosynthesis, DO sinks due to phytoplankton respiration, zooplankton metabolism, nitrification, microbial decomposition of dissolved and particulate organic matters, and sedimentary oxygen consumption). In Eq.4, we provided expressions of these terms. Also, we updated Eq. 4 as the decomposition of particulate organic matter was missed in the previous submission.

In the updated manuscript, DO source/sink terms were averaged over the layers within the bottom 2 m and over layers above, respectively. The bottom DO concentration was represented by the average DO concentration over layers within the bottom 2 m.

*L441: why only selected the bottom water column, instead of using the water column beneath pycnocline as previous study?*

Responses: See the above responses.

*Figure 9: the DO budget terms are generally shown with signs rather than absolute value. Positive sign indicates replenishment of DO while negative sign represents consumption. Horizontal and vertical advection mostly cancel out each other. Therefore, using absolute value can be misleading. Please modify Figure 9 accordingly.*

Responses: The section 4.1 and 4.2 have been removed.

*Replace "water column biochemistry" with "water column process" throughout this section and figures*

Responses: Done.

*L450-451: where did you get this conclusion? Please explain a bit more.*

Responses: The section 4.1 and 4.2 have been removed.

*L455: again, what was the depth range of bottom layer of this study?*

Responses: See the above responses.

*L494-496: Lack of explanation on the physical mechanisms of stratification on advection terms. Please add the physical understanding here, including the circulation and mixing dynamics in LaTex area. Why the enhanced stratification would suppress DO advection, which is not the case in other hypoxic estuaries like the Chesapeake Bay? What did it mean that "the enhanced water stratification in summer usually leads to less DO exchanges due to advection at the bottom layer"?*

Responses: Sections 4.1 and 4.2 have been removed.

*L501-505: with this explanation, the negative correlation between PEA and vertical diffusion of DO makes little sense since the bottom DO concentration was the controller. The stratification just covaried with seasonal freshwater discharge and/or wind strength in this area.*

Responses: Sections 4.1 and 4.2 have been removed.

---

## Author Response (AR3)

Responses to Comments

General comments:

The revised manuscript is somewhat refocused toward nutrient limitation, plankton community dynamics and their effect on hypoxia. The study concludes that Si and P are the most limiting nutrients in the region. Although this is a legitimate conclusion based on their model results, I am not convinced that this is a good representation of the system.

The model indicates widespread P limitation in PS and widespread Si limitation in PL, with N limitation found sometimes on the western shelf. Does this agree with our current understanding of nutrient dynamics in this area? Si limitation patterns do not seem to agree with the literature cited in the manuscript.

My worry is that the model is not well tuned and constrained by observations, including resource limitation data. The authors mention the importance of having more complexity in the model to better represent resource limitation and oxygen dynamics. However, more complexity results in more nonlinearities, which may not be real if the model is not well constrained by observations. The fact that N mitigation results in less oxygen in bottom waters may be one on these unconstrained nonlinearities.

Another issue is that results are interpreted in length but not discussed. The Discussion section presents new results about nutrient mitigation experiments but those are not discussed in light of the literature. Results presented in section 4 are not discussed later on. The lack of discussion limits the amount of trust the reader has regarding the model results.

Overall, model results should be more supported by observations and by a mechanistic understanding of resource limitation in the region.

Responses to the general comments:

After carefully double-checking the parameterization of our biogeochemical model, we found that the half-saturation coefficients for $PO_4$ (KPO4S and KPO4L) needed to be appropriately designed. We updated the half-saturation coefficients for $PO_4$ to be 0.03125 mmol P $m^{-3}$ for the PS and 0.1875 mmol P $m^{-3}$ for the PL, 1/16 of the corresponding half-saturation coefficients for $NO_3$. This parametrization method was also applied by Laurent et al. (2012) to discuss the effects of P limitation on the LaTex shelf. We reran the 2006-2020 hindcast and updated all the results accordingly.

According to the new results, P limitation usually occurs around the river outlets, while N and Si limitations are found in the middle and west LaTex shelf. We added section 3.3 for nutrient limitation validation in the updated manuscript. Our N and P limitation patterns align well with previous bioassay studies. While we could not find any bioassay studies in the west shelf (<-92°W) in the recent two decades related to the discussion of Si limitation, indirect evidence from concentration measurements (Dortch and Whitledge, 1992) suggested that Si limitation could overwhelm the N limitation in the deep gulf waters (depth > 50 m). Our model studies show that the Si limitation is most induced by the intrusion of open ocean water to the western shelf (Figures 11 f and g). Recent studies also pointed out that marine diatoms require a lower N:P:Si (=16:1:20) ratio (Billen and Garnier, 2007; Royer, 2020), indicating that Si limitation is highly possible even if Si concentration is higher than N concentration. We posted the discussion on Si limitation in sections 3.3 and 4.1.

We tried to illustrate the impacts of plankton complexity on productions and DO dynamics, starting with intensive validation and ending with analysis based on

multiple snapshots from the model results. Our model successfully reproduced a bi-peak primary production pattern in spring and early summer, aligned with the pattern from satellite-derived chlorophyll *a* concentration (see section 4.2 and Fig. 12 in the manuscript). This pattern was attributed to the competition of different phytoplankton functional groups for nutrients and grazing pressure from the zooplankton groups. The combined effects can lead to spatial differences in PS and PL distribution and further the bi-peak total primary production in the LaTex shelf. We found direct evidence of the spatial difference of the dominated phytoplankton species from a cruise study in 2013 and 2014 (Anglès et al., 2019). Our results (Fig. C4–C5) aligned well with their findings. We further sampled multiple snapshots of different DO contribution terms (Fig. 16–17) and demonstrated that different planktonic groups contributed differently to DO changes in the upper water column and further affected the DO gain/loss patterns in the bottom layer through physical transport processes (e.g., vertical diffusion).

In the revised one, we merged the results and discussion in section 4, which were divided into three parts focusing on (1) nutrient limitation, (2) plankton community interactions, and (3) DO dynamics. In each subsection, we posted our findings, followed by a discussion, including a comparison against previous studies, more observational evidence supporting our findings, and suggestions on further observational studies and model development.

Specific comments:

L28: it is odd to associate the timing of hypoxia with mid summer cruises (which do not provide temporal information)

Responses: This conclusion was based on monthly observational data and continuously recorded data from earlier hypoxia research (Rabalais et al., 1991; 2002). We cited these works in our revised manuscript on L26–27.

L76: doesn't that contradict the previous sentences?

Responses: This statement was based on the McCarthy et al.'s (2013) results where the SOC was measured 7-fold greater than the respiration rate at water overlaying the sediment. Then we calculated the ratio of SOC/(SOC+overlaying respiration)=0.87 to emphasize the importance of SOC in changing the bottom DO. We removed this sentence in the revised manuscript to avoid confusion.

L86-87: It is the other way around, at the peak of the bloom there is more SOC because more deposition and less SOC in the subsequent months.

Responses: Thanks. We have corrected this statement in L48–49.

L174-178: This is weak evidence of the importance of Si limitation in this system

Responses: We kept this part in the introduction and added more observational evidence by Quigg et al. (2011), seeing L75-76.

We added more evidence in section 3.3 (validation of nutrient limitation) and section 4.1 (results and discussion of nutrient limitation). In section 3.3, Si limitation is supported by bioassay studies by Nelson and Dortch (1996), Quigg et al. (2011), and Smith and Hitchcock (1994). Although we did not find bioassay studies related to Si limitation over our analysis period (2007–2020), strong clue of Si limitation has been documented and should not hinder model development.

In section 4.1, we discussed that Si limitation is possible as the riverine N:Si is near 1:1 and marine diatoms require a lower N:P:Si ratio (16:1:20, Billen and Garnier, 2007; Royer, 2020) than the Redfield ratio (16:1:16). As riverine Si:P is larger than 16:1 and 20:1, P limitation is more pronounced around the river mouths as evidenced by observations and our model results (Fig. 4–5). However, in waters far from the mouths, P limitation is usually replaced by N or Si limitation. Accordingly, although Si is excessive over P around the river mouths, Si can be more limited than N in other shelf regions as the uptake efficiency of Si by phytoplankton is somewhat higher than that of N. Our model results show that Si limitation develops as currents turn eastward in summer, allowing intrusion of waters with a higher N:Si ratio than 1:1 (Fig. 11 and C3). This mechanism has not yet been discussed in previous studies.

L180: That is not true. What observations that emphasize N and P limitation?

There is no biology in Hetland and DiMarco (2008) so unless you refer to oxygen data your statement does not apply to this study. For the other cited studies, the models were validated against these observations.

Response: We rewrote this statement and removed the citation of Hetland and DiMarco (2008). Please see L82–83 in the revised manuscript.

L183-187: You need to provide more support for your study, saying that previous models are based on misleading observations and oversimplified is not enough to justify

your study. Why and how do you think Si is an important limiting nutrient on the shelf? Why and how do you think multiple plankton groups help to better characterize hypoxia?

Responses: We done a throughout literature research and listed supports of Si limitation by previous observational studies in L73–80. We added statements about the bi-peak primary production pattern that is captured in satellited-estimated chlorophyll *a* and Gomez et al.'s (2018) model (two phytoplankton + three zooplankton function groups) study but is not captured in the models with a less complex planktonic community (L87–91). We highlighted our objective to investigate the possible Si limitation and to assess the impacts of the complexity of the plankton community on DO dynamics and bottom hypoxia development (L92–94).

L286-287: what do you mean?

Responses: We would like to address the biological concentration (e.g., PS, PL, ZS) and organic matter concentration are represented by N (i.e., in mmol N m$^{-3}$) rather by P or Si. We rewrote this sentence to avoid confusion (see L129–130).

L339: other way around, see earlier comments

Responses: Have corrected. Please see L168–169.

L495-496: why only 2.5 years?

Responses: We only found 2.5-year WOD records (https://www.ncei.noaa.gov/products/world-ocean-database). To expand our observation dataset for nutrient validation, we also incorporated the shelf-wide cruise measurements of nutrients (please see the updated section 3.2).

L508: but this is average in space and time over 2.5 years right? Also, as previously mentioned, validating by looking only at means is misleading.

Responses: We updated section 3.2, focusing on validating surface nutrient concentration. In the revised section, we performed one-to-one comparisons between modeled hindcast and observed records from the WOD and shelf-wide datasets. Bar graphs showing the percentages of concentration differences within specific concentration intervals and concentration differences against the distances between the Mississippi River mouths and the sampled locations illustrated the summarization of the model-observation misfits (Fig. 3).

L509-510: I don't think these levels qualify as oligotrophic

Responses: We updated this section and this statement has been removed. Please see the updated section 3.2 from L288 to L304.

L510-511: You cannot say that from Figure 3

Responses: We updated this section, and this statement has been removed. Please see the updated section 3.2.

Figure 3: the bias is quite large in some regions and the sign of the bias (positive for NO3, negative for PO4, Si) may favor the development of Si or P limitation.

Responses: We updated this section after we re-parametrized and reran our model. New results suggested that one-to-one biases were acceptable. Please see the updated section 3.2.

L588: "reasonably well" would be a better statement

Responses: We have updated this section and removed this statement. In the revised section 3.4, we first provided more detailed information about how we performed the model-observation comparison than we did in the previously submitted manuscript. Second, we found a better alignment between hindcast and measurements than in the last version after reparameterizations.

L665: Figure 3 shows well that although there is a good agreement between mean profiles, 1 to 1 comparisons indicate large biases.
Responses: As for the validation of DO profiles against the SEAMAP and shelf-wide cruise measurements, we provided a one-to-one comparison in our revised manuscript. Similar to the validation of diatom ratios, in section 3.6, we first provided a detailed description of how we performed the model-observation comparison. That is, observed DO profiles were interpolated to the modeled layers using the nearest interpolation method as the number of observed layers is close to or even more than that of the modeled layers. We then plotted the vertical profiles of mean, median, and 25–75th percentile ranges of the one-to-one model-measurement differences. We argued that our model results provided a better representation of measured DO profiles than previous numerical studies (e.g., Yu et al., 2015). Please see the updated section.

L678: As mentioned in a previous review, in Figure 5 you compare regional mean profiles, which is not a very good comparison, especially along such a large longitudinal range. Also I believe Yu et al compared actual profiles, which is a different level of validation. Point to point comparison might show large biases, cf Figure 3
Responses: Please see our previous responses.

Table 2 (0.02): this is odd. I am not sure I believe the explanation for the mismatch
Responses: We updated Table 2 according to our new experiment results and performed 10-year running correlation coefficients (CCs) between hindcast and shelf-wide measurements. The 10-year running CCs should be more statistically meaningful than the 5-year running CCs. We also provided significant tests for these CCs and found they all significantly showcased the model's accuracy in reproducing year-to-year variations of hypoxic areas.

L785: I am not sure that it is pertinent to use shelf and depth averaged nutrient ratios to discuss growth limitation.
Responses: We rewrote the entire section 4.1 to discuss nutrient limitation as we agreed with the reviewer's doubts that using shelf and depth-averaged nutrient ratios may be problematic.

In the revised section 4.1, we first addressed the possible types of limited nutrients based on the ratios of riverine nutrient supplies. We found that P can be more limited than N and Si around the river mouths, while N and Si limitations may vary in other shelf waters. Si limitation is possible that riverine N:Si was near 1:1 and, during some summers, was greater than 1:1 and that marine diatoms require a lower N:P:Si ratio (16:1:20, Billen and Garnier, 2007; Royer, 2020).

Secondly, we moved the discussion of half-saturation of $Si(OH)_4$ uptake by PL ($K_{SiOH4}$) from section 5.1 in previous submission to here in this revised manuscript. We would like to address that our selection of $K_{SiOH4}$ is reasonable as it was based on multiple bioassay studies.

Finally, we tried to illustrate the Si limitation, its causes, and its impacts on the plankton distribution and $PON_{sed}$ (directly related to SOC) distribution. We further

argued that N and P limitations were reported more frequently than Si limitations along the shelf because samples collected in previous studies were mainly from the eastern shelf. Yet a lack of data on the western shelf should not hinder our attempt to perform numerical investigation and suggest possible Si limitation in the LaTex shelf and low-Si waters in the deep gulf. Please see the updated section 4.1 for more details.

Figure 8: nutrient ratios indicate the potential for growth limitation but this limitation does not occur until one of the nutrients runs out.
Responses: We updated this figure and this section. Nutrient limitation can occur even if such nutrient has yet to run out. We found evidence in many bioassay studies that addressed co-limitation conditions. For example, in Quigg et al. (2011) (we pasted Table 2 in their work here), N, P, and Si limitations were found to coexist in one sample even though when the background nutrients were still detectable.

**Table 2** Resource limitations assays conducted across the Louisiana shelf during three research cruises in 2004 at locations shown on Fig. 1

| Cruise | RLA # | Lat | Long | Salinity | Temp (°C) | DIN (µmol $L^{-1}$) | $P_i$ (µmol $L^{-1}$) | Si (µmol $L^{-1}$) | DIN/$P_i$ | DIN/Si | Chl a (mg $m^{-3}$) | APA (nmol $L^{-1}$ $h^{-1}$) | SPP (mgC mg $Chl^{-1}$ $h^{-1}$) | Primary limiting resource |
|---|---|---|---|---|---|---|---|---|---|---|---|---|---|---|
| March | M1 | 28.78 | 89.83 | 30.2 | 19.1 | 2.73 | 0.21 | 7.41 | 13.0 | 0.37 | 3.5 | 64.4 | 5.12 | NP$_i$Si |
| | M2 | 28.77 | 90.77 | 28.0 | 18.8 | 1.18 | 0.09 | 6.81 | 13.1 | 0.17 | 1.1 | 112 | 2.78 | NP$_i$Si |
| | M3 | 28.84 | 91.59 | 32.3 | 19.4 | 0.14 | 0.11 | 0.91 | 1.27 | 0.13 | 1.6 | 85.6 | 3.66 | N |
| May | Y1 | 28.78 | 89.43 | 28.0 | 27.0 | 0.29 | 0.08 | 1.38 | 3.63 | 0.21 | 0.2 | nd | nd | No growth |
| | Y2 | 29.08 | 90.04 | 23.2 | 27.6 | 12.3 | 0.05 | 23.2 | 246 | 0.53 | 2.4 | 55.7 | 9.02 | P |
| | Y3 | 28.97 | 91.03 | 24.8 | 28.2 | 8.20 | 0.07 | 18.3 | 117 | 0.45 | 0.9 | 30.6 | 7.69 | P |
| | Y4 | 29.14 | 91.96 | 24.8 | 28.6 | 3.76 | 0.03 | bdl | 125 | – | 3.9 | 440 | 8.09 | NP$_i$Si |
| July | J1 | 28.93 | 89.86 | 23.3 | 26.2 | 0.50 | 0.04 | bdl | 12.5 | – | 5.0 | 664 | nd | P + DGlu |
| | J2 | 28.88 | 89.43 | 33.0 | 27.6 | 32.6 | 1.49 | 38.4 | 21.8 | 0.85 | 0.2 | 44.8 | nd | L |
| | J3 | 29.11 | 89.83 | 21.7 | 30.8 | 5.66 | 0.18 | 1.31 | 31.4 | 4.32 | 2.7 | 28.6 | 8.24 | NP$_i$Si |
| | J4 | 28.89 | 90.57 | 25.9 | 29.5 | bdl | bdl | bdl | bdl | – | 1.3 | 16.9 | 18.0 | L |
| | J5 | 28.56 | 91.45 | 23.9 | 28.9 | 4.59 | 0.28 | 12.7 | 16.4 | 0.36 | 0.2 | 3.53 | 19.0 | L |

All water quality, nutrient, Chl a, and APA values presented were measured on the initial bulk water sample used to start each RLA. During the July cruise only, two additional treatments (+phosphono-acetic acid and +D glucose-6-phosphate) were added to examine the potential role of organic P (vs $P_i$) on phytoplankton productivity (see methods section for specific details)

*DIN* dissolved inorganic nitrogen, *$P_i$* dissolved inorganic phosphate, *APA* alkaline phosphatase activity, *SPP* surface primary productivity (=$P_m^B$ defined in methods), *L* light limitation, *bdl* below detection limit, *nd* no data, *DGlu* D glucose-6-phosphate

L931-933: This is very surprising. Offshore waters should be N limited. P limitation does not typically occur there, but the model indicate the opposite.
Responses: According to our new results, offshore waters were limited by N or Si (depending on current patterns), and P limitation occurred around the river mouths (Fig. 11 and C3). Please see the updated section 4.1 for more details.

L941-945: I don't think the results from Quigg et al support your findings.
Responses: In our new results, P limitation occurred around the river mouths, and Si limitation was also detected when the dominated current turned eastward or northward in the shelf. Quigg et al. (2011) suggested the P limitation regime and also the potential of Si limitation. Please see the discussion of Si limitation from L526 to L542.

Figure 9: does your model reproduces the seasonal cycle on resource limitation? Caption: can you indicate that this is at the surface?
Responses: We have removed Figure 9. The revised manuscript discussed nutrient limitation based on multiple surface snapshots sampled from the hindcast results. Please see the updated section 4.1 for more details.

L967-977: are these patterns supported by observations?
Responses: We re-wrote the section 4.2. We tried to demonstrate and validate the bi-peak production pattern by comparing the daily time series of model primary

production (PS+PL; Fig. 12) and monthly time series of chlorophyll concentration from satellite products. In the satellite-derived chlorophyll *a* concentration, a bi-peak pattern was also found in spring and summer. Quigg et al. (2011) also found two chlorophyll peaks in May and July in cruise observation. Please check the L551–L562 in the revised manuscript.

L976: can you explain mechanistically why there is a lag for PS and not PL
Responses: We removed this part.

Figure 10a: do you believe these oscillations?
Responses: We removed this part.

L1010-1023: you can remove this paragraph and keep the last sentence. The rest is confusing and not necessary.
Not sure I believe these results, is this supported by earlier studies? It seems that rather than sinking PL is eaten up and then Z sinks and contribute to PONsed, wheras PS sinks rather than being eaten. This dynamics needs to be supported by observations
Responses: We have removed this part. We tried to demonstrate two mechanisms (bottom-up and top-down) that alternate the PS and PL variations using snapshots sampled from the model hindcast. Please see the brief description in the general responses above and the detailed one in the revised section 4.2.

L1025-1035: I do not understand this paragraph. Also, there is a lot of discussion here but the data are not shown to support it.
Responses: We have removed this part.

1058-1067: similar to section 4.3, I get lost here.
Responses: We have removed this part.

L1063-1067: this is not shown in your results.
Responses: We have removed this part.

L1072: the results presented in section 4 are not discussed. Below you introduce new analyses (nurient load reduction experiments), without much discussion.
Responses: We removed all sensitivity tests and reshaped section 4, merging results and discussion. Please see the brief description in the general responses above and the detailed one in the revised manuscript.

L1067: "while simulated N:Si"
Responses: We have removed this part.

L1231: "ZS biomass increase..."
Responses: We have removed this part.

L1243: this is hard to follow. Can you start your paragraphs by a sentence that summarizes your main point?
Responses: We have removed this part. And by following the review's suggestion, in the revised manuscript, we tried to start our paragraphs by a leading sentence that summarized the main points.

L1260: you are not showing these results
Responses: We have removed this part.

L1270: "Reductions in Si supplies lead"
Responses: We have removed this part.

L1277: this response is the direct result of more complexity in the model and therefore more nonlinearities. But are those real? Is the model well constrained by observations?
Responses: We have removed this part. To discuss the impacts of complexity on bottom DO and hypoxia, in the updated manuscript, we started with the validation of various factors from nutrient dynamics (concentration and limitation types) to phytoplankton composition (diatom ratio and temporal variations in total primary production) and oxygen variables (SOC, DO profiles, and hypoxia patterns). Then, we organized our discussion from nutrient to plankton and DO based on hindcast snapshots of various physical and biogeochemical metrics. During each part, we post observational evidence and related discussions to support our findings.

L1392: I don't think that meets the BG requirement
Responses: The model data is big (in several TBs); we will, of course, share all hindcast results online via a valid link.